



# Newly dated permafrost deposits and their paleo-ecological inventory reveal a much warmer-than-today Eemian in Arctic Siberia

Lutz Schirrmeister[1], Margret C. Fuchs[2], Thomas Opel[3], Andrei Andreev[3], Frank Kienast[4], Andrea Schneider[5], Larisa Nazarova[6], Larisa Frolova[7, 21], Svetlana Kuzmina[8], Tatiana Kuznetsova[9], Vladimir Tumskoy[10], Heidrun Matthes[11], Gerrit Lohmann[12], Guido Grosse[1, 13], Viktor Kunitsky[10], Hanno Meyer[3], Heike H. Zimmermann[14], Ulrike Herzschuh[3, 15], Thomas Böhmer[3], Stuart Umbo[16], Sevi Modestou[16], Sebastian F.M. Breitenbach[16], Anfisa Pismeniuk[17], Georg Schwamborn[18], Stephanie Kusch[19], Sebastian Wetterich[1, 20]

[1]Permafrost Research Section, Alfred Wegener Institute, Helmholtz Center for Polar and Marine Research, Potsdam, Germany

[2]Exploration, Helmholtz Institute for Resource Technology, Freiberg, Germany

[3]Polar Terrestrial Environmental Systems Section, Alfred Wegener Institute, Helmholtz Center for Polar and Marine Research, Potsdam, Germany

[4]Quaternary Paleontology, Senckenberg Research Institute and Natural History Museum, Weimar, Germany

[5]Arctic University of Norway, Tromsø, Norway

[6]Krasnoyarsk Science Center, SB RAS, Krasnoyarsk, Russia

[7]Kazan Federal University, Russia

[8]Boryssyak Paleontological Institute, RAS, Moscow, Russia

[9]Faculty of Geology, Lomonosov Moscow State University, Russia

[10]Mel'nikov Permafrost Institute, SB RAS, Yakutsk, Russia

[11]Atmospheric Physics Section, Alfred Wegener Institute, Helmholtz Center for Polar and Marine Research, Potsdam, Germany

[12]Paleoclimate Dynamics Section, Alfred Wegener Institute, Helmholtz Center for Polar and Marine Research, Bremerhaven, Germany

[13]Institute of Geosciences, University of Potsdam, Germany

[14]Glaciology and Climate Department, Geological Survey of Denmark and Greenland, Copenhagen, Denmark

[15]Institute of Environmental Science and Geography, Institute of Biochemistry and Biology, University of Potsdam, Germany

[16]Geography and Environmental Sciences, Northumbria University, Newcastle, UK

[17] Department of Geosciences, University of Oslo, Oslo, Norway

[18]Eberswalde University for Sustainable Development, Department of Landscape Management and Nature Conservation, Eberswalde, Germany

[19]Institute of Marine Sciences (ISMER), University of Quebec Rimouski, Rimouski, Canada

[20]Current address: Institute of Geography, Technische Universität Dresden, Germany

[21]Institute of Archaeology and Ethnography, SB RAS, Novosibirsk, Russia

*Correspondence to*: Lutz Schirrmeister (lutz.schirrmeister@awi.de)

**Keywords: Marine Isotope Stage 5e, luminescence dating, cryolithology, paleo-proxy, model-based climate reconstructions, MTWA, MTCO, MAP, clumped isotopes**



**Abstract.** Fossil proxy records in Last Interglacial (LIG, ca. 130-115 ka) lacustrine thermokarst deposits now
preserved in permafrost can provide insights into terrestrial Arctic environments during a period when northern
hemisphere climate conditions were warmer than today and which might be considered a potential analog for a
near-future warmer Arctic. Still, such records are scarce on a circum-Arctic scale and often poorly dated. Even
more, the quantitative climate signals of LIG permafrost-preserved deposits have not yet been systematically
explored.
Here, we synthesize geochronological, cryolithological, paleo-ecological, and modeling data from one of the most
thoroughly studied LIG sites in NE Siberia, the permafrost sequences along the coasts of the Dmitry Laptev Strait,
i.e., on Bol'shoy Lyakhovsky Island and at the Oyogos Yar coast. We provide chronostratigraphic evidence by
new luminescence ages from lacustrine deposits exposed at the southern coast of Bol'shoy Lyakhovsky Island.
The infrared-stimulated luminescence (IRSL) ages of 127.3±6.1 ka, 117.8±6.8 ka, and 117.6±6.0 ka capture the
MIS 5e sub-stage, i.e., the LIG.
The LIG lacustrine deposits are mostly preserved in ice-wedge pseudomorphs of 1-3 m thickness with alternating
layers of peaty plant detritus and clayish silt. Ripples and synsedimentary slumping structures indicate shallow-
water conditions. The rich fossil record was examined for plant remains (macro-fossils, pollen, *seda*DNA), lipid
biomarkers, and aquatic and terrestrial invertebrates (cladocera, mussels, snails, ostracods, chironomids, and
beetles).
Most proxy data and also paleoclimate model results indicate a regional LIG climate significantly (ca. 5 to 10 °C)
warmer than today. Plant macrofossil data reflect mean temperatures of the warmest month (MTWA) of 12.7–15.3
°C for Oyogos Yar and 10.3–12.9 °C for Bol'shoy Lyakhovsky, while pollen-based reconstructions show mean
MTWA of 9.0±3.0 °C and 9.7±2.9 °C as well as mean annual precipitation (MAP) of 271±56 mm and 229±22
mm, respectively. The biomarker-based reconstruction of the Air Growing Season Temperature (Air GST) using
GDGTs is 2.8±0.3 °C. The fossil beetle-based mutual climatic range is 8 to 10.5 °C for MTWA and –34 to –26 °C
for the mean temperature of the coldest month (MTCO) on Bol'shoy Lyakhovsky Island and 8 to 14 °C for MWTA
and –38 to –26 °C for MTCO on Oyogos Yar. The chironomid-based MTWA varies between 9.4±1.7 and 15.3±1.5
°C and the water depth (WD) between 1.7±0.9 and 5.6±1.0 m on Bol'shoy Lyakhovsky Island. Prior findings from
Oyogos Yar in the literature suggest an MTWA of 12.9±0.9 °C and a WD of 2.2±1.1 m. The first-time application
of clumped isotopes to permafrost-preserved biogenic calcite of ostracods and bivalves reconstruct near-surface
water temperature of 10.3±3.0 °C and bottom water temperatures of 1.5±5.3 °C in thermokarst lakes during
summers. PaleoMIP Model simulations (PIobs+(lig127k-PI)) of the LIG show warmer MTWA compared to
modern conditions (by 4.4±1.0 °C for Bol'shoy Lyakhovsky and 4.5±1.2 °C for Oyogos Yar) but currently
underestimate the Eemian warming reconstructed from our multiple paleoecological proxies.
The LIG warming mainly affected summer conditions, whereas modern and future warming will rather impact
winter conditions. As the LIG annual mean temperature is often used as an analog for the future climate in the
High Arctic, the proxy-model mismatch highlights the urgent need for more systematic quantitative proxy-based
temperature reconstructions in the Arctic and more sophisticated Earth system models capable of capturing Arctic
paleoenvironmental conditions.



# 1 Introduction

The climate control on permafrost dynamics during the late Quaternary is reflected in large-scale permafrost aggradation during glacial periods and extensive thaw in interglacial periods (Shur and Jorgenson, 2007; Jones et al., 2023; Opel et al., 2024). These broader climate-driven dynamics were superimposed by local factors, including topography, hydrology, and vegetation cover, as well as natural disturbances causing localized rapid thaw such as thermokarst lakes or wildfires. During warm periods, rising air temperatures and, consequently, rising ground temperatures and active-layer thickening promote ground-ice melting, surface subsidence, and the formation of thermokarst lakes and basins in areas underlain by ice-rich permafrost (Czudek and Demek, 1970; Grosse et al., 2013; Brosius et al., 2021). Shallow thermokarst lakes might develop that can preserve traces of landscape evolution, paleoclimate, and paleoecology in lacustrine sediments (Murton, 1996; Lenz et al., 2016; Bouchard et al., 2017). Thawing ice-wedge polygons below the lake-water level form ice-wedge casts (pseudomorphs), which may also preserve laminated lacustrine deposits (Farquharson et al., 2016). Lake drainage or desiccation over time promotes the formation of typically palustrine (peaty) deposits above the lacustrine sequences in thermokarst basins (e.g., Morgenstern et al., 2013). However, permafrost aggradation and degradation are not restricted to glacial and interglacial periods, respectively. At least on the local to regional scale, permafrost formation is also reported for interglacial times such as associated with pingo formation (Wetterich et al., 2018) or peat growth during the Holocene (Kaplina, 2011a, 2011b; Wetterich et al., 2009) and permafrost degradation occurred during interstadial stages of glacial periods (Vaks et al., 2020, 2013).

The globally warmer-than-today Last Interglacial (LIG, ca. 130-115 ka) is commonly seen as a potential analog for future climate warming. Due to summer insolation higher than today and additional feedback effects such as the retreat of ice sheets, reduction of summer sea ice, and expansion of boreal forests, the LIG warming in the Arctic was amplified compared to the Northern Hemisphere as a whole (CAPE-LIG Project Members, 2006). Recent studies combining proxy evidence across the Arctic (mainly from terrestrial and lacustrine pollen and plant macrofossils and Greenland ice cores) and paleoclimate modeling largely agree on 4-5 °C higher summer temperatures than today and a nearly sea-ice free Arctic Ocean during summers (Guarino et al., 2020; Sime et al., 2023; Vermassen et al., 2023), accompanied by a High Arctic greening (Crump et al., 2021). For a reduced Greenland Ice Sheet, up to +5 °C regional warming is modeled for the LIG (Pfeiffer and Lohmann, 2016). However, it is important to note that the warmer than today conditions during the LIG were predominantly driven by increased summer solar insolation, unlike modern Arctic warming, which is most pronounced in winter due to anthropogenic forcing. Despite this difference, many of the consequences of summer warming in the LIG – such as ice sheet retreat, reduction of sea ice, and permafrost degradation – are observable in the changing Arctic today, supporting the continued relevance of the LIG as an analog setting. There are only limited quantitative paleoclimate data on the terrestrial interglacial MIS 5e conditions in the Arctic (e.g., CAPE-LIG Project Members, 2006; Sime et al., 2023), with sparse sites that are widely spaced and which often have only single or few selected paleoproxies and rather poor temporal constraints. Part of the mismatch between model and proxy data can be reconciled by considering the potential seasonal bias of the proxy record and/or the uncertainties in the dating of the proxy records for the LIG thermal maximum (Pfeiffer and Lohmann, 2016). Many terrestrial records of the LIG in the Arctic are found in fluvial deposits exposed in river bluffs or artificial placer mining cuts in Alaska (Hamilton & Brigham-Grette, 1991; Edwards et al., 2003; Jensen et al., 2013), Siberia (Velichko et al., 2008), or NW Canada (Reyes et al., 2010). As interglacial climate conditions degraded pre-existing ice-rich permafrost deposits and also promoted the formation of thermokarst basins and lakes, the interglacial legacy in permafrost regions is also

commonly represented by thaw unconformities and lacustrine and palustrine deposits. Some of these sites
attributed to the LIG are known from eastern Siberia in the Yana-Indigirka Lowland (site Achagy-Allaikha,
Kaplina, 1980, 1981) and the Kolyma Lowland (site Duvanny Yar; Kaplina, 2011a). Furthermore, the world's
currently largest permafrost degradation feature, the Batagay megaslump in the Yana Upland in eastern Siberia,
exposes a distinct woody debris layer below an erosional disconformity of Marine Isotope Stage (MIS) 5e age,
i.e., the LIG that is overlain by MIS 4 to MIS 2 Yedoma Ice Complex deposits (Ashastina et al., 2017; Murton et
al., 2022).
Extensive late Pleistocene paleoecological studies on ice-wedge pseudomorphs and other lacustrine sequences
were conducted on coastal exposures at both coasts of the Dmitry Laptev Strait that connects the Laptev and the
East Siberian seas (Figure 1). Both the southern coast of Bol'shoy Lyakhovsky Island near the Zimov'e River
mouth and the Oyogos Yar mainland coast near the Kondrat'eva River mouth have been studied for LIG pollen,
plant macrofossils, fossil insect remains including beetles and chironomids, lacustrine invertebrates such as
ostracods, cladocera, and mollusks, and testate amoebae and sedimentary ancient DNA (Andreev et al., 2004,
2011; Ilyashuk et al., 2006; Kienast et al., 2008, 2011; Wetterich et al., 2009; Schneider, 2010; Zimmermann et
al., 2017). However, only scarce chronological control is available (Andreev et al., 2004; Opel et al., 2017) for
lacustrine deposits, which are locally named the Krest-Yuryakh stratum (Tumskoy and Kuznetsova, 2022) and
commonly interpreted as deposits of the LIG, i.e., MIS 5e (Eemian). Mammal bones belonging to the late
Pleistocene Mammoth fauna in Krest-Yuryakh deposits are very rare and poorly dated (Kuznetsova et al., 2022).
Ancient thermokarst deposits and their paleoenvironmental inventory can provide interesting analogs for modern
and (near-)future warming and its impact on periglacial landscapes and ecosystems. During the Lateglacial to
Holocene transition, thermokarst processes re-shaped vast areas of the Beringian periglacial landscapes by thawing
the MIS 4 to MIS 2 Yedoma Ice Complex deposits (Jones et al., 2022; Walther Anthony et al., 2014, Morgenstern
et al., 2013), and paleoecological records from past warming episodes such as the Lateglacial Interstadials (e.g.,
Allerød) and the early Holocene with rapid permafrost degradation are fairly well characterized, helping to
understand interactions between climate, permafrost, and ecosystem dynamics. For example, previous
paleobotanical studies associated with thermokarst deposits from NE Siberia revealed quantitative estimates of the
Lateglacial-early Holocene warming with a pollen-based mean temperature of the warmest month (MTWA) of 8-
12 °C (Andreev et al., 2009, 2011). Likewise, the LIG climate warming led to the degradation of previously
aggraded ice-rich permafrost deposits by thermokarst processes. A LIG (MIS 5e) macrofossil-based reconstruction
of MTWA reached >12.5 °C (Kienast et al., 2008, 2011), suggesting the potential for more intense warming during
MIS 5e, which in turn could be considered – at least in the amplitude – a representative analog for stronger modern
warming in the Arctic than during the Lateglacial or early Holocene.
According to Rovere et al. (2016), the global sea level of MIS 5e might have reached levels 5 to 9.4 m higher than
today. For north-eastern Siberia, especially for the Laptev Sea and the East Siberian Sea coastal regions, only
scarce information on the MIS 5e coastline is available and outlined below. According to Ivanenko (1998), a
distinctive feature of the Fadeevsky and Novaya Sibir islands (especially at their northern coasts; Figure 1) are
widely distributed marine deposits (called the Kanarchak Formation), which are overlain by terrestrial Yedoma
Ice Complex or, rarely, by terrestrial Holocene deposits. A smooth boundary between the marine and terrestrial
strata is striking, indicating a continuous sedimentation regime and uninterrupted transition from marine to
terrestrial conditions. According to Alekseev et al. (1991a, 2001), the lower part of the sections consists of marine
terrace deposits, assumed to have formed during the LIG (Kazantsevo period in Russian). The sandy series





corresponding to the Kazantsevo transgression overlays the marine terrace deposits, which are today situated at 8-
10 m above sea level (a.s.l.). Remnants of the marine terrace deposits with mollusk fauna are widely distributed
and well expressed on the Novaya Sibir, Fadeevsky, and Kotel'ny islands, mainly in the estuarine parts of river
valleys. Later studies on Novaya Sibir and Fadeevsky islands were undertaken by Basilyan et al. (2010), Tumskoy
(2012), and Nikolskiy et al. (2017), including first detailed investigations of the Kanarchak Formation that contains
a prominent massive ground ice body. These studies recognized the massive ground ice as a relic of a late mid-
Pleistocene (MIS 6) glaciation, and the upper part of the Kanarchak Formation was assigned to the LIG (Basilyan
et al., 2010). We conclude that the northern coast of the Fadeevsky and Novaya Sibir islands is characterized by
marine and coastal deposits of the LIG, which delineate the approximate position of the coastline during MIS 5e.
Linking the paleoecologic records from the different locations in order to understand the regional context requires
a robust geochronological framework. While Lateglacial and early Holocene thermokarst deposits are commonly
well-constrained via radiocarbon dating (Wetterich et al., 2009), the chronostratigraphy of MIS 5 deposits
attributed to the LIG suffers from large dating uncertainties of the available dating methods that include
radioisotope disequilibria ($^{230}$Th/U) of peat (Schirrmeister et al., 2002; Wetterich et al., 2016) and optically
stimulated luminescence of quartz (OSL) or infrared stimulated luminescence of feldspar (IRSL) (Andreev et al.,
2004; Opel et al., 2017). Therefore, the highly variable millennial climate dynamics from about 130 to 80 ka during
MIS 5 (expressed as MIS 5 sub-stages 5e to 5a; Shackleton et al., 2003) are not yet resolved in terrestrial permafrost
records and hinder the paleoclimatic interpretation of permafrost-preserved fossil proxy records.
Our study summarizes previously published and newly obtained data from coastal permafrost exposures at both
coasts of the Dmitry Laptev Strait to (1) provide new luminescence dates that constrain the timing of the LIG and
thus to resolve regional MIS 5e climate variability better; (2) summarize cryolithological and geochemical
characteristics of LIG deposits that capture depositional processes and preservation conditions; (3) deduce the
ecological response and quantify the paleoclimatic parameters linked to LIG warming as reflected by fossil proxy
data of vegetation, terrestrial and aquatic invertebrates, and compare them with results of climate model
simulations; and (4) discuss LIG climate-ecology-permafrost dynamics and their potential as analogs for the
ongoing and future climate warming in the terrestrial Siberian Arctic.
**2 Study sites**
The study sites stretch along the Dmitry Laptev Strait at the southern coast of Bol'shoy Lyakhovsky Island and the
opposite mainland coast of Oyogos Yar (Figure 1). Various coastal outcrops, thaw slumps, and drill sites were
studied along an approximately 16.5 km long section west and east of the Zimov'e River mouth on Bol'shoy
Lyakhovsky (Figure 1b; Figure 2a) and along a 5.5 km long section west of the Kondrat'eva River mouth at the
Oyogos Yar mainland coast (Figure 1c; Figure 2b). A recent review of the permafrost research history of the
Dmitry Laptev Strait shores is provided in Tumskoy and Kuznetsova (2022).
The modern climate of the study area is characterized by short, cold summers with a mean July temperature
(MTWA) of 2.5 to 2.8 °C (WMO stations 216470, 216360). Long, harsh winters of eight months are characterized
by low light availability and low temperatures, with a mean January temperature (mean temperature of the coldest
month, MTCO) of −34.4 to −33.1°C. The annual precipitation varies from 243 to 262 mm, yet overall conditions
are humid due to low evaporation rates and poor drainage of the wet active layer during summer (Hersbach et al.,
2020, ERA5, 1990-2019). The area belongs to the Arctic tundra subzone (Chernov and Makarova, 2008), more



specifically, the moist to dry tundra vegetation zone with open to continuous plant cover (G2) on Bol'shoy
Lyakhovsky Island and sedge/grass, moss wetland (W1) on the Oyogos Yar coast (CAVM Team, 2003).
The region is underlain by continuous permafrost with a thickness of 400-600 m, and the mean annual ground
temperature is about −14 to −12 °C (Yershov, 1989). The mean thickness of the active layer is about 30-40 cm
(Schwamborn and Wetterich, 2015).

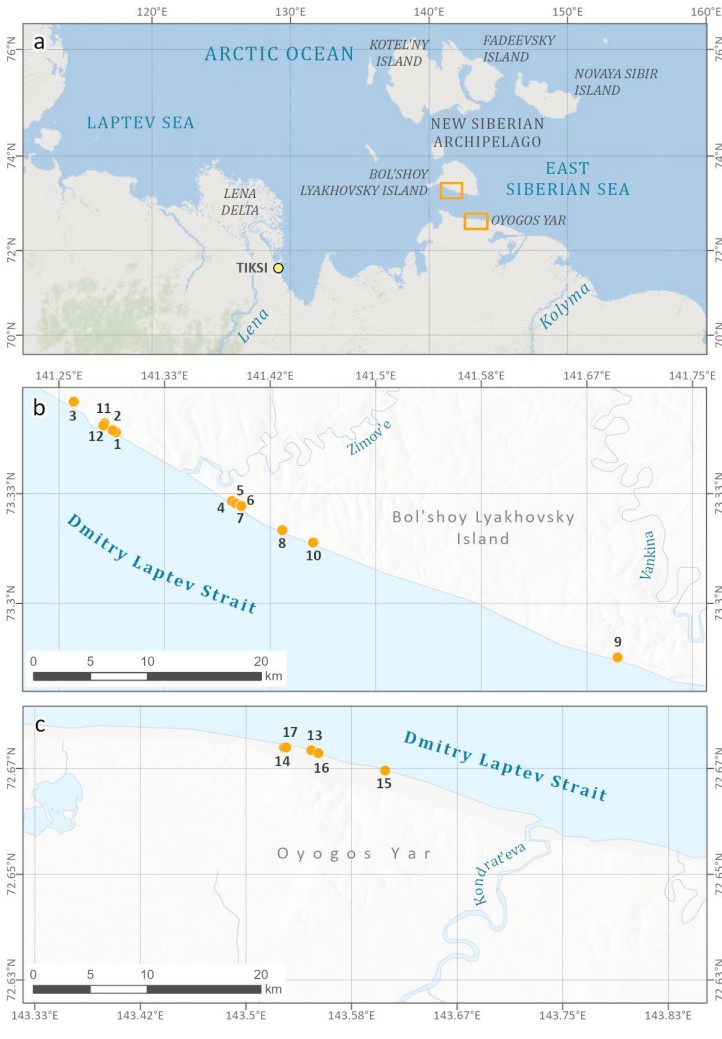

**Figure 1.** Study area (a) in northeastern Siberia along the Dmitry Laptev Strait, (b) at the southern coast of Bol'shoy
Lyakhovsky Island, and (c) the opposite mainland coast of Oyogos Yar. The locations of sampling profiles of LIG
deposits are also indicated in Table A1 and Figure 2. The maps were compiled by Sebastian Laboor, AWI Potsdam,
using World Imagery (Credits: Esri, Maxar, GeoEye, Earthstar Geographics, CNES/Airbus DS, USDA, USGS,
AeroGRID, IGN, and the GIS User Community) and World Ocean Base (Credits: Esri, Garmin, GEBCO, NOAA
NGDC, and other contributors).





In its western, central, and eastern parts, Bol'shoy Lyakhovsky Island is shaped by hills reaching elevations of
about 100 to 300 m a.s.l. In the central part and the coastal region, Yedoma uplands up to 40 m a.s.l. are present
and dissected by large thermokarst basins (alases), wide and flat thermo-erosional valleys (logs), gullies or ravines
(ovrags). The southern coast of Bol'shoy Lyakhovsky Island is characterized by vast retrogressive thaw slumps
(thermo-cirques), the mouth of the Zimov'e River, and numerous smaller streams. Widespread thermokarst
characterizes the Oyogos Yar mainland coast (Günther et al., 2013), covered by polygonal peatlands, shallow
thermokarst lakes, and erosional remnants of Yedoma uplands. The general stratigraphy of both coastal sections
with deposits between MIS 7 and MIS 1 is presented in Figure 2.

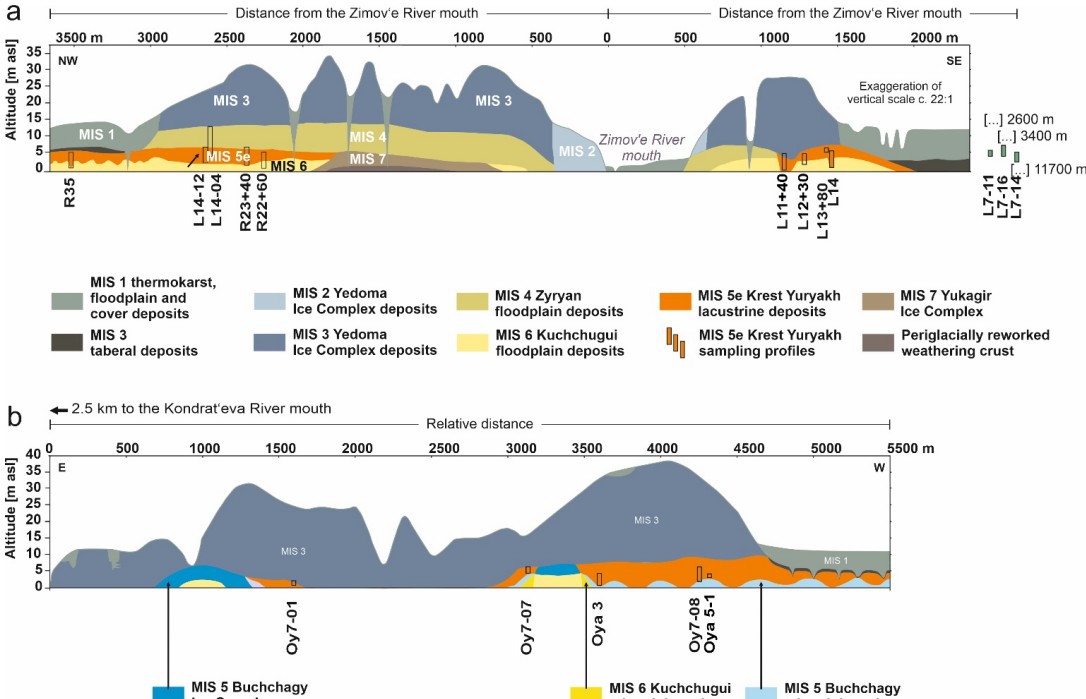

**Figure 2. Cryostratigraphic schematics of both coasts of the Dmitry Laptev Strait with LIG (Krest-Yuryakh) profile**
**locations: (a) Bol'shoy Lyakhovsky Island (re-drawn from Andreev et al., 2009 and Wetterich et al., 2021), (b) Oyogos**
**Yar coast (re-drawn from Tumskoy and Kuznetsova, 2022).**
**3      Material and methods**
**3.1     Fieldwork**
Field studies on both sides of the Laptev Strait were conducted in the summers of 1999, 2002, 2007, and 2014, as
well as in spring of 2014 (Schirrmeister et al., 2000, 2003; Boike et al., 2008; Schwamborn and Wetterich, 2015).
After an initial survey, selected coastal exposures (Figure 2) were sampled in detail. Vertical profiles were cleaned
with spades and hoes to remove the outermost thawed material. The exposed sequences were surveyed, described,
photographed, and sketched to document sediment structures, cryostructures, color, and visible organic content.
The frozen deposits were sampled for further studies using hammers and small axes. In spring 2014, a Russian



drill rig (KMB3-15M) mounted on an all-terrain vehicle was used to retrieve permafrost cores on Bol'shoy
Lyakhovsky Island using a rotary mechanism in dry holes.
Sections of Krest-Yuryakh deposits sampled at the southern coast of Bol'shoy Lyakhovsky Island included the
profiles R35, R22+60, R23+40, L11+40, L12+30, L13+80 and L14 in 1999; the profiles L7-14, L7-11 and L7-16
in 2007; and the profile L14-12 and the permafrost core L14-04 in 2014 (Figure 2; Table A1). On the Oyogos Yar
coast, the profiles Oya-3-10, Oya-3-11, and Oya 5-1 were obtained in 2002, and the profiles Oy7-01, Oy7-07, and
Oy7-08 A-C in 2007 (Figure 2; Table A1).
Except for the permafrost core, the ice content of all permafrost samples was determined in the field in closed
aluminum boxes. The samples were weighed while still frozen, dried in a field oven, and weighed again. The
absolute gravimetric ice content is the ratio of the ice mass in a sample to the dry sample mass, expressed as a
percentage (van Everdingen, 2005).
In 1999, screening for rodent remains using a woody screen box with a metallic screen (1mm mesh) and a motor
pump was carried out. Although no rodents were found, the screened sediment was used to extract insects. In
addition, insects were collected by screening the thawed sediment with a 0.4 mm mesh. Plant macrofossils were
excavated from insect samples and large (up to 2 kg) bulk samples. Large mammal bones were collected from the
section and below at the seashore. Some small rodent teeth were picked up from mineral and plant debris below
the section.

### 3.2    Luminescence dating

Previous luminescence sampling in 1999 on Bol'shoy Lyakhovsky is described in Andreev et al. (2004, 2007) and
on Oyogos Yar in Opel et al. (2017). In 2014, Krest-Yuryakh deposits were sampled at the southern coast of
Bol'shoy Lyakhovsky Island for luminescence dating (profile L14-12, 73.34055°N, 141.28498°E; Figure 1). After
cleaning and cryolithological description, the ca. 7 m high steep exposure was sampled for luminescence dating at
heights of 4.5 m a.s.l. (L14-12-OSL1) and 2.7 m a.s.l. (L14-12-OSL3) (Table A1). We used a HILTI TE6 - A36
cordless rotary hammer to obtain frozen cores protected from sunlight that were packed dark, stored in an ice cellar
next to the camp at –4 °C, and kept frozen until arrival in the laboratory.
The processing of sample cores aimed at extracting K-feldspars with grain sizes of 40–63 μm (L14-12-OSL1, L14-
12-OSL3) and 63–90 μm (L14-12-OSL1). Coarser material (>100 μm) was lacking, or only a minor component,
and the quartz OSL signals were close to saturation. Pure mineral extracts were obtained via carbonate and organic
removal (10% HCl and 30% $H_2O_2$, respectively), feldspar flotation (0.2% HF, pH 2.4–2.7, dodecylamine) for
efficient separation from quartz, density separation to enrich K feldspars (2.53–2.58 $\text{g cm}^{-3}$) and 5 min final etching
in 10% HF. Extracted medium-sized feldspar grains were used to prepare sets of 24 aliquots with a 2 mm diameter.
Measurements and analyses for age estimation were based on 20 aliquots, while four aliquots per sample and grain
size fraction were used for quality tests. For L14-12-OSL3, five additional aliquots were used in equivalent dose
screening sequences to optimize the regenerative dose points. The IRSL signals of feldspars were measured using
a TL/OSL DA-20 reader (Bøtter-Jensen et al., 2003) equipped with a $^{90}$Sr beta irradiation source (4.95 $\text{Gy min}^{-1}$).
Signals were stimulated at 870 nm (IR diodes, 125°C for 100 s) and detected through a 410 nm optical interference
filter (Krbetschek et al., 1997).
The measurement sequence followed the single-aliquot regenerative dose (SAR) protocol according to Murray
and Wintle (2000), including cycles to record recycling ratios and recuperation and to correct for sensitivity



changes. Preheat and cut-heat temperatures were set to 230 °C and 160 °C, respectively, according to preheat tests.
Emitted signals were recorded for 100 s to ensure the acquisition of pure background signals at the end of each
measurement cycle. Dose-recovery tests (Murray and Wintle, 2003) confirmed suitable luminescence properties
under the chosen conditions with coefficients of variations between 3.8% and 5.3%. The data processing was
performed using the R package "Luminescence" (Kreutzer et al., 2012), version 0.9.20.
The datasets (n = 20 to 25) revealed equivalent dose distributions of low skewness (below 0.8%) and low standard
deviations (below 5%). No evidence of insufficient bleaching or significant post-depositional mixing was found,
and hence, paleodose estimates were based on the Central Age Model (CAM) (Galbraith et al., 1999). The CAM-
based paleodose was then processed together with all sample-specific dose rates (sediment-internal, mineral-
internal, cosmic) and correction (grain sizes, water content, sediment cover, etc.) parameters using the online
DRAC calculator (Durcan et al., 2015). For the mineral-internal dose rate of the K feldspars, a potassium content
of $12.5 \pm 0.5\%$ was assumed (Huntley and Baril, 1997).

## 3.3 Sediment analyses

In the laboratory, the sediment samples were freeze-dried, carefully manually homogenized, and split into sub-
samples for sedimentological, geochemical, and paleo-ecological analyses. Grain-size distribution was analyzed
using a laser-particle analyzer (Beckmann Coulter LS 200) and computed with GRADISTAT 4.0 software (Blott
and Pye, 2001). The mass-specific magnetic susceptibility (MS) was measured using a Bartington MS2 instrument
(MS2B sensor), and values are given in SI units (Le Système International d'Unités; $10^{-8}\,\mathrm{m^3\,kg^{-1}}$). The total organic
carbon (TOC) and total nitrogen (TN) contents were measured by a carbon-nitrogen-sulfur (CNS) analyzer
(Elementar Vario EL III), and the total organic carbon to total nitrogen ratio was calculated as TOC/TN ratio.
Stable carbon isotopes ($\delta^{13}$C) of TOC were measured until 2014 with a Finnigan DELTA S mass spectrometer
coupled to a FLASH element analyzer and a CONFLO III gas mixing system after the removal of carbonates with
10% HCl in Ag-cups and combustion to $CO_2$. The accuracy of the measurements was determined by parallel
analysis of internal and international standard reference material. The analyses were accurate to $\pm 0.2$ ‰. Later, the
measurement was undertaken using a Thermo Scientific Delta V Advantage isotope ratio MS equipped with a
Flash 2000 organic elemental analyzer using helium as a carrier gas. The accuracy was better than $\pm 0.15$ ‰. The
$\delta^{13}$C values are expressed in delta per mil notation ($\delta$, ‰) relative to the Vienna Pee Dee Belemnite (VPDB)
standard.
Values are given as per mil (‰) difference from the Vienna Pee Dee Belemnite (VPDB) standard for $\delta^{13}$C and
from nitrogen in ambient air (AIR) for $\delta^{15}$N. The accuracy was better than $\pm 0.15$ ‰ for $\delta^{13}$C and $\pm 0.2$ ‰ for $\delta^{15}$N.

## 3.4 Paleo-ecological analyses and paleoclimate reconstructions

Pollen data are available (Table A1) from Bol'shoy Lyakhovsky Island from Krest-Yuryakh deposits in profiles
R22+60, L11+40, L12+30 and L14 (Andreev et al., 2004), R23+40 (Andreev et al., 2009), R35 (Ilyashuk et al.,
2006), L7-14 (Wetterich et al., 2009), and in the drill core and hand pieces of the permafrost core L14-04
(Zimmermann et al., 2017). On Oyogos Yar, pollen data were obtained from profiles Oy7-08 (Wetterich et al.,
2009) and Oya 5-1 (Kienast et al., 2011). The pollen sample preparation followed standard methods (e.g., Andreev
et al., 2004).



Pollen-based climate reconstructions were based on a northern hemispheric modern pollen training dataset
comprising 15,379 sites in Eurasia and North America (Herzschuh et al., 2023). Only terrestrial pollen taxa
(including Cyperaceae) were used for reconstructions, while aquatic pollen taxa, as well as spores from mosses,
ferns, fungi, and algae, were excluded. Woody taxa, as well as some very common herbaceous taxa (e.g.,
Artemisia, Thalictrum, or Rumex), were harmonized to the genus level, and all other herbaceous taxa were
harmonized to the family level (Herzschuh et al., 2022). Site-specific mean July temperatures ($T_{July}$) and annual
precipitation ($P_{ann}$) were derived from WorldClim 2 version 2.1 (https://www.worldclim.org; Fick and Hijmans,
2017) by extracting the climate data at the location of the modern samples. The pollen taxa in the fossil pollen
samples noted above were harmonized in the same way as the taxa in the modern training dataset. For each
location, we calculated the geographic distance between each sampling site in the modern training dataset and the
fossil pollen record. Climate reconstructions were performed using the modern analog technique (MAT, Overpeck
et al., 1985) by applying the MAT function from the rioja package (version 0.9-21, Juggins, 2019) for R (R Core
Team, 2020) to the pollen percentages of the selected fossil pollen taxa, looking for seven analogs between the
pollen data and the calibration dataset. The dissimilarity between the fossil samples and the modern pollen
assemblages was determined by the squared-chord distance of the percentage data (Cao et al., 2014; Simpson,

16   2012).

Plant macrofossils were examined (Table A1) in profiles R22+60, L12+30, R35 (Kienast et al., 2008), and L7-11
(Schneider, 2010) on Bol'shoy Lyakhovsky Island and in profile Oya 5-1 on Oyogos Yar (Kienast et al., 2011).
Sample preparation followed standard methods (e.g., Kienast et al., 2008). The MTWA tolerances of plant species
identified by macrofossils were calculated by correlating their modern distribution in Yakutia, pooled in the online
database GBIF (2023), which comprises geocoded distribution data from maps published in the Flora of Siberia
(Artemov and Egorova, 2021), permanently updated iNaturalist research-grade observations (iNaturalist, 2023)
and records from literature on local floras of Russia (Bochkov and Seregin, 2022) with monthly mean temperatures
from the updated database of Leemans and Cramer (1991). We considered only the Yakutian distribution of
recovered plant species because of the relative climatic stability of
Yakutia throughout the late Quaternary and the related conservative genetic configuration of plant populations in
Yakutia with a low percentage of polyploidy and a narrower ecological tolerance in comparison with more western
populations. The temperature range of a certain species was determined by the correlation of dot occurrences
within Yakutia published in GBIF (2023) with the mean July temperature as MTWA at the closest grid point with
a resolution of 0.5° longitude/latitude (Leemans and Cramer, 1991). The temperature extremes within the Yakutian
distribution range, e.g., the northernmost occurrence as the minimum and the presence in the Central Yakutian
Plain with an MTWA of up to 18.4 °C as the possible maximum, reveal the temperature range of a species. The
minimum requirement for MTWA of the most thermophilous species, together with the maximum MTWA
tolerance of the most cold-adapted plant within a paleo-flora, reveal the temperature interval (or mutual climatic
range, MCR) for the coexistence of all species. We focused on species with particularly high (boreal) and low
(arctic) temperature demands (Table S1).
Sedimentary ancient DNA (sedaDNA) refers to the desoxyribonucleic acid (DNA) preserved in sedimentary
deposits that stem from biological material such as plants, animals, and microorganisms that live in or near the
depositional environment. In permafrost, sedaDNA is predominantly local in origin (Alsos et al., 2018), and its
source can be derived from preserved plant tissues, from extracellular DNA bound to mineral particles, and from
feces, and only to a lesser degree from pollen (Crump, 2021). That allows for the reconstruction of past community





composition, diversity, and temporal dynamics up to geological timescales (Courtin et al., 2022; Kjær et al., 2022).
We used DNA metabarcoding to amplify and sequence a short plant-specific genetic marker (see details in
Zimmermann et al., 2017). DNA metabarcoding was successfully applied (Table A1) to seven samples of the
permafrost core L14-04, three samples from profile L14-04-B, and seven samples of profile L14-04-C
(Zimmermann et al., 2017).
Lipid biomarkers were analyzed in permafrost core L14-04. A total of 10 samples were processed, and microbial
ether lipids (branched and isoprenoid glycerol dialkyl glycerol tetraethers; brGDGTs, isoGDGTs) were analyzed
as described by Kusch et al. (2019). The methylation index of 5-methyl branched tetraethers ($MBT'_{5ME}$) was
calculated using $MBT'_{5ME}$ = (Ia + Ib + Ic)/(Ia + Ib + Ic +IIa + IIb + IIc + IIIa) (de Jonge et al., 2014). The Isomer
Ratio (IR) of penta- and hexamethylated brGDGTs was calculated as IR = (IIa′ + IIIa′)/(IIa + IIIa + IIa′ + IIIa′)
(Yang et al., 2015). We use the $MBT'_{5ME}$-based Air Growing Season Temperature (Air GST, April to October)
calibration (calibration D; Air GST = –3.82 + 22.71 *$MBT'_{5ME}$ + 8.78 * IR) recently developed by de Jonge et al.
(2024) since it is the only calibration available that accounts for seasonal production bias by including soils frozen
during part of the year and corrects for the influence of pH on $MBT'_{5ME}$. This calibration has a residual square
mean error of 2.2°C. The branched and isoprenoid tetraether index (BIT; Hopmans et al., 2004) was calculated
following BIT = (Ia + IIa + IIIa + IIa' + IIIa')/(Ia +IIa +IIIa +IIa' + IIIa' + crenarchaeol), and the ratio of isoGDGTs
to brGDGTs (Ri/b; Xie et al., 2012) was calculated using Ri/b = ΣisoGDGTs/ΣbrGDGTs. For details about the
chemical structures and nomenclature of GDGTs, we refer to Kusch et al. (2019).
Terrestrial insect remains (mostly beetles) were studied (Table A1) in four samples from Bol'shoy Lyakhovsky
Island (L-11-B17, L-11-B19, L-12+30-B-18, R-22-B15, R-22-B16; Andreev et al., 2004, 2009; Kuzmina, 2015b)
and one sample from Oyogos Yar (Oya 5-1; Kienast et al., 2011). Sample preparation for terrestrial insect fossils
followed standard methods (Kuzmina, 2015b). The MCR method described above for plant macrofossils allows
reconstructing the MTWA and MTCO of the year by overlapping coexistence intervals of several species of insects
(or any other taxa) in single samples (Atkinson et al., 1987). The MCR method is widely used on fossil beetle
remains in Europe mainly because of the continuous research project and database BugsCEP (Buckland, 2007,
2014). The method has been adapted for North America (Elias, 2000, 2001) for the study of the late Quaternary
beetle fauna of East Beringia. To evaluate the LIG warming in the Dmitry Laptev Strait region, two sources of
thermal requirements are used, which are a West Beringian list (including phytophagous species; Alfimov et al.,
2003) and a Transberingian list (excluding phytophagous species; Elias, 2000), both based on museum collections.
Chironomids were studied (Table A1) in profile R35 (Ilyashuk et al., 2006) on Bol'shoy Lyakhovsky Island and
profile Oya 5-1 on Oyogos Yar (Kienast et al., 2011). Chironomid sample preparation followed standard methods
(Brooks et al., 2007). For the paleotemperature reconstruction from chironomid data, we inferred the $T_{July}$
(MWTA) by using a North Russian (NR) chironomid-based temperature inference model (WA-PLS, 2 component;
$r^2$ boot = 0.81; RMSEP boot=1.43°C) based on a modern calibration data set of 193 lakes and 162 taxa from
northern Russia (spanning from 61 to 75°N, and 50 to 140 °E, $T_{July}$ range 1.8 to 18.8 °C; Nazarova et al., 2015).
Water depths (WD) were reconstructed using a modern chironomid-based calibration dataset from East Siberia
that includes 147 lakes (WD range 0.1 to 17.1 m). The one-component WA-PLS model had the best performance:
$r^2$ boot = 0.62, RMSEP boot = 0.35 m for WD reconstructions (Nazarova et al., 2011). Both the $T_{July}$ NR and the
WD East Siberia models were previously applied for paleoclimatic inferences in East Siberia and the Russian Far
East and demonstrated high reliability of the reconstructed parameters (Syrykh et al., 2017; Nazarova et al., 2017a,
2017b; Wetterich et al., 2018). Chironomid-based reconstructions were performed in C2 version 1.7.7 (Juggins,





2007). The data were square-rooted to stabilize species variance. Information on the ecology of chironomid taxa
was taken from Brooks et al. (2007), Moller Pilot (2009), and Nazarova et al. (2008, 2011, 2015, 2017a).
Cladocera were newly studied (Table A1) in profile L7-11 on Bol'shoy Lyakhovsky Island, in profiles Oya-3-11,
Oy7-01, Oy7-08, and in profile Oya 5-1 (Kienast et al., 2011) on Oyogos Yar. The cladocera sample preparation
followed the standard methods (e.g., Kienast et al., 2011).
Mollusk fossil remains were obtained from samples BL-R-M1 (taken 1.8 m a.s.l. at profile R 41+50 m in 1999)
and BL-R-M4 (taken at profile R32 in 1999) on Bol'shoy Lyakhovsky Island (E.E. Taldenkova, T.A. Yanina,
unpublished data (Table A1), and from sample Oya 5-1 on Oyogos Yar (Kienast et al., 2011). The mollusk sample
preparation followed standard methods (e.g., Kienast et al., 2011).
Ostracod valves from Bol'shoy Lyakhovsky Island were studied in profiles R23+40, L11+40 (S. Wetterich,
unpublished, Table A1), L7-14 (Wetterich et al., 2009) and L7-11 (Schneider, 2010), and on Oyogos Yar in profiles
Oy7-08 (Wetterich et al., 2009), Oy7-01 (Schneider, 2010) and Oya 5-1 (Kienast et al., 2011). The ostracod sample
preparation followed standard methods (e.g., Wetterich et al., 2009).
**3.5    Clumped isotope analysis of biogenic carbonates and derivation of lake water $\delta^{18}O$**
An emerging method to derive quantitative paleotemperature estimates from ostracods is clumped isotope
thermometry (Song et al., 2022). The advantage of this method is its independence of the temperature estimate
from the $\delta^{18}O$ signal of the water from which the ostracod and mollusk carbonate formed (Eiler, 2007). Two
ostracod species (*Cytherissa lacustris*, *Candona candida*) and a bivalve mollusk (*Pisidium casertanum*) were
selected from sample Oya 5-1 for clumped isotope analysis based on their relatively high abundance providing
sufficient sample material. Complete adult valves, with no visual evidence of dissolution or degradation, were
selected and cleaned. We manually removed sediment contamination under a binocular microscope with a
paintbrush and deionized water prior to the homogenization of the carbonate with a clean agate pestle and mortar.
Clumped isotope analysis was conducted in the NICEST laboratory at Northumbria University on a Nu Instruments
Perspective IRMS coupled with a NuCarb dual inlet prep system. Powdered samples of $325 \pm 25$ µg were loaded
into sample vials, evacuated, and reacted with concentrated orthophosphoric acid at 70 °C. Analyte gas was
dehydrated and cleaned following established methodologies (e.g., Bernasconi et al., 2018; Eiler and Schauble,
2004; Petersen et al., 2015). Briefly, $CO_2$ was dehydrated at –80 °C in two liquid nitrogen-cooled coldfingers and
scrubbed of contaminants by passing through a static 1 cm cryotrap filled with Porapak™ Q absorbent (Waters
Corporation) cooled to –30 °C. The sample preparation system was baked out at 80 °C after each measurement to
avoid cross-contamination. A minimum of 17 replicate measurements was made of each sample, sufficient to
achieve standard errors ≤ 0.01 ‰ (i.e., 95% confidence interval < 0.02 ‰). Long-term instrument performance
was monitored with an internal standard, POL-2, giving a long-term $\Delta_{47}$ external standard deviation of 0.032 ‰.
Isotopic outliers (stable and clumped) and samples with elevated $\Delta_{48}$ values, indicative of sample contamination,
were discarded before final $\Delta_{47}$ values were calculated in the free software Easotope (www.easotope.org; John and
Bowen, 2016) using the IUPAC parameters for $^{17}O$ correction and calculation of isotopic ratios for VPDB and
VSMOW (Bernasconi et al., 2018; Brand et al., 2010; Daëron et al., 2016). Internal $\Delta_{47}$ values were projected onto
the carbon dioxide equilibrium space (ICDES-90) using standards ETH1, ETH2, and ETH3 (ETH Zurich,
Bernasconi et al., 2018), following the methods of Dennis et al. (2011), using ICDES $\Delta_{47}$ values (Bernasconi et
al., 2021). Clumped isotope-based carbonate precipitation temperatures ($T\Delta_{47}$) were calculated using the




composite calibration of Anderson et al. (2021), which has recently been shown to produce reliable temperature
estimates for ostracods (Marchegiano et al., 2024).
We determined the δ18O of the water in which ostracod carbonates formed using TΔ47 values and the δ18O values
of fossil ostracod and bivalve carbonate (measured during clumped isotope analysis). A constant δ18O vital offset
+2.2 ‰ to *Candona candida*, +1.2 ‰ to *Cytherissa lacustris*, and +0.86 ‰ to *Pisidium casertanum* (von
Grafenstein et al., 1999) was applied before δ18O of the formation waters were calculated using the calibration of
Coplen (2007) to describe the temperature-dependent water-calcite oxygen isotope fractionation.

## 3.6      Paleoclimate modeling

Within the framework of the 6th phase of the IPCC Climate Model Intercomparison Project (CMIP6), the
PaleoMIP intercomparison project was endorsed, leading to the identification of several past time periods for
paleo-climate simulations to focus on, following a standardized modeling protocol (Kageyama et al., 2018). One
of these time periods is the LIG, with a center on the time slice around 127,000 years ago, which lead to modeling
experiments summarized under the acronym lig127k. Monthly mean air temperature and precipitation are available
from 12 different global coupled climate models in the ligk127 experiment, providing between 100 and 700 years
of data for the LIG for the calculation of long-term averages (climatological means). In order to provide a reference
for these simulations, all models also provided simulations of the pre-industrial period (PIcontrol). Commonly,
model results are presented as anomalies to this reference period to reduce the impact of systematic model biases
in the results (delta change method, e.g., Maraun and Widmann, 2018). The horizontal resolution of the models
varies between 100 km and 500 km (Table S8). Monthly mean temperatures of MTWA and MTCO were calculated
from each model on the model's native grid, first finding the maximum and minimum monthly values for each
year and then averaging over all years the model provided, both for the lig127k simulations and the pre-industrial
simulations. To obtain the anomalies, the resulting values were subtracted from each other (lig127k minus
PIcontrol). Mean annual precipitation (MAP) was calculated on each model's native grid, summing up monthly
precipitation values. Then, climatological means were calculated, averaging the years the model provided for both
the lig127 and the PIcontrol simulations. Anomalies were again obtained by subtracting both averages (lig127k
minus PIcontrol). Estimates and uncertainties for the sample sites Bol'shoy Lyakhovsky and Oyogos Yar, as well
as a generic reference point, were computed from the multi-model ensemble mean anomalies and of the grid cell
values the sites fall in, without spatial interpolation. To visualize temperature and precipitation patterns of the
Laptev Sea area, temporal averages of all models were regridded to a common 1° x 1° grid.
In order to compare the modeled anomalies with the absolute values determined from the proxies, it is necessary
to account for the discrepancy between the pre-industrial reference used for the model and the observed pre-
industrial time values. For temperature, we used the gridded temperature anomalies from NOAA GlobTemp V6,
the most comprehensive gridded observational data record covering both present-day and pre-industrial climate
(Huang et al., 2024), to obtain anomalies for the observed pre-industrial time (1850-1900, Allen at al., 2018). For
the present-day reference time period in NOAA GlobTemp, the latest reanalysis product of the European Center
for Medium-Range Weather Forecast, ERA5, was used as a basis to calculate the absolute values (methods in
accordance with Allen et al. (2018) and Capron et al., (2014)). For pre-industrial precipitation, no equivalent data
set to NOAA GlobTemp exists. Therefore, it was not possible to obtain absolute values for MAP from the
PaleoMIP model anomalies. Instead, we use the multi-model ensemble mean. All models contributing to this

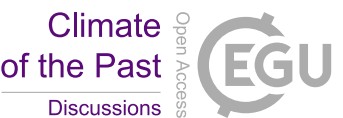

experiment use a present-day land-sea mask and sea level (Figure S3) that does not conform to the suggested land-
sea mask of the area during the LIG. The warmest temperatures and mean annual precipitation are influenced by
a site's proximity to the coast. In the models, the horizontal resolution influences the position of the coastline and
its distance to the sample sites. The model grid layout also influences the partitioning between the water and land
of the grid cells in which the sample sites are located. To estimate the impact of the differences in coastlines among
the models as well as between the models and the conditions of the LIG, an additional reference point was chosen
for computation of MTWA, MTCO, and MAP that is situated within a land grid cell in every model and as close
to the sample sites as possible, referred to as LandPoint, situated at 71.2°N, 142°E (Figure S3).
As a reference for the present day, monthly mean air temperatures and precipitation ERA5 were used to calculate
MTWA, MTCO, and MAP. Values were calculated on the ERA5's native equal-area grid (nominal resolution
about 35 km) and then interpolated onto a regular grid for plotting maps. Climatological means are calculated over
the period 1990-2019 (WMO present-day climate reference period). Values for the sample sites and the reference
point were taken from respective grid cell values without spatial interpolation. Please note furthermore, that we
use the modern calendar instead of an adjusted angular calendar that would account for shifts in solstice and the
related seasons during the lig127 period caused by the high eccentricity of Earth's orbit around the sun. While
studies like  Shi et al. (2022) and Xu et al. (2024) demonstrate the profound impacts of using the classical calendar
in particular in autumn, we expect the impacts on MTWA and MTCO calculations to be negligible because shifts
in the position of a single month are small, especially winter and summer. We do not predefine which month of
the year is considered warmest or coldest; rather, we determine them for each year individually.

## 4     Results

### 4.1     Field observations

Ice-wedge pseudomorphs of 1-3 m thickness of the Krest-Yuryakh stratum are exposed between 0.5-10 m a.s.l. in
places along the southern coast of Bol'shoy Lyakhovsky Island (Figure 2a, Figure 3a). Such ice-wedge
pseudomorphs are filled with alternating beds of peaty brownish plant detritus layers, partly with twig and wood
fragments up to 5-8 cm in diameter and gray clayish silt layers. The thickness of individual layers varies from a
few millimeters to 1-2 cm. Ripple bedding (ripples 1-2 cm high, 2-5 cm distance), finely laminated layers (each
lamina 5-10 mm thick), and small-scale syn-sedimentary slumping structures are common (Figure 3b). Several
layers contain 5-10 mm large mussel fragments. Larger twig fragments and peat inclusions of 2-3 cm are present.
The cryostructure is dominantly massive, i.e., without visible ice structures. Only single thin ice veins (< 1 mm
thick) are observed in places parallel to the sedimentary bedding. Besides ice-wedge pseudomorphs, there are also
lacustrine deposits with horizontally alternating layers of well-laminated silty sand and peat.





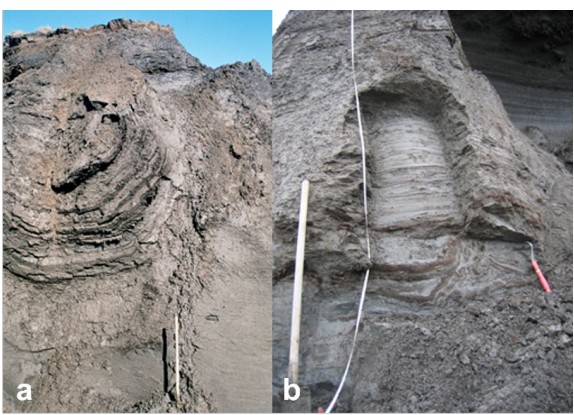

Figure 3. LIG (Krest-Yuryakh) deposits exposed at the southern coast of Bol'shoy Lyakhovsky Island (Figure 2a): (a) ice-wedge pseudomorph (profile L7-11) with well-bedded lacustrine deposits and (b) laminated lacustrine deposits (profile L7-14 A-C) with alternate bedding of peaty and silty sand layers, and a slumping structure (next to the hoe).

Ice wedge pseudomorphs like those on Bol'shoy Lyakhovsky Island have been studied at various locations along the Oyogos Yar coast (Figure 4a). In addition, exposure of lacustrine gray silty fine sand (partly stratified) deposits was studied over a length of about 110 m at 1–3 m a.s.l. with two-meter-thick deposits (profile Oy7-01; Figure 4b). About 50 m toward the east, the profile is characterized by alternating beds of grayish-brown silty material and dark plant detritus, and mollusk shells.

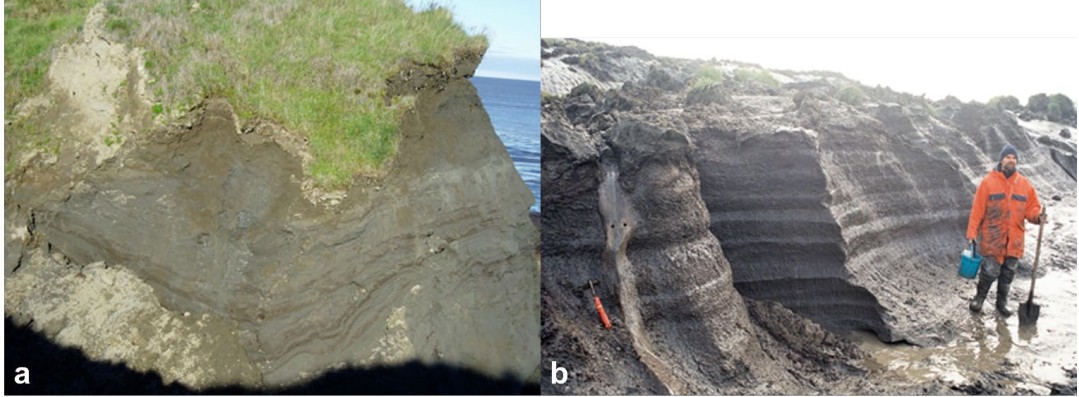

Figure 4. LIG (Krest-Yuryakh) deposits at the Oyogos Yar coast (Figure 2b): (a) ice-wedge pseudomorph of alternately bedded peat and silty sand layers (ca. one meter high from the bottom to the grass cover); (b) stratified lacustrine deposits (profile Oy7-01) above the beach.

## 4.2    Luminescence dating

After preparation, two samples from stratified lacustrine Krest-Yuryakh deposits (profile L14-12) revealed sufficient material for IRSL dating of one sub-sample in the coarse silt fraction (40-63 µm) and two sub-samples in the fine sand fraction (63-90 µm). The obtained ages range from 127.3±6.1 ka to 117.6±6.0 ka (Table 1). The



three ages agree within errors, nevertheless the luminescence signal dispersion corresponding to the oldest age
shows a slightly higher skewness and hence, indicates a slight overestimation.
**Table 1: IRSL sample characteristics, including paleodose and dose rate parameters.**

| Sample ID (Grain size fraction) | | L14-12-OSL3 (63-90 µm) | L14-12-OSL1 (63-90 µm) | L14-12-OSL1 (40-63 µm) |
|---|---|---|---|---|
| Paleodose parameters | | | | |
| n | | 25 | 20 | 20 |
| Mean | [Gy] | 295.1±1.2 | 287.5±2.7 | 271.9±0.6 |
| Standard deviation | [%] | 5.8 | 4,2 | 2,8 |
| Skewness | | 0.2 | 0.8 | 0.4 |
| Coefficient of variation | [%] | 5.1 | 3.8 | 5.3 |
| CAM | [Gy] | 295.2±3.5 | 287.2±3.14 | 271.8±2.9 |
| Dose rate parameters | | | | |
| $^{238}$U-series | [Bq kg$^{-1}$] | 28.11±0.90 | 20.96±0.72 | 20.96±0.72 |
| $^{232}$Th-series | [Bq kg$^{-1}$] | 31.14±1.40 | 24.22±1.07 | 24.22±1.07 |
| $^{40}$K | [Bq kg$^{-1}$] | 475.32±1.82 | 481.98±1.66 | 481.98±1.66 |
| Water content | [%] | 35±5 | 35±5 | 35±5 |
| Height | [m a.s.l.] | 2.7 | 4.5 | 4.5 |
| Cover thickness | [m] | 30 | 28 | 28 |
| Total dose rate | [Gy ka$^{-1}$] | 2.6±0.1 | 2.4±0.1 | 2.4±0.1 |
| **Age** | **[ka]** | **117.6±6.0** | **127.3±6.1** | **117.8±6.8** |

**4.3    Sedimentology and biogeochemistry**
Krest-Yuryakh deposits on Bol'shoy Lyakhovsky Island are characterized by low ice contents (20-33 wt%) and no
wedge ice at all. The magnetic susceptibility (MS) values vary between 22 and 55 10$^{-8}$ m$^3$kg$^{-1}$, differing from
outcrop to outcrop, while within single profiles, the differences are much smaller (Figure 5a). The TOC contents
range widely from 0.6 to 15.3 wt%. High TOC values (> 5 wt%) are related to plant detritus, peat, and woody
remains. The TOC/TN ratio reflecting the degree of organic matter decomposition is between 1.8 and 26.2. Low
TOC/TN ratios indicate high decomposition and vice versa (Carter and Gregorich, 2008; White, 2005). The δ$^{13}$C
values range from –30.1 ‰ to –25.8 ‰. Differences of 3-4‰ occur within individual profiles. The carbonate
contents derived from the TIC values range from 0.2 to 6.7 wt% (Figure 5a). High values are linked to the higher
presence of mollusk remains. The mean arithmetic grain size ranges between 15 and 78 µm, and the grain-size
distribution curves are characterized by a three-modal shape, with peaks in the fine silt, fine sand, and medium
sand fractions (Figure 5b).



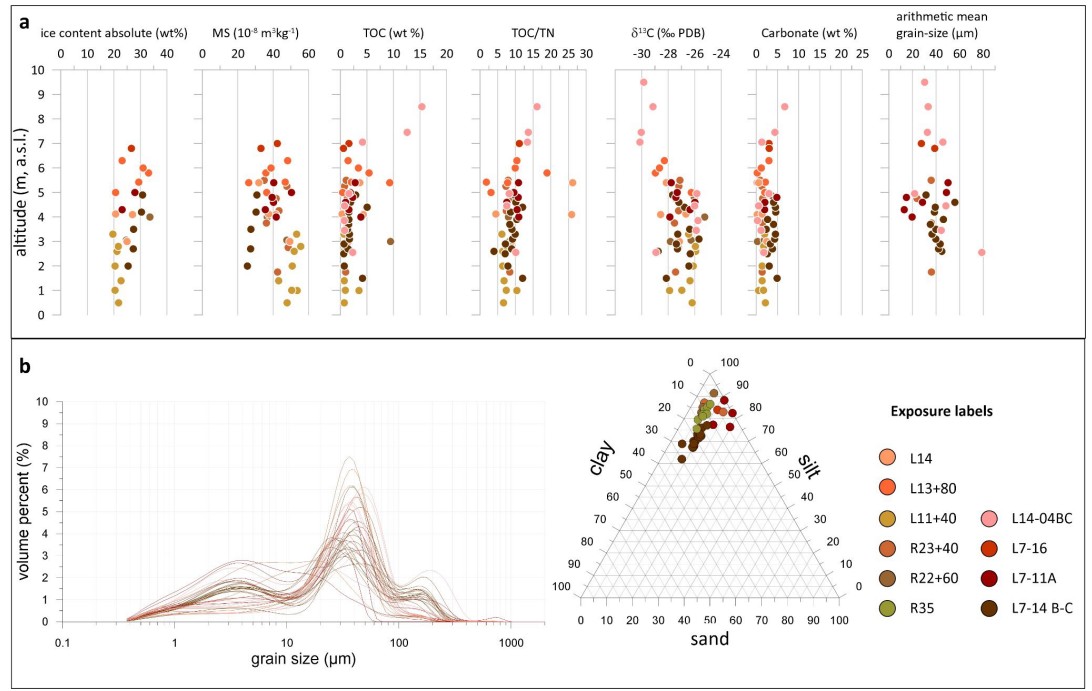

**Figure 5. Sediment data of LIG (Krest-Yuryakh) profiles on Bol'shoy Lyakhovsky Island: (a) absolute gravimetric ice content, mass-specific magnetic susceptibility (MS), TOC content, TOC/TN ratio, $\delta^{13}$C, carbonate content, arithmetic grain-size mean; (b) grain-size distribution curves and sand-silt-clay percentages.**

The cryolithological characteristics of Krest-Yuryakh deposits from the Oyogos Yar coast are similar to those of Bol'shoy Lyakhovsky Island (Figure 6a). The absolute ice contents range from 18 to 34 wt%. The MS varies between 12 and 45 $10^{-8}$ m$^3$kg$^{-1}$. Within individual profiles, the differences are rather low, but there are stronger differences in MS values between the profiles. The TOC contents range from 0.6 to 21.3 wt%. The TOC/TN ratio is between 3.5 and 18.7. The $\delta^{13}$C values range from –30.8 ‰ to –26 ‰. The carbonate contents range from 1.4 to 22.4 wt%. The mean arithmetic grain sizes are between 15 and 59 μm, and the grain-size distribution curves are three-modal (Figure 6b).




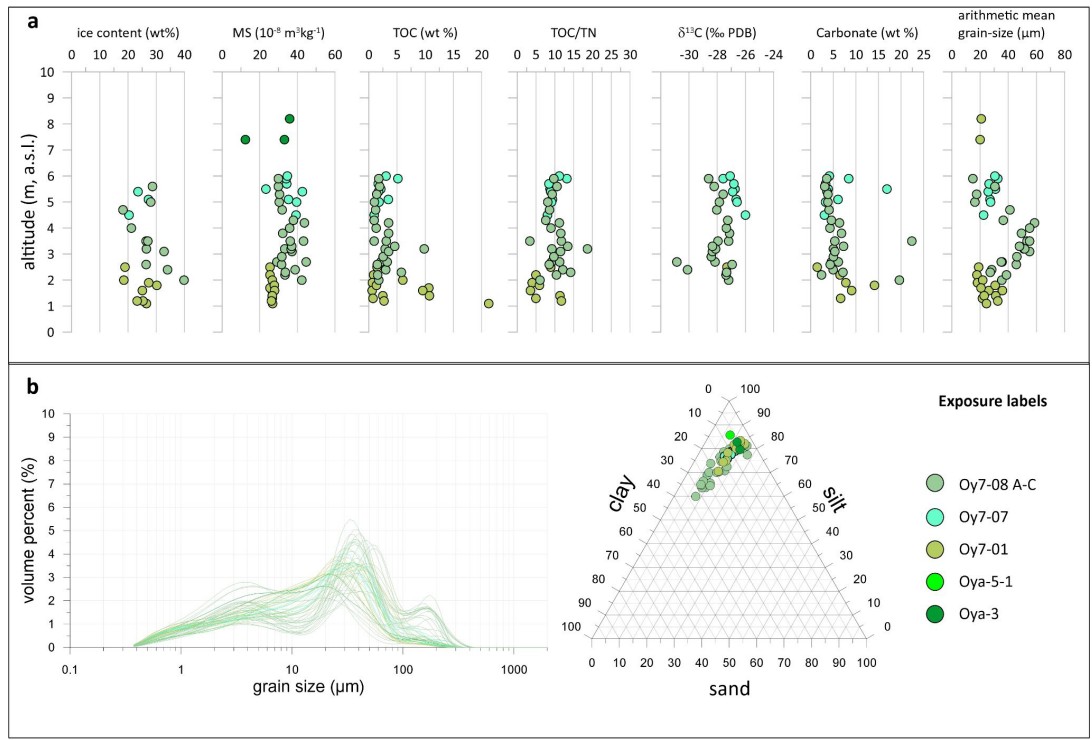

**Figure 6.** Sediment data of LIG (Krest-Yuryakh) profiles on Oyogos Yar: (a) absolute gravimetric ice content, mass-specific magnetic susceptibility (MS), TOC content, TOC/TN ratio, δ¹³C, carbonate content, arithmetic grain-size mean; (b) grain-size distribution curves and sand-silt-clay percentages.

### 4.4 Pollen-spores-based vegetation and paleoclimate reconstructions

The pollen assemblages of the bottom sections of ice-wedge pseudomorphs of Krest-Yuryakh deposits on Bol'shoy Lyakhovsky Island indicate that open steppe or tundra-steppe habitats with Poaceae and *Artemisia* dominated the vegetation at the beginning of the LIG (Andreev et al., 2004; Ilyashuk et al., 2006). However, relatively high abundances of *Alnus fruticosa*, *Salix*, and *Betula* sect. *nana* pollen indicate shrub presence in more protected places such as thermokarst basins and river valleys. The large amounts of coprophilous Sordariaceae fungal spores indirectly point to the presence of grazing herds of the late Pleistocene mammoth fauna (Andreev et al., 2011). The early to middle LIG pollen spectra are dominated by Poaceae, Cyperaceae, *Betula*, and *Alnus* (Andreev et al., 2004; Ilyashuk et al., 2006; Wetterich et al., 2009), reflecting a shrub-tundra vegetation. Relatively high concentrations of herb pollen taxa (*Artemisia,* Brassicaceae, Caryophyllaceae, Asteraceae) indicate that open habitats were also common. Relatively high amounts of *Glomus* fungal spores suggest that the local vegetation was frequently disturbed, probably due to active erosion processes connected with melting ice wedges and the formation of thermokarst lakes.

Pollen assemblages of the middle LIG from the Oyogos Yar coastal sections are dominated by Poaceae, Cyperaceae, *Larix*, *Alnus fruticosa*, and *Betula* sect. *nana* and spores of *Equisetum*, dung-inhabiting Sordariaceae, and *Glomus* (Wetterich et al., 2009; Andreev et al., 2011). Based on the relatively high percentage of *Larix* pollen, we may infer that larch forest or forest-tundra with shrub alder and dwarf birch stands dominated the vegetation



in the Oyogos Yar area and document that the treeline was at least 270 km north of its current position during the
LIG optimum.
Based on seven Krest Yuryakh profiles with 69 samples from Bol'shoy Lyakhovsky and one profile with 25
samples from Oyogos Yar, the reconstructed mean MTWA was 9.0±3.0 °C for Bol'shoy Lyakhovsky (median
9.7±3.1 °C) and 9.7±2.9 °C at Oyogos Yar (median 9.6±2.8 °C). The MAP was 271±56 mm (median 264±52 mm)
on Bol'shoy Lyakhovsky and 229±22 mm (median 230±23 mm) at Oyogos Yar, implying that the summer climate
conditions on Bol'shoy Lyakhovsky were a bit colder and moister than on Oyogos Yar.
**4.5    Plant macrofossil-based vegetation and paleoclimate reconstructions**
Plant macrofossil assemblages from Krest-Yuryakh deposits at both sides of the Dmitry Laptev Strait are
exceptionally well preserved and frequently allowed for identification at the species level, giving a detailed picture
of local vegetation and habitat conditions at the time of deposition (Table S2). On Bol'shoy Lyakhovsky Island,
assemblages are composed of remains of 100 vascular plant taxa, including aquatic macrophytes living in the
thermokarst lakes (Table A1). The detected species reflect a wide range of plant communities comprising open
subarctic shrub tundra with the tall shrubs *Alnus alnobetula* ssp. *fruticosa* and *Betula fruticosa,* as well as dwarf
shrubs like *Betula nana* s.l., *Vaccinium vitis-idaea*, *Rhododendron tomentosum*, and *Empetrum nigrum*
interspersed with patches of dry grasslands with *Carex* (formerly *Kobresia*) *myosuroides*, *Potentilla* spp.,
*Artemisia* sp., *Androsace septentrionalis,* several steppe sedge species, and *Ranunculus pedatifidus* ssp. *affinis*
suggesting the existence of arid habitats during MIS 5e. Arid conditions are confirmed by the halophytes
(*Puccinellia, Tripleurospermum hookeri, Stellaria crassifolia, Rumex maritimus*) characteristic of salt marsh and
saline meadow vegetation near lake shores with fluctuating water levels. The salt accumulation in the upper soil
layer results from the capillary rise of solutes due to high evaporation under arid continental climates. The lakes
formed habitats of aquatic macrophytes like *Callitriche hermaphroditica, Stuckenia vaginata, Myriophyllum*
*spicatum*, *Batrachium* sp., *Nitella* sp., *Hippuris vulgaris, Sparganium hyperboreum* and *S. minimum*, species
absent in the study area today. Constantly wet habitats were occupied by littoral and tundra wetland vegetation
with *Eriophorum* spp., *Carex aquatilis*, *C*. sect. *Phacocystis*, *Juncus biglumis*, *Chrysosplenium alternifolium*,
*Comarum palustre, Caltha palustris, Parnassia palustris*, *Ranunculus hyperboreus*, *R. lapponicus* and
*Gastrolychnis violascens*. The estimated LIG MTWA on Bol'shoy Lyakhovsky Island is based on the MCR of 15
selected taxa (Table 2) and ranges from 10.3 °C to 12.9 °C.
The plant remains at the mainland coast of Oyogos Yar reflect a LIG vegetation similar to Bol'shoy Lyakhovsky
Island consisting of open shrub tundra and forest tundra interspersed with patches of steppe and meadow grassland.
In contrast to modern larch-dominated forest tundra, the LIG woodland was dominated by birches, as shown by
the abundance of birch remains from both trees and shrubs. In addition to tall and dwarf shrubs already recovered
from Bol'shoy Lyakhovsky Island, extralimital Ericaceae taxa (*Arctostaphylos uva-ursi, Andromeda polifolia,*
*Chamaedaphne calyculata*) and forbs (*Moehringia laterifolia, Stellaria longifolia, Chamaenerion angustifolium*)
occurred at Oyogos Yar. Characteristic of the undergrowth of modern boreal forests, they likewise indicate a
relatively long and warm growing season during MIS 5e.
Analogous to Bol'shoy Lyakhovsky Island, the reconstructed wooded tundra at Oyogos Yar was rather open as
suggested by abundant remains of tundra-steppe plants like *Carex myosuroides, Dryas octopetala* s.l.,
*Rhododendron* sp., *Potentilla stipularis, P. nivea, Ranunculus pedatifidus* ssp. *affinis* and meadow steppe species,



such as *Odontarrhena obovata, Allium* sp.*, Artemisia* sp., *Carex duriuscula, C. supina* s.l., *Eritrichium sericeum,*
*Rumex acetosella* s.l.. These meadow steppes merged into productive alkali grass meadows indicated by abundant
remains of *Puccinellia. Puccinellia* sp. and other halophilic taxa, like *Chenopodium* sp. and *Spergularia salina,*
have an affinity to brackish conditions, which suggests high evaporation and low lake levels in response to seasonal
aridity at Oyogos Yar during the LIG, similar to Bol'shoy Lyakhovsky Island.
The inventory of water plants at Oyogos Yar resembles that of the Bol'shoy Lyakhovsky assemblage and was
supplemented by abundant extralimital, i.e., thermophilous, aquatic macrophytes like *Stuckenia filiformis* and
*Potamogeton perfoliatus*. The estimated LIG MTWA at Oyogos Yar is based on the MCR of 17 selected taxa
(Table 2) and ranges from 12.7 °C to 15.3 °C.
**Table 2: MTWA requirements and updated coexistence interval of selected vascular plant species that were identified**
**in LIG (Krest-Yuryakh) deposits on Bol'shoy Lyakhovsky Island and the Oyogos Yar mainland coast. Determining**
**values of the coexistence intervals are highlighted in bold.**

| Taxon | MTWA$_{Min}$ [°C] | MTWA$_{Max}$ [°C] | Bol'shoy Lyakhovsky Island | Oyogos Yar |
|---|---|---|---|---|
| *Alnus hirsuta* | **12.7** | 18.4 | | X |
| *Alnus alnobetula* subsp. *fruticosa* | 7.5 | 18.4 | X | X |
| *Betula nana* s.l. | 5.2 | 18.0 | X | X |
| *Betula fruticosa* | 9.1 | 18.4 | X | X |
| *Potamogeton perfoliatus* | 10.3 | 18.4 | | X |
| *Myriophyllum spicatum / M. sibiricum* | **10.3** | 18.4 | X | X |
| *Larix gmelinii* | 8.0 | 18.4 | | X |
| *Betula divaricata* | 7.6 | 18.4 | | X |
| *Arctostaphylos uva-ursi* | 10.6 | 18.4 | | X |
| *Moehringia lateriflora* | 9.9 | 18.4 | X | X |
| *Callitriche hermaphroditica* | 8.1 | 18.4 | X | X |
| *Menyanthes trifoliata* | 5.2 | 18.1 | X | |
| *Sparganium hyperboreum* | 8.1 | 18.1 | X | X |
| *Sparganium minimum* | 10.3 | 18 | X | X |
| *Cherleria arctica* | 3.3 | 13.3 | X | |
| *Coptidium lapponicum* | 3.5 | 16.9 | X | X |
| *Silene involucrata* | 3.3 | 15.7 | X | X |
| *Ranunculus pedatifidus* ssp. *affinis* | 5.6 | 17.6 | X | X |
| *Sagina nivalis* | 2.6 | **12.9** | X | |
| *Potentilla hyparctica* | 2.6 | 14.6 | X | |
| *Ranunculus nivalis* | 1.6 | **15.3** | | X |
| **Coexistence interval MTWA [°C]** | | | **10.3 to 12.9** | **12.7 to 15.3** |

**4.6    *Sed*aDNA-based vegetation reconstruction**
The Krest-Yuryakh deposits on Bol'shoy Lyakhovsky Island exhibit rich vegetation with *sed*aDNA derived from
several woody taxa, including trees, shrubs, and sub-shrubs, suggesting interglacial, warmer-than-present
conditions (Figure 7). The proportions of sequences assigned to woody taxa (trees < 1% and shrubs 6-87%), forbs
(8-91%), and grasses (2-39%) were highest, while the proportions of sedges (< 1%), cryptogams (< 1%), and



aquatic forbs (up to 1.6%) were very low. Among woody taxa, we detected *Larix, Picea, Populus, Alnus, Betula,*
*Ribes, Saliceae,* and *Cornus,* as well as sub-shrubs such as arctic-alpine *Dryas* and Ericaceae, including *Arctous,*
*Pyrola* spp., *Vaccinium uliginosum,* and *Vaccinium vitis-idaea* (Zimmermann et al., 2017) which are typical
components of the understory in boreal forests but also of subarctic tundra habitats. The forbs contain mainly taxa
adapted to dry steppe conditions (*Artemisia gmelinii,* including halophilic *Puccinellia*), arctic-alpine tundra
(*Braya, Draba,* and *Dryas*), or pioneer plants (*Papaver* and *Oxyria digyna*). Other forbs such as *Geum, Myosotis*
*alpestris,* and *Bistorta* are typical components of forest margins or meadows. The Krest-Yuryakh deposits formed
in a shallow lake in which aquatic and riparian forbs included *Menyanthes trifoliata, Stuckenia, Potamogenton,,*
*Hippuris,* and *Caltha palustris.* Nevertheless, sedges in the lower part of the core L14-04 were composed of two
distinct *Kobresia* variants (same barcode shared between *K. filifolia* and *K. simpliciuscula* and between *K. sibirica*
and *K.myosuroides* (now *Carex myosuroides*) typical of dry to wet habitats while the upper section of the profile
also contained the sedges *Carex* and *Eriophorum* that are typical for wet habitats but also steppe. This shift between
8 and 12 m a.s.l. is accompanied by high proportions (up to 33%) of cryptogams and graminoids (*Eriophorum,*
Bryophytes), while woody taxa were only represented by *Salix,* overall suggesting a transition to cooler and more
moist conditions.

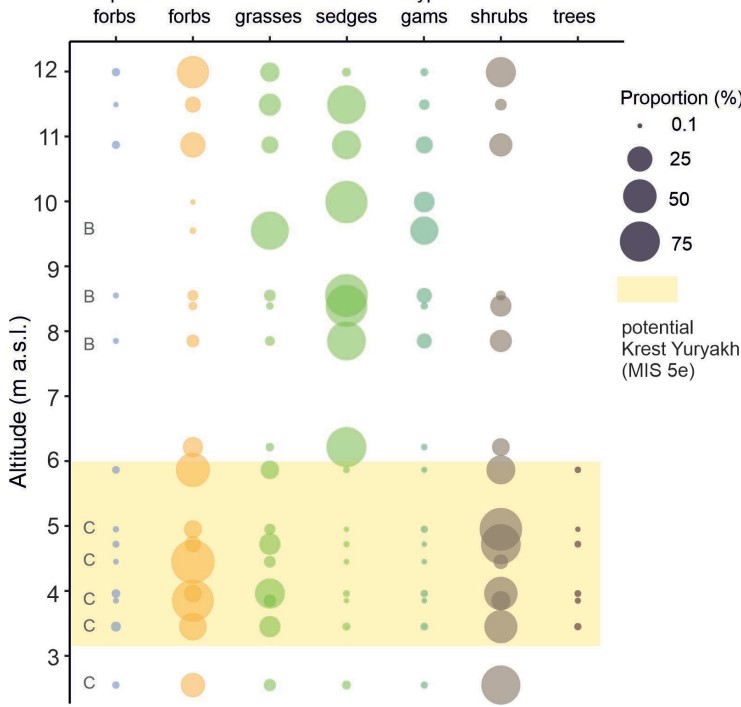

**Figure 7. Composition of plant functional groups in the *sed*aDNA metabarcoding record of combined LIG (Krest-**
**Yuryakh) samples from core L14-04 and profiles L14-04-B (indicated by B) and L14-04-C (indicated by C) from**
**Bol'shoy Lyakhovsky Island. Figure 7 was generated using ggplot2 v. 3.4.2 (Wickham, 2016) in R v. 4.1.3 (R Core Team,**
**2022).**
**4.7      Biomarker-based paleoclimate reconstruction**



Samples from core L14-04 analyzed for GDGTs (Table S3) include those from the MIS 5e (Krest-Yuryakh stratum
(n = 3) as well as deposits from younger MIS 5d-a horizons (n = 6) and MIS 1 (active layer; n = 1). BrGDGT
distributions globally have a near-universal relationship with temperature irrespective of sample type (Raberg et
al., 2022), yet their producers are ubiquitous in nature. Since the Krest-Yuryakh stratum is composed of both
lacustrine deposits and peaty plant detritus layers, we tested the potential influence of in-situ production by benthic
bacteria in sediments using #rings$_{tetra}$ (Sinninghe Damsté, 2016). The #rings$_{tetra}$ values throughout core L14-04
range from 0.04 to 0.25, with values from 0.12 to 0.15 in the LIG deposits. These values indicate that brGDGTs
are not produced by benthic bacteria (#rings$_{tetra}$ >0.7), but instead mostly derive from soil/peat and, thus, record
Air Growing Season Temperature (GST). Reconstructed Air GST is 0.9 °C for the active layer, 2.8±0.3 °C for the
MIS 5e deposits and 1.3±0.9 °C for the MIS 5d-a deposits (Figure 8).
Although brGDGT distributional changes have been observed as a direct physiological response to temperature
(as well as pH and O$_2$ concentrations) in acidobacterial cultures (Halamka et al., 2021, 2023), various confounding
factors such as soil chemical properties (pH, cation availability) and bacterial community composition (e.g.,
Halffman et al., 2021; de Jonge et al., 2024) can also affect brGDGT distributions. For example, the BIT and Ri/b
indices in soils show a relationship with mean annual precipitation/soil moisture. At lower soil moisture, BIT index
decreases and the Ri/b increases (Xie et al., 2012; de Jonge et al., 2024). For core L14-04, Ri/b values are
substantially higher in the Krest-Yuryakh stratum (Figure 8), suggesting more arid conditions during MIS 5e
compared to MIS 5d-a and MIS1. The (isoGDGT-1 + isoGDGT-3)/(isoGDGT-1 + crenarchaeol) values, which
seem to correlate with mean monthly precipitation (de Jonge et al., 2024) also follow this overall trend. Absolute
crenarchaeol concentrations do not correlate with Air GST (Pearson correlation coefficient r = 0.058, p = 0.87),
suggesting the observed trends are indeed controlled by soil moisture/precipitation rather than temperature. Yet,
no calibration exists that would allow us to calculate mean annual precipitation, thus we only use this information
qualitatively.

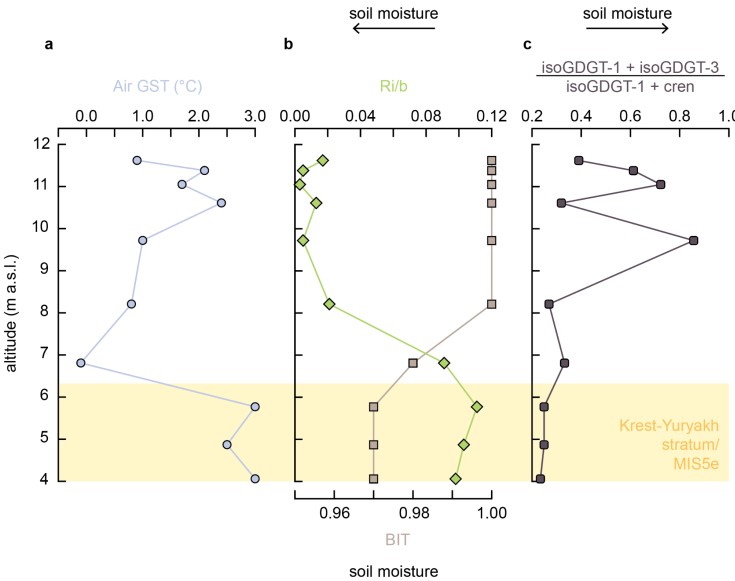

**Figure 8. BrGDGT-based proxies in core L14-04: (a) MBT'$_{5ME}$-based Air Growing Season Temperature (Air GST)**
**following de Jonge et al. (2024); (b) branched and isoprenoid tetraether index (BIT) and ratio of isoGDGTs to brGDGTs**
**(Ri/b); (c) ratio (isoGDGT-1 + isoGDGT-3)/(isoGDGT-1 + crenarchaeol).**



**4.8    Beetle-based faunal habitat and paleoclimate reconstructions**
The fossil insect assemblages of the LIG on Bol'shoy Lyakhovsky Island and on Oyogos Yar are rich in species,
and the concentration of remains is very high in comparison to other stratigraphic units in the study area (Kuzmina,
2015a, 2015b, Table S4). Insect remains are well preserved. The fossil insect fauna shows a high amount of steppe
species (Table S5). The share of *Morychus viridis* reaches up to 15%. Several identified thermophilous steppe
species (*Cymindis arctica, Chrysolina brunnicornis bermani, Stephanocleonus eruditus, S. fossulatus*) are absent
in other Pleistocene samples. Arctic species (*Chrysolina subsulcata, Ch. bungei*) present only 3 %. The weevil
*Dorytomus imbecillus* indicates shrub vegetation. Several species prefer habitats in and on plant litter
(*Cyrtodactylus irregularis, Eucnecosum tenue, Lathrobium* sp., *Philonthus* sp., *Quedius* sp.). A number of riparian
and aquatic insects (*Colymbetes dolabratus, Aegialia kamtschatica, Agonum impressum, Sericoda quadripunctata,*
*Scymnus* sp., *Notaris bimaculatus*) identified in the Interglacial samples are not recorded on the island today. The
predaceous diving beetle *Colymbetes dolabratus* lives in the north of boreal forest and tundra zones up to Baffin
Island and Greenland. In Eurasia, the species is common in the north but is not found in the high Arctic. Other
species are nowadays distributed mostly in the forest zone, but their life cycle is not connected directly to the trees.
The ground beetle *Sericoda quadripunctata* is known as post forest-fire species but can also occur in any open
disturbed habitats. A brief comparison of representative insects from older and younger stratigraphic horizons are
summarized in a separate text as Text S1 and a possible periglacial landscape is presented in Figure S2.
To evaluate the LIG climate conditions in the Dmitry Laptev Strait region, two sources of thermal requirements
are used, which are a West Beringian list (including phytophagous species; Alfimov et al., 2003) and a
Transberingian list (excluding phytophagous species; Elias, 2000), both based on museum collections. In these
datasets, several species have slightly different temperature ranges in East and West Beringia (Table S6). The
overlap in coexistence intervals for West Beringian species is shown in Figure S1. Combined results are presented
in Table 3. Including phytophagous beetles provides important environmental information. The weevils
*Stephanocleonus eruditus* and *S. fossulatus* need high soil temperature (>12 °C) for the larvae to grow. Larvae of
these weevils are root eaters and live in the soil horizon. They are active in warm seasons only. Winter temperature
is not critical (Berman et al., 2011). The coexistence of thermophilous weevils and cold-adapted leaf beetles
(MTWA range of *Chrysolina subsulcata* is 2 to 10 °C; Table S6) in one fossil assemblage highlights where the
coexistence intervals do not overlap (Alfimov et al., 2003) as also observed in samples R-22-B15 and L-11-B19
(Table 3).
The thermal coexistence intervals of all considered beetle species, i.e., their MCR is 8 to 10.5 °C for MTWA and
−34 to −26 °C for MTCO on Bol'shoy Lyakhovsky, and 8 to 14 °C for MTWA and −38 to −26 °C for MTCO on
Oyogos Yar.
**Table 3: MTWA and MTCO requirements and coexistence intervals of beetles from LIG (Krest-Yuryakh) samples**
**based on modern reference data in Alfimov et al. (2003) and Elias (2000) applying the MCR method. Determining values**
**of the coexistence intervals are highlighted in bold.**

| Sample ID | MTWA$_{Min}$ [C°] | MTWA$_{Max}$ [C°] | MTCO$_{Min}$ [C°] | MTCO$_{Max}$ [C°] |
|-----------|------------------|------------------|------------------|------------------|
| **Bol'shoy Lyakhovsky Island** | | | | |
| L-11-B17 | **8** | 13 | −37 | **−26** |
| L-11-B19 | 4 | **10.5** | −38 | −24 |





| | | | |
|---|---|---|---|
| R-22-B15 | 4 | 18 | **−34** | −26 |
| R-22-B16 | 8 | 14 | −35 | −26 |
| **Oyogos Yar** | | | |
| Oya 5-1 | **8** | **14** | −38 | −26 |

## 4.9    Chironomid-based habitat and paleoclimate reconstructions
In profile R35 (Bol`shoy Lyakhovsky Island), 33 chironomid taxa were identified. The assemblage from the
lowermost sample (1.2 m a.s.l.) includes a relatively high share of semi-terrestrial taxa *Metriocnemus-*
*Thienemannia, Smittia*, and *Limnophyes-Paralimnophyes* indicative of low and variable water level. Between 1.4
and 2.2 m a.s.l., we find a high relative abundance of the taxa typical of warm and more eutrophic conditions
(*Chironomus plumosus*-type, *Cricotopus-Orthocladius*, *Procladius*). Here, taxa characteristic for shallow water,
semi-terrestrial conditions, or temporary waters (*Limnophyes-Paralimnophyes*, *Georthocladius, Hydrobaenus*)
and taxa that can tolerate acidic conditions (*Tanytarsus, Psectrocladius sordidellus*-type) are less abundant.
Between 2.2 and 5.2 m a.s.l., a high share of eutrophic taxa (*Chironomus plumosus*-type, *C. anthracinus*-type,
*Procladius*) and those indicative of cooler and more acidic conditions (*Sergentia coracina*-type) are present.
The chironomid-inferred MTWA of R35 varies between 3.4 °C (at 1.2 m a.s.l.) and 15.3 °C (at 4.4 m a.s.l.) (Table
4). The median MTWA is 12.7 °C for the middle section (1.6–3.2 m a.s.l) and 13.9 °C for the upper section (3.6–
5.2 m a.s.l.). The highest error of prediction (SE of ± 4.8 °C) occurs in the lowermost sample (at 1.2 m a.s.l.). That
can be explained by the dominance of semi-terrestrial taxa, especially *Metriocnemus,* which is also often found in
lake sediments but has still debated ecological requirements (Moller Pillot, 2009 and reference therein). In the NR
dataset, *Metriocnemus* appears with a broad range of ecological conditions with a high-temperature tolerance of
9.3±4.6 °C (Nazarova et al., 2015), which leads to a high error in the temperature reconstruction. Therefore, these
data from the lowermost sample of profile R35 (at 1.2 m a.s.l.) are not considered in further paleoclimatic
interpretation. For all other samples, the errors of prediction remain at the average level of the transfer function
(1.4–1.5 °C; Nazarova et al., 2015). The inferred WD reflects a period of shallow water (WD of 1.7–2.4 m) during
deposition of the strata between 1.2 and 2 m a.s.l., rising water level (WD 4.5–5.6 m) between 2.4 and 3.2 m a.s.l.,
and decreasing water level (mean WD of 3.8±0.5 m) between 3.6 and 5.2 m a.s.l. (Table 4).
**Table 4: Mean air temperature of the warmest month of the year (MTWA) and water depth (WD) and the errors of**
**prediction (SE) reconstructed from the chironomid communities of LIG (Krest-Yuryakh) deposits of Bol'shoy**
**Lyakhovsky Island (profile R35; Ilyashuk et al., 2006) and Oyogos Yar (profile Oya 5-1; Kienast et al., 2011). Data in**
**brackets are not considered for paleoclimatic interpretation.**

| Sampling height [m a.s.l.] | MTWA ± SE [°C] | WD ± SE [m] |
|---|---|---|
| **Bol`shoy Lyakhovsky Island, profile R35** | | |
| 5.2 | 13.7 ± 1.4 | 3.3 ± 1.0 |
| 4.8 | 13.2 ± 1.5 | 3.8 ± 1.0 |
| 4.4 | **15.3 ± 1.5** | 4.1 ± 1.1 |
| 4.0 | 13.9 ± 1.5 | 3.9 ± 1.1 |





| 3.6 | 13.9 ± 1.4 | 2.7 ± 0.9 |
| 3.2 | 12.7 ± 1.4 | **5.6 ± 1.0** |
| 2.8 | 13.7 ± 1.5 | 4.5 ± 1.0 |
| 2.4 | **9.4 ± 1.7** | 5.4 ± 1.0 |
| 2.0 | 15.1 ± 1.5 | **1.7 ± 0.9** |
| 1.6 | 10.3 ± 2.0 | 2.4 ± 1.0 |
| (1.2) | (3.4 ± 4.8) | (2.4 ± 1.2) |
| **Oyogos Yar, sample Oya5-1** | | |
| 3.5 | **12.9 ± 0.9** | **2.2 ± 1.1** |

The chironomid assemblage in sample Oya 5-1 is diverse and includes 16 taxa. The semi-terrestrial *Limnophyes*,
*Smittia,* and the acidophilic *Psectrocladius sordidellus*-type dominate it. Phytophilic taxa indicative for temperate
shallow lakes or littoral conditions are less abundant (*Cricotopus laricomalis*-type, *Tanytarsus pallidicornis*-type,
*Endochoronomus albipennis*-type). The inferred MTWA from the chironomid community of the Oyogos Yar
sample (Oya 5-1) is 12.9±0.9 °C and WD 2.2±1.1 m (Table 4).
**4.10    Cladocera-based habitat reconstruction**
The fossil cladocera remains of LIG deposits on Bol'shoy Lyakhovsky Island and Oyogos Yar are exceptionally
well preserved. The overall cladocera record comprises 13 taxa, of which six were identified at the species level,
four to groups or taxa, and three at the genus level (Table S7). The most common species that occur in at least four
of the five profiles are *Chydorus* cf. *sphaericus*, *Bosmina* sp., and *Daphnia pulex* gr. The cladocera communities
are dominated by littoral shallow-water taxa, such as *Ch.* cf. *sphaericus* and *Alona guttata*/*Coronatella rectangula*
representing 79 % of the total number of individuals, while the proportion of planktic taxa (*Bosmina* sp., *D. pulex*
gr.) amounts to 21 %.
Profile L7-11 on Bol'shoy Lyakhovsky shows very low concentrations of 1-2 specimens per gram of dry sediment.
In total, only 11 individuals of *Ch.* cf. *sphaericus*, *Bosmina* sp., and *D. pulex* gr. are found (Table S7). Of those,
*Ch.* cf. *sphaericus* is the most common species that is a widely distributed, eurytopic, phytophilous pioneer species
inhabiting the littoral (Bledzki and Rybak, 2016). This taxon is highly adaptive, resistant to adverse environmental
conditions and low temperatures, and often migrates further north than other cladocera species (e.g. Frolova et al.,
2014; Luoto et al., 2011).
The cladoceran records on Oyogos Yar are more diverse and had much higher concentrations than those on
Bol'shoy Lyakhovsky. Exemplarily, the Oya 5-1 record (Kienast et al., 2011) revealed the most numerous record
comprising >150 specimens per sample, representing a total of nine species, most of which belong to the family
Chydoridae (seven species). The assemblage is dominated by *Ch.* cf. *spahericus* (37%), *A. guttata*/*C. rectangula*
(29%), and *Bosmina* sp. (29 %). Littoral species that inhabit macrophytes or detritus-rich silty lake margins, mainly
*Ch.* cf. *sphaericus* and *A. guttata*/*C. rectangula*, represent two-thirds of the assemblage, while one-third is
planktonic (mainly *Bosmina* sp. and *D. pulex* gr.) (Kienast et al., 2011).



Cladocera remains are also well represented in profile Oy7-08, where the high species richness (9 taxa in sample
Oy7-08-19) and the highest concentration of remains in sediments (39 specimens per gram of dry sediment in
Oy7-08-19) are noted (Table S7). Most remains belong to littoral phytophilous species associated with
macrophytes (*Ch.* cf. *sphaericus, Acroperus harpae, Alonella excisa, Eurycercus* sp., *Sida crystallina*). Besides
typical northern or Arctic species such as *A. harpae, Ch.* cf. *sphaericus*, and *Alona affinis* taxa indicative of higher
water temperatures are observed, such as *Leydigia leidigi* and *S. crystallina* that were not found in modern bottom
sediments of >30 water bodies on the coast of the Laptev Sea (L. Frolova, unpublished data). The cladoceran
assemblages of Oyogos Yar indicate habitats with a well-developed vegetated shallow littoral zone and pelagic
open-water zones.
**4.11 Mollusk-based habitat reconstruction**
LIG deposits on Bol'shoy Lyakhovsky Island contain *Sphaerium corneum* with ten complete valves and fragments,
*Valvata piscinalis* with ten complete shells and fragments, *Lymnaea* cf. *peregra* with one shell, and *Pisidium* sp.
with six valves (E.E. Taldenkova, T.A. Yanina, unpublished data).
At Oyogos Yar, mollusks were identified by A. Kossler. There, two freshwater gastropod taxa of the genera *Radix*
and *Gyraulus* are represented only by a few juvenile shell fragments impeding species identification and deduction
of precise environmental implications (Kienast et al., 2011). The distribution of *Radix* further to the north than
today can be interpreted as an indication of warmer than present climate conditions. Additional identified shell
fragments include *Valvata* cf. *piscinalis* and *Lymnea* cf. *stagnalis* (E.E. Taldenkova, T.A. Yanina, unpublished
data). Furthermore, five bivalve species have been identified (Kienast et al., 2011): *Pisidium casertanum, P.*
*subtruncatum*, *P.* cf. *lilljeborgii*, *P. obtusale* f. *lapponicum*, and *P. stewarti* of which the most frequent (*P.*
*casertanum* and *P. subtruncatum*) are eurytopic and widely distributed, while the rare *P. obtusale* f. *lapponicum*
inhabits typically arctic and subarctic regions. The stenoecious species *P. lilljeborgii* indicates oxygen-rich,
oligotrophic, and stagnant water bodies. *P. stewarti* is only known from the Tibetan Plateau and the Siberian Irtysh
region (Kuiper, 1962, 1968). Additionally, the species *Sphaerium* cf. *corneum* has been found (E.E. Taldenkova,
T.A. Yanina, unpublished data).
**4.12 Ostracod-based habitat reconstruction**
The LIG ostracod record obtained at both shores of the Dmitry Laptev Strait comprises 23 taxa, of which 20 were
identified at the species level, two at the genus level, and one taxon comprises juvenile Candoninae (Table S8).
The most common species which have occurrences in at least four of the five studied profiles are *Candona candida*,
*Fabaeformiscandona harmsworthi*, *F. levanderi*, *F. rawsoni*, *F. tricicatricosa*, *Eucypris dulcifons*, *Ilyocypris*
*lacustris, Cytherissa lacustris,* and *Limnocytherina sanctipatricii*. Exemplarily, the Oya 5-1 record (Kienast et al.,
2011) that revealed the most numerous record comprising >1000 specimens per sample belonging to a total of 11
species is dominated by *C. candida* (34 %), *Cy. lacustris* (26 %), and *F. rawsoni* (15 %).
The species *Cy. lacustris*, *F. tricicatricosa,* and *L. sanctipatricii* are adapted to cool water temperatures (Meisch,
2000), and *T.* cf. *glacialis*, *F. rawsoni,* and *F. harmsworthi* are cold-stenothermic (Wetterich et al., 2008a, 2008b).
Increased salinity in the water is tolerated by *L. sanctipatricii* (0.5-5 ‰), *C. lacustris* (up to 1.5 ‰) and *F. levanderi*
(up to 6 ‰; Meisch, 2000).



### 4.13 Clumped isotope temperature and thermokarst lake δ¹⁸O reconstructions from ostracod and bivalve calcite

Clumped isotope $\Delta_{47}$ values range between 0.641 and 0.658 ‰, with standard errors between 0.005 and 0.011 ‰ (95% confidence intervals between 0.01 and 0.022 ‰). Of the 54 sample replicates, three erroneous replicates (with stable or clumped isotope values greater than $\bar{x} \pm 2\sigma$), and one contaminated replicate ($\Delta_{48} > 1$ ‰) were removed from the final $\Delta_{47}$ calculations. The final values suggest carbonate precipitation temperatures (T$\Delta_{47}$) between 5.3 and 10.3°C. The raw data is presented in Table S9 and results are summarized in Table 5.

Fossil carbonate $\delta^{18}O$ values range from $-14.65\pm0.02$ ‰ to $-14.12\pm0.02$ ‰ VPDB, resulting in reconstructed water $\delta^{18}O$ estimates ($\delta^{18}O_w$) between $-18.9\pm0.3$ ‰ and $-17.8\pm0.6$ ‰ VSMOW.

**Table 5: Clumped isotope results from fossil biogenic carbonates from profile Oya 5-1. N = number of replicate measurements used to calculate $\Delta_{47}$ with the number of rejected samples in parenthesis. $\Delta_{47}$ and $\delta^{18}O$ of carbonate ($\delta^{18}O_{cc}$) uncertainties are given as external standard errors over multiple replicates. $\delta^{18}O_w$ is the estimated $\delta^{18}O$ of water from which the carbonate formed, with uncertainty estimated through the propagation of temperature and isotope uncertainties.**

| Sample | N | $\Delta_{47}$ [‰ ICDES] | T$\Delta_{47}$ [°C] | $\delta^{18}O_{cc}$ measured [‰ VPDB] | $\delta^{18}O_w$ [‰ VSMOW] |
|---|---|---|---|---|---|
| *Candona candida* | 18 (1) | 0.641 ± 0.011 | 10.2±3.2 | −14.12±0.02 | −18.6±0.7 |
| *Cytherissa lacustris* | 21 (1) | 0.641±0.010 | 10.3±3.0 | −14.31±0.02 | −17.8±0.6 |
| *Pisidium casertanum* | 15 (2) | 0.658±0.005 | 5.3±1.5 | −14.65±0.02 | −18.9±0.3 |

### 4.14 Paleoclimate modeling

Using the PaleoMIP lig127k model simulations (Otto-Bliesner et al., 2021; Kageyama et al., 2021, Table S10) together with the NOAA GlobTempV6 ERA5 data, we derived maps of climatological means of MTWA, MTCO, and MAP for the Bol'shoy Lyakhovsky and Oyogos Yar region according to the methodology described above. These maps cover the Laptev Sea to the west and north, the Eastern Siberian Sea to the east, and a small part of continental Siberia to the south and provide regional patterns of these climate variables in addition to values for the sample sites (Figure 9). Climate change signals for both warmest and coldest months as well as their ensemble spreads are in agreement with the seasonal signals shown in Otto-Bliesner et al. (2021), where warmer winters occur over the Arctic Ocean, albeit with a high inter-model spread, and warmer summers are evident dominantly over the continents and less pronounced over the Arctic ocean, associated with low inter-model spread. Note that due to the lack of a reference data set for pre-industrial precipitation, MAP values are PaleoMIP lig127k ensemble means.

The MTCO for the LIG derived from modeled anomalies are between –40 and –26 °C, with a distinct north-south gradient and colder temperatures over grid cells that include land (Figure 9). The modern MTCO is higher than modeled temperatures for the LIG with larger differences over the sea than over land. The model agreement is highest over grid cells on the continent distal to the coastline, lower over the ocean, and lowest along the coastline, which indicates that a significant fraction of the uncertainty is related to the different land-sea masks in the different models. This is reflected in the values calculated for the sample sites and the generic LandPoint (see Table 6), with the highest MTCO for Bol'shoy Lyakhovsky and lowest MTCO for the generic LandPoint in both the PMIP multi-model ensemble mean and ERA5, and a higher uncertainty for the sample points than for the generic land point.





LIG MTWA estimates from the PMIP multi-model ensemble mean range from 0°C over the sea to 18°C over the
south-western land area. Model temperatures are consistently higher than present-day temperatures from the ERA5
reanalysis. At the same time, the general spatial patterns are very similar (lowest temperatures over the northeastern
sea corner of the plotted area, highest temperatures over the southwestern land area). The agreement between the
PaleoMIP models is higher than for the MTCO, with good agreement in all areas except the coastlines, again
indicating that higher uncertainties are related to the different land-sea masks in the different models. MTWA
values of the sample points and the generic land point reflect the north-south gradient shown in the map (Figure
9), with the lowest MTWA values for Bol'shoy Lyakhovsky and the highest MTWA for the LandPoint. Both the
LandPoint and Bol'shoy Lyakhovsky are located in grid cells that are considered land by most models, leading to
an uncertainty only half as high as for Oyogos Yar, which is situated in a grid cell with varying land content.
MAP from the PMIP multi-model ensemble mean is between 250 and 450 mm, showing lower values over the sea
and higher values over the continental land area. This general pattern is similar to that of the present-day reanalysis
results, where precipitation is lower than those modeled for the LIG. The PMIP models show the highest agreement
over the sea and the lowest agreement over the continental grid cells.

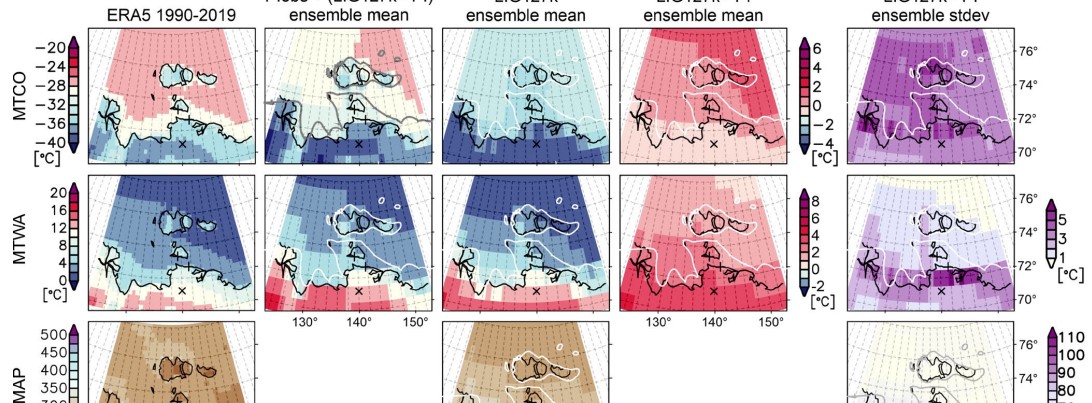

**Figure 9. Climatological means of monthly mean temperature of the coldest (MTCO, top row) and the warmest month**
**(MTWA, middle row) and of mean annual precipitation (MAP, bottom row) from the ERA5 reanalysis (left column),**
**PaleoMIP multi-model anomaly added to observed PI values (PIobs+(lig127k-PI) for temperatures and PaleoMIP**
**multi-model ensemble mean (lig127k ensemble mean) for precipitation, PaleoMIP ensemble mean anomaly with respect**
**to modeled PI (lig127k-PI ensemble mean) and PMIP multi-model anomaly standard deviation (ensemble stdev). The**
**plus signs denote the position of the sample sites, and the x signs denote the position of the generic LandPoint. Black**
**lines mark today's coastlines, the white resp. gray lines mark the coastlines of the lig127 (after Alekseev et al., 1991b)**





**Table 6: Evaluation of MTWA and MTCO from the PIobs+(lig127k-PI) multi-model ensemble mean and MAP from the PaloeMIP lig127k multi-model ensemble mean. Values in brackets refer to the present-day reference from ERA5 (1990-2019).**

| Site | MTWA [C°] | MTCO [C°] | MAP [mm] |
|---|---|---|---|
| Bol'shoy Lyakhovsky Island | 4.4±1.0 (2.7) | −31.1±1.4 (-32.7) | 278±50 (262) |
| Oyogos Yar | 4.5±1.2 (7.5) | −31.6±1.4 (-34.1) | 285±55 (243) |
| LandPoint | 13.6±0.9 (11.0) | −38.7±1.0 (−36.3) | 328±70 (259) |

## 5    Discussion

### 5.1    Cryolithology of LIG thermokarst deposits

The cryolithological parameters of the LIG deposits are shown in Figure 10, each in comparison to the stratigraphically younger Yedoma Ice Complex (mostly MIS 3) and Holocene (MIS 1) thermokarst deposits according to Schirrmeister et al. (2011b) and Wetterich et al. (2009) respectively.

The absolute ice contents of the LIG deposits are very similar for Bol'shoy Lyakhovsky Island and the Oyogos Yar coast and up to half as low as those of the Yedoma Ice Complex and Holocene deposits at both sites. Thus, the freezing and thawing processes and moisture content of LIG thermokarst lake sediments are similar but clearly different from those of the younger horizons. The MS for the LIG deposits is similar at both sites, showing comparably high contents of magnetic minerals. These contents are up to twice as high as those of the Yedoma Ice Complex and Holocene horizons. This could mean that the source material in the LIG was different.

The TOC contents show slight differences for both sites, with a mean of 2.4 wt% at Bol'shoy Lyakhovsky and 3.5 wt% at Oyogos Yar. In contrast, Yedoma Ice Complex and Holocene deposits show higher TOC values on Bol'shoy Lyakhovsky with 3.2 and 7.8 wt%, respectively, if compared to Oyogos Yar with 2.7 and 6.2 wt%, respectively. Different environmental conditions could play an important role and influence the preservation of organic matter. Regardless, the Holocene deposits contain two to three times higher wt% of organic matter. The TOC/TN ratio as an indicator for the source of/degree of decomposition of the organic matter in the LIG deposits is very similar on Bol'shoy Lyakhovsky and on Oyogos Yar. The values for the younger deposits are in a similar range (Figure 10), indicating a similar degree of decomposition of organic matter.

The mean carbonate content in LIG (2.3 wt%), Yedoma Ice Complex (0.5 wt%), and Holocene (1.6 wt%) deposits on Bol'shoy Lyakhovsky Island is much lower if compared to the respective horizons on Oyogos Yar with 6.4, 3.9, and 3.5 wt%, respectively. This reflects a higher share of shells of mussels, gastropods, and ostracods in the Oyogos Yar deposits. At each site, the Krest-Yuryakh deposits have the highest carbonate content.


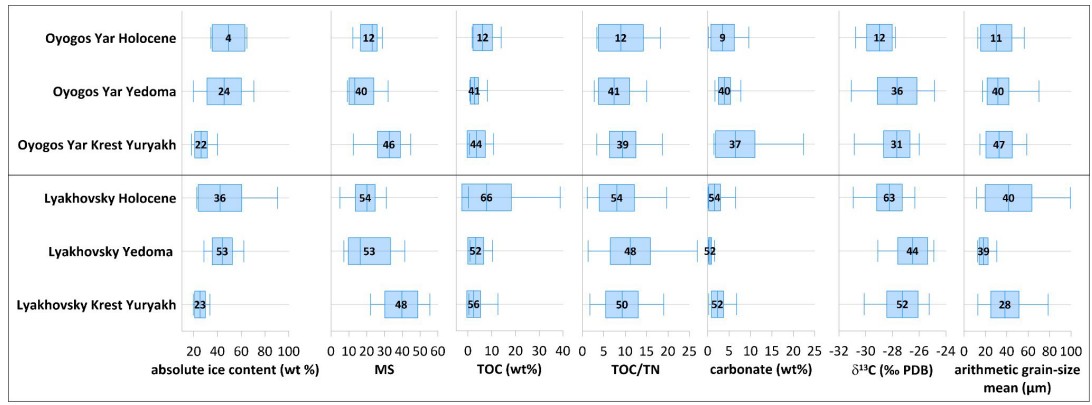

Figure 10. Boxplots of sediment data from both study sites, including the LIG (Krest-Yuryakh) horizon and the stratigraphic younger late Pleistocene Yedoma Ice Complex (mostly MIS 3) and Holocene thermokarst (MIS 1) horizons. The boxplots show the standard deviation, the arithmetic mean, and the IQR (interquartile range). The numbers correspond to the respective number of samples included (Schirrmeister et al., 2011a, 2011b; Wetterich et al., 2009).

The mean $\delta^{13}$C values of TOC for the LIG deposits are almost identical, with –27.3‰ on Bol'shoy Lyakhovsky and –27.7‰ on Oyogos Yar. The respective values from Yedoma Ice Complex deposits are –26.5‰ and –27.6‰, and from Holocene thermokarst deposits are lower with –28.3‰ and –29.0‰. The relationship between TOC/TN and $\delta^{13}$C indicates a mixture of organic matter derived from C3-type terrestrial plants and lacustrine algae. A large number of data points plot along the mixing line between both endmembers (Figure S4).

The mean arithmetic grain size for the LIG deposits is 38.2 µm on Bol'shoy Lyakhovsky and 32.8 µm on Oyogos Yar. The Yedoma deposits of Bol'shoy Lyakhovsky have a much smaller mean grain size of 18.2 µm, while the Holocene deposits are coarser, with a mean of 41.6 µm (Figure 10). The younger Yedoma and Holocene deposits of Oyogos Yar have a similar mean arithmetic grain size as the LIG sediments with 31.7 and 30.0 µm. This shows that the deposition conditions in the thermokarst lakes were similar in both areas during the LIG. On Bol'shoy Lyakhovsky, the deposition conditions and/or the sources of material in the stratigraphically younger units have changed more than at Oyogos Yar.

## 5.2 Last Interglacial chronology and dating uncertainties

Different geochronological method results are available for the reconstruction of the timescale of Arctic permafrost dynamic and corresponding investigations on feedbacks of periglacial landscapes and climate changes. Of core importance are radiocarbon dating (e.g. Wetterich et al., 2014), tephrachronology (e.g. Froese et al., 2009), optically- and infrared-stimulated luminescence (OSL and IRSL; e.g., Murton et al., 2022), radioisotope disequilibria ($^{230}$Th/$^{234}$U) of frozen peat (e.g. Wetterich et al., 2016), uranium isotope ($^{234}$U/$^{238}$U) series (Ewing et al., 2015), and $^{36}$Cl/Cl radionuclide ratios of ground ice (e.g. Blinov et al., 2009). However, dating methods ranging beyond radiocarbon maximum finite ages show specific challenges when applied to frozen material. Additionally, when fusing the dating results into one single multi-method chronology, a careful interpretation is required because individual methods use different components of permafrost deposits, including organic, mineral, or ice



components. Uncertainties also arise from unknown influences of freezing and thawing dynamics on chemical and
physical parameters, which are important to many age-determination techniques.
The chronostratigraphy of the LIG and its MIS 5 context relies on a few luminescence ages available from
Bol'shoy Lyakhovsky Island and the Oyogos Yar coast. The stratigraphic position at studied sites still outlines
challenges, especially from vertical discontinuities and hiatus. Currently, available age information of MIS 5-
related deposits is summarized in Figure 11. Sediments, which are stratigraphically older than Krest Yuryakh,
were found along the Laptev Strait coast as Buchchagy Ice Complex dated using 230Th/U to MIS 5e - MIS 5b
(126 +16/-13 ka to 89 ± 5 ka, Wetterich et al., 2019, Opel at al., 2017). The large variation of 230Th/U-ages
impedes highly resolved, millennial paleo-climate interpretations.
Stadial conditions are recorded in floodplain sediments locally named as Kuchchugui stratum that has been
cryostratigraphically aligned with the MIS 6 (Tumskoy and Kuznetsova, 2022),
IRSL ages from Bol'shoy Lyakhovsky of 102±16 and 99±15 ka suggest its generation during MIS 5c, a period
younger than the LIG. Nevertheless, corresponding deposits associated with the Kuchchugui stratum were dated
by IRSL to slightly older ages (112.5 +9.6 -102.4±9.7 ka) at the mainland coast of Oyogos Yar (Opel et al., 2017).
In this context, the newly IRSL-dated Krest Yuryakh thermokarst deposits provide, for the first time, robust age
control for the LIG based on three consistent ages from two samples and two grain size fractions and with smaller
age uncertainties compared to previous luminescence dating results., with ages of 127.3±6.1, 117.8±6.8, and
117.6±6.0 ka (Table 1; Figure 11).
The observed luminescence properties confirm a reliable luminescence - dose correlation for successful curve
fitting and age modeling. The coefficients of variation below 6 % testify suitable measurement conditions and
reproducible signals. The low standard deviation below 6 % and corresponding low skewness below 0.8 document
sufficient signal reset and call for the application of the central age model (CAM). The typical challenges in dating
permafrost samples related to sediment mixing are not indicated in our samples nor in the sampled sediment
section. The estimates of the paleo-water content were primarily based on the measured in-situ water content.
Although slight variations may have occurred over time, we assume that the sampled layers remained frozen and
hence, have kept the measured water content relatively constant. Nevertheless, uncertainties remain due to the fact
that the water was present not in its liquid but in its frozen form with potential effects on the penetration depth of
ionizing radiation. To account for the potential variations in the past plus uncertainties from radiation field
modeling based on water instead of ice, we included an overall water content uncertainty of 5 %. Site-specific
uncertainties may also arise from the unknown evolution (esp. with respect to timing) of the overburden and its
effects on the cosmogenic dose rate due to permafrost formation with sediment aggradation and permafrost
degradation with thaw subsidence. We evaluated the influence of an early reduction of overburden thickness by
comparing the full thickness of 35 m to 29 m, which would be more comparable to modern thickness assumptions.
As both overburden thicknesses are already strongly attenuating the incoming cosmic radiation, the effect on ages
is less than 1 %.



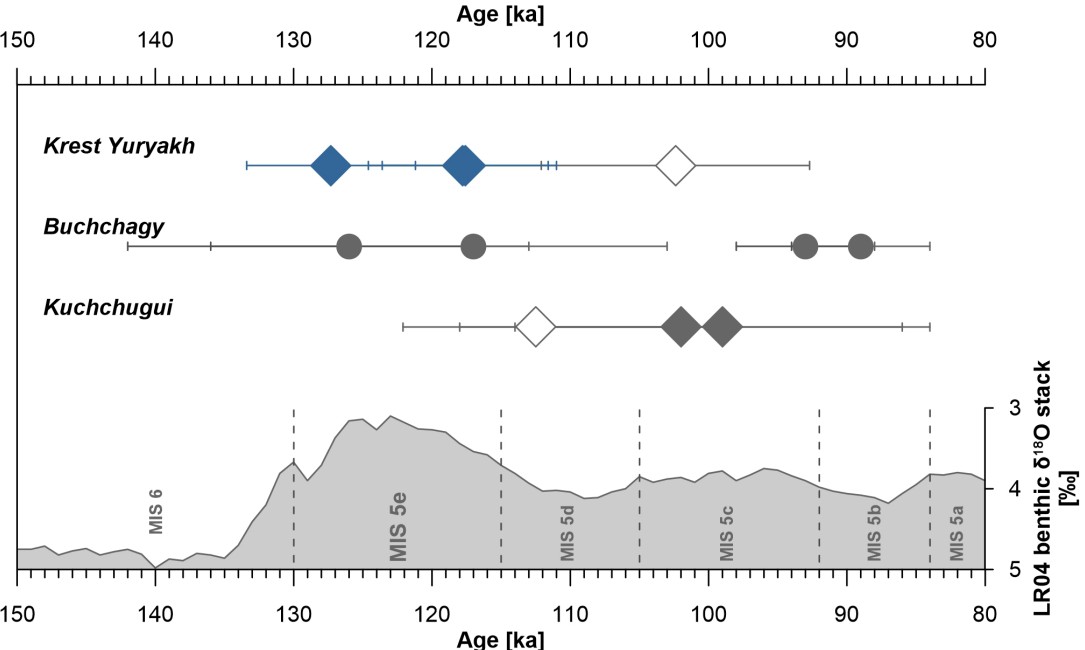

**Figure 11.** Age information obtained by IRSL (diamonds) and [230]Th/U dating (circles) of cryostratigraphic units exposed at both coasts of the Dmitry Laptev Strait and aligned with MIS 5. White symbols refer to samples from the Oyogos Yar mainland coast, and filled symbols to those from the southern coast of Bol'shoy Lyakhovsky Island – compared to the LR04 benthic stack (Lisiecki and Raymo, 2005). Age determinations of Krest-Yuryakh lake deposits are from this study (highlighted in blue; Bol'shoy Lyakhovsky) and Opel et al. (2017; Oyogos Yar), Buchchagy Ice Complex from Wetterich et al. (2016; Bol'shoy Lyakhovsky) and Kuchchugui floodplain deposits from Opel et al. (2017; Oyogos Yar) and Andreev et al. (2004; Bol'shoy Lyakhovsky).

### 5.3 Proxy-based quantitative paleoclimate reconstructions compared to PMIP model simulations

When we compare the proxy-based and modeled climate and environmental parameters, regional differences between Bol'shoy Lyakhovsky Island and the Oyogos Yar coast are obvious, as well as differences between different proxy reconstructions at the same site (Table 7). The general pattern, however, is that all proxy-based MTWA reconstructions indicate significantly warmer-than-today conditions.

For MTWA on Bol'shoy Lyakhovsky Island pollen, plant macrofossils, beetles, and chironomids overlap at 10.3-10.5 °C . The modeled temperatures (Plobs+(lig127k-PI)) are distinctly lower  (Figure 12). BrGDGT-based AIR GST are lower (2.8±0.3 °C) but fit well with the MTWA estimates since they integrate the entire growing season of bacteria. At Oyogos Yar, the MTWA of pollen, beetles, and chironomids overlaps at 12-12.6 °C, while plant macrofossils indicate higher MTWA. The modeled temperatures for Oyogos Yar are again distinctly lower but are similar to those of Bol'shoy Lyakhovsky. The mean temperatures of the coldest month (MTCO) show joint overlaps between beetles and model data between –32.6 and –29.8 °C for Bol'shoy Lyakhovsky Island and –33.0 and –30.2 °C. For the mean annual precipitation (MAP), pollen and model data of PaleoMIP concur between 228-327 mm for Bol'shoy Lyakhovsky Island (i.e., higher than at present, which is also implied by GDGTs) and 230-251 mm for Oyogos Yar. The WD of thermokarst lakes, which were reconstructed using chironomids, overlap between 1.7 and 3.3 m for both sites. The clumped-isotope-reconstructed water temperature of these lakes is 10.2±3.0 and 10.3±3.2 °C near the surface and 5.3 °C at the bottom.



The generic LandPoint data of PaleoMIP overlap quite well for MTWA with most of the proxy data from Oyogos
Yar, with chironomid data, and a bit with plant macrofossil data from Bol'shoy Lyakhovsky (Figure 12). For
MTCO, a small overlap is visible with beetle data from Oyogos Yar. For MAP PaleoMIP reconstruction, the three
points and pollen data from Bol'shoy Lyakhovsky overlap between 258 and 327 mm.
The differences between different proxy reconstructions and between the two locations can have various causes.
For example, pollen, plant macrofossils, beetles, and chironomids may have reached their detectable stage in the
fossils at different times during the summer months. Thus, different conditions may have prevailed between early,
peak, and late summer during MIS 5e. On the other hand, MIS 5e lasted about 9 ka between 125 and 116 ka (van
Nieuwenhove et al., 2011). Unfortunately, the dating results available to us do not allow us to determine more
precisely whether our records relate to early, middle, or late MIS 5e. Therefore, differences may also have
stratigraphic and/or chronological causes.
**Table 7: LIG proxy-based quantitative reconstructions and paleoclimate modeling results of Mean Air Temperature of**
**the Warmest Months (MTWA), Mean Temperature of the Coldest Month (MTCO), Growing Season Temperature (Air**
**GST, April to October), Mean Annual Precipitation (MAP), and Water Depth (WD) and Water Temperature ($T_{water}$)**
**of thermokarst lakes on Bol'shoy Lyakhovsly Island and the Oyogos Yar mainland coast. Values in brackets refer to**
**the present-day reference from ERA5 (1990-2019). n/a – not applicable.**

| Fossil proxy | MTWA [°C] | Air GST [°C] | MTCO [°C] | MAP [mm] | WD [m] | Open $T_{water}$ [°C] | Bottom $T_{water}$ [°C] |
|---|---|---|---|---|---|---|---|
| **LandPoint** | | | | | | | |
| PaleoMIP Modeling | n/a | n/a | n/a | 328±70 | n/a | n/a | n/a |
| PIobs+(lig127k-PI) | 13.6±0.9 | n/a | −38.7±1.0 | n/a | n/a | n/a | n/a |
| (ERA5 reanalysis) | (11.0) | n/a | (−36.3) | (259) | n/a | n/a | n/a |
| **Oyogos Yar coast** | | | | | | | |
| PaleoMIP Modeling | n/a | n/a | n/a | 285±55 | n/a | n/a | n/a |
| PIobs+(lig127k-PI) | 4.5±1.2 | n/a | −31.6±1.4 | n/a | n/a | n/a | n/a |
| (ERA5 reanalysis) | (7.8) | n/a | (−34.4) | (243) | n/a | n/a | n/a |
| Clumped isotopes (*Pisidium casertanum*) | n/a | n/a | n/a | n/a | n/a | n/a | 5.3±1.5 |
| Clumped isotopes (*Cytherissa lacustris*) | n/a | n/a | n/a | n/a | n/a | 10.3±3.0 | n/a |
| Clumped isotopes (*Candona candida*) | n/a | n/a | n/a | n/a | n/a | 10.2±3.2 | n/a |
| Chironomids | 12.0 to 13.8 | n/a | n/a | n/a | 1.1 to 3.3 | n/a | n/a |
| Beetles | 8 to 14.0 | n/a | −38.0 to −26.0 | n/a | n/a | n/a | n/a |
| Plant macrofossils | 12.7 to 15.3 | n/a | n/a | n/a | n/a | n/a | n/a |
| Pollen | 9.7±2.9 | n/a | n/a | 229±22 | n/a | n/a | n/a |
| **Bol'shoy Lyakhovsky Island** | | | | | | | |
| PaleoMIP Modeling | n/a | n/a | n/a | 278±50 | n/a | n/a | n/a |





| | | | | | | | |
|---|---|---|---|---|---|---|---|
| PIobs+(lig127k-PI) | 4.4±1.0 | n/a | –31.2±1.4 | n/a | n/a | n/a | n/a |
| (ERA5 reanalysis) | (2.5) | n/a | (–33.1) | (262) | n/a | n/a | n/a |
| Air GST (GDGTs) | n/a | 2.8 ± 0.3 | n/a | n/a | n/a | n/a | n/a |
| Chironomids | 9.4 to 15.3 | n/a | n/a | n/a | 1.7 to 5.6 | n/a | n/a |
| Beetles | 8.0 to 10.5 | n/a | –34 to –26 | n/a | n/a | n/a | n/a |
| Plant macrofossils | 10.3 to 12.9 | n/a | n/a | n/a | n/a | n/a | n/a |
| Pollen | 9.0±3.0 | n/a | n/a | 271±56 | n/a | n/a | n/a |

Present-day ERA5 data has a severe bias over sea ice, especially during periods with cold temperatures; temperatures are overestimated by up to 10°C when the air temperature is around –40°C (Wang et al., 2019, Batrak & Müller, 2019). The differences between PaleoMIP model's land-sea mask and the actual coastline during the LIG, concerning especially MAP and MTWA, might lead to underestimated MAP and MTWA. The MTWA for the generic land point (13.6±0.9 °C, Table 6) is in the range of the reconstructed data sets between about 6 and 15 °C (Table 7). On the other hand, analog-based climate reconstruction methods applied to biotic proxies may overestimate temperatures in northern Siberia (Klemm et al., 2013). The sites are presently located at the lower temperature range of the training dataset, and as such, the taxa present in the observations are not covered with their full occurrence range, which typically results in biases toward higher temperature values.

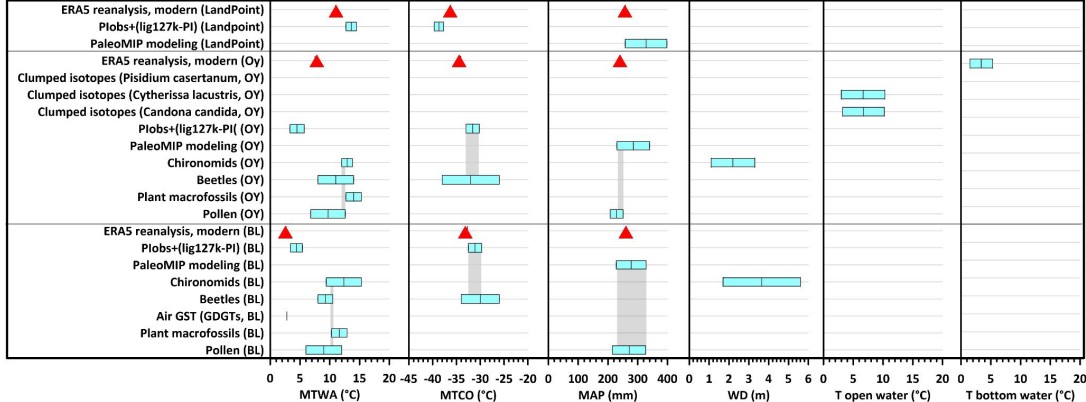

**Figure 12. Boxplot of proxy-reconstructed and modeled climate and environmental data according to Table 7. BL – Bol'shoy Lyakhovsky, OY – Oyogos Yar, MTWA - mean temperature of the warmest month, MTCO – mean temperature of the coldest month, WD – water depth of thermokarst lakes, T open water – surface water temperature of lakes, T bottom water – bottom water temperature of lakes. Grey-shaded areas indicate overlap ranges; red triangles indicate the present-day reference.**

### 5.4    Biogenic carbonate: Clumped and stable isotopes

All species analyzed (*Candona candida, Cytherissa lacustris, Pisidia casertanum*) require freshwater habitats and thus depend on the presence of open water bodies, which are present solely in summer in modern times. Thus, the



derived temperatures are interpreted as reflecting the mean temperature of the warm season that integrates all ice-
free periods, i.e., the growth window of ostracods and bivalves.
Ostracod-derived clumped isotope temperatures (T$\Delta_{47}$) are considerably warmer (10.2±3.2 and 10.3±3.0 °C) for
*C. candida* and *Cy. lacustris*, respectively) than those of *P. casertanum* (+5.3±1.5 °C) (Table 5). This difference
is attributed to differing habitation depths, with ostracods proliferating throughout the water column (Scharf, 1998;
Decrouy et al., 2012), and therefore recording surface and upper water column temperatures, and *P. casertanum*,
a benthic species that burrows continually into sediments (McMahon and Bogan, 2001) and thus records bottom
water temperature. Monitoring data in a polygon pond at Oyogos Yar during August 2007 shows that mean air
and surface water temperatures differed by only 0.3°C, while bottom water at 0.6 m WD and only about 0.2 m
above the permafrost table showed almost constant water temperature of 3.6 °C (Boike et al., 2008). Thus, we
conclude that our reconstructed ostracod temperatures are likely similar to, or slightly below, warm season mean
surface air paleotemperature. Our summer surface T$\Delta_{47}$ reconstructions of ca. 10 °C are considerably warmer (ca.
+2 °C) than modern and in good agreement with other air temperatures derived in this study.
Our measured *Cy. lacustris* $\delta^{18}$O values show good agreement with those previously recorded from the LIG at
Oyogos Yar in section Oy7-08 (−14.5 to −12.2 ‰, Wetterich et al., 2009). However, our *C. candida* values are ca.
2 to 3 ‰ more negative than those previously recorded in Oy7-08 (–12.6 to –11.3 ‰, Wetterich et al., 2009).
Wetterich et al. (2008a) showed that offsets in $\delta^{18}$O between single species in neighboring polygon ponds of 2 to
3 ‰ are common and this likely could explain the discrepancies observed here.
Stable isotope compositions of ostracod calcite are driven by temperature and the isotopic composition of the
surrounding precipitation waters, making them effective proxies for paleoenvironmental conditions (Xia et al.,
1997; von Grafenstein et al., 1999). For comparison with our paleorecord, modern *C. candida* $\delta^{18}$O has been
measured at Samoylov Island (72.37°N, 126.48°E), in the Lena Delta, ranging between −17.7 and −10.4 ‰
(Wetterich et al., 2008a), the Moma region (NE Yakutia, 66 °N, 143 °E) where values between –15.2 and –11.9
‰ were recorded, and in Central Yakutia (62 °N, 129 °E) with values between –11.0 and–8.9 ‰ (Wetterich et al.,
2008b). The more continental location of the latter two records, compared to the Samoylov record, induces higher
temperature amplitudes, warmer summers, and higher evaporation and is reflected in more enriched (less negative)
modern carbonate $\delta^{18}$O. The similarity of Oya 5-1 data to Samoylov Island, compared with the more continental
locations, suggests a precipitation and temperature regime closer to the former site in the modern day. Samoylov
Island is at a similar latitude as Oyogos Yar but records mean July air temperatures between 4 to 8 °C,
approximately 1 to 5 °C warmer due to its delta setting influenced by warm Lena River water (Boike et al., 2019).
Calculated $\delta^{18}$O of carbonate precipitation waters between –18.5 and –16.6 ‰ are similar to modern summer
rainfall values, which range from –20.2 to –11.7 ‰ (Opel et al., 2011). Small water bodies, such as those inhabited
by ostracods, have been shown to be predominantly fed by precipitation in northern Siberia (Wetterich et al.,
2008a). Thus, their $\delta^{18}$O will largely reflect that of precipitation with a small shift towards more positive values
(ca. +2 ‰), driven by evaporative processes (Wetterich et al., 2008b). The similarity of our reconstructed water
$\delta^{18}$O to modern precipitation would suggest a summer precipitation regime similar to the modern day.
**5.5     Last Interglacial ecosystems**
The paleobotanical and entomological data prove the presence of an open subarctic shrub tundra with restricted
steppe areas in the area of Bol'shoy Lyakhovsky Island and the existence of an open forest tundra at Oyogos Yar
during the LIG. The presence of the ant *Leptothorax acervorum* and the true bug *Sciocoris microphthalmos* in the



Oyogos Yar records is notable as both species at present inhabit forested areas, while tundra-steppe indicators are
rare and only represented by *Morychus viridis* and the meadow-steppe species *Protapalochrus arcticus* (former
*Troglocollops arcticus*). True steppe insects are not present in Oyogos Yar but two steppe weevils
(*Stephanocleonus eruditus* and *S. fossulatus*) were recorded in the Bol'shoy Lyakhovsky fauna. The presence of
tall boreal shrubs and even trees during the LIG in an area that is covered with arctic tundra today is noteworthy.
The only woody plants that are currently occurring at Oyogos Yar are the dwarf shrubs *Dryas octopetala* ssp.
*punctata* and *Salix polaris*, the latter forming thin prostrate stems piercing through and protected by the moss
cover. Ericaceae, Betulaceae, or other mainly boreal taxa do not occur in the study area today (Aleksandrova,
1980; Kienast et al., 2008) but were represented by many species during the LIG, e.g., *Betula pendula, B.*
*divaricata, B. fruticosa, B. nana* s.l., *Arctostaphylos uva-ursi, Andromeda polifolia, Chamaedaphne calyculata,*
*Rhododendron tomentosum,* and *Vaccinium vitis-idaea*. The majority of plant species recovered in LIG deposits
at both sides of the Dmitry Laptev Strait are extralimital, i.e., they do not occur in high arctic tundra today but
considerably further south. At Oyogos Yar, only ca. 80 km to the southeast of the Bol'shoy Lyakhovsky site, tree
species like *Larix gmelinii, Betula pendula* s.l*.,* and *Alnus hirsuta* were already present during the LIG, indicating
that the treeline was shifted about 270 km further north than currently (Kienast et al., 2011). The nearest known
modern occurrence of grey alder is located at the Sobolokh Mayan River 910 km southwest of Oyogos Yar
(Krasnoborov and Malyschev, 2003; GBIF, 2023).
Numerous coprophilous Sordariaceae fungi spores found in our LIG deposits point to the presence of grazing herds
during this time. In West Beringia, forest-steppe-tundra provided grazing areas for large Pleistocene mammals so
they could survive the Pleistocene interglaciations (Kuzmina, 2015b).
Plant-derived *sed*aDNA shows substantial overlap with the plant macrofossil record in recovering taxa that do not
occur at Bol'shoy Lyakhovsky today, including *Betula*, several Ericaceae species, and *Potamogeton*. The
taxonomic composition in profile L14-04 from Bol'shoy Lyakhovsky Island indicates that the interglacial
vegetation was likely a mosaic of subarctic shrub-tundra and dry steppe communities with arctic-alpine and pioneer
plants. The record shows many similarities in the sedaDNA plant record with the MIS 5e record from the Batagay
megaslump (Courtin et al., 2022).
A major discrepancy to prior studies but also a highlight of our record is the detection of *Larix*, *Picea*, and *Populus*
in the interglacial strata, suggesting that the treeline reached indeed as far north as Bol'shoy Lyakhovsky Island
(73.3 °N). *Larix*-DNA was detected in several samples, all of which showed at least 2% *Larix* in the pollen
assemblage (Zimmermann et al., 2017). The pollen records of *Larix* are usually underrepresented because pollen
is produced in low quantities, and due to a high fall speed, the potential for long-distance dispersal is low (Sjögren
et al., 2008; Jørgensen et al., 2012; Niemeyer et al., 2015). Hence, single pollen grains have been accepted as
evidence for the local presence of *Larix* (Edwards et al., 2005). We verified the presence of *Larix*-DNA by re-
amplification and re-sequencing with *Larix*-specific primer pairs, corroborating the authenticity of even low *Larix*
sequence counts in the metabarcoding record. Beetle remains and plant macrofossils derived from profile R35,
about a kilometer away from this site, indicate subarctic shrub tundra and the absence of trees at Bol'shoy
Lyakhovsky Island. Hence, we conclude that *Larix* was heterogeneously distributed in the landscape of Bol'shoy
Lyakhovsky Island, likely as individual trees or stands of trees within an open forest tundra, where trees were
growing in more protected sites with a favorable microclimate. Evidence of the presence of *Larix* trees as far north
as Kotelny Island during MIS 3 (van Geel et al., 2017) substantiates our assumptions.





In our LIG *sed*aDNA record, *Larix* was mostly accompanied by evergreen *Picea* and deciduous, broadleaved
*Populus*, yet the absence of macrofossil and pollen evidence requires a critical evaluation. The genetic marker
used in this study is located on the chloroplast genome, which in the genus *Picea* is inherited paternally via pollen
(Sutton et al., 1991). As such, *Picea* pollen, susceptible to long-distance transport, could be a source of DNA in
our record. Indeed, samples in which *Picea*-DNA was detected contained relatively high amounts of *Picea* pollen
grains. However, not all samples in which pollen proportions were relatively high also contained *Picea*-DNA,
rendering this explanation uncertain as well. Moreover, previous work implies that pollen may not be the source
of *Picea sed*aDNA (Sjögren et al., 2017; Niemeyer et al., 2017; Parducci et al., 2017) likely because the pollen
exine is very durable, making the pollen resistant to degradation but at the same time prevent the release of DNA
during our gentle DNA extraction procedure. A similar discussion about the authenticity of *Picea abies* in a
*sed*aDNA record from the Ural Mountains was led by Clarke et al. (2020), where findings of stomata were
considered strong evidence for local occurrence. One of our samples indeed contained *Picea* cf. stomata, but
identifying the species-specific source of stomata was difficult, and they cannot be considered as strong evidence.
In contrast to *Picea*, the chloroplast genome of *Populus* is maternally inherited, and therefore, long-distance
transported pollen as a source of *Populus*-DNA can be excluded (Rajora and Dancik, 1992). Today, *Populus*
*suaveolens* populations occur relatively far north, with the northernmost documented individual in the Lena Delta
(73.2 °N, 128.6 °E; Seregin, 2023). As modern woody taxa penetrate the subarctic tundra belt along rivers
(Aleksandrova, 1980), it is possible that during the LIG, *P. suaveolens* could have progressed into our study area
at riversides and floodplains. The map "Late Pleistocene (Riss-Würm), 120 000 Years" of Alekseev et al. (1991b)
shows river valleys and temporary lakes on Bol'shoy Lyakhovsky Island and around Svyatoy Nos. In addition,
most of the area is labeled as relict alluvial and lacustrine plains.
Unless macrofossil evidence confirms the presence of *Picea* or *Populus* in the area, the question of whether these
taxa were truly part of the interglacial vegetation cannot be answered fully. However, given that (1) absence of
evidence is not evidence of absence, (2) *Populus* has been detected more often in the DNA than in the macrofossil
record (Kjær et al., 2022) or the surrounding vegetation (Alsos et al., 2018), (3) the findings of extralimital species
in interglacial strata (Kienast et al., 2008), and (4) the uncertainty about alternative DNA sources, the possibility
remains that *Picea* and *Populus* were growing that far north during the LIG.
Aquatic macrophytes such as *Potamogeton perfoliatus*, *Stuckenia filiformis, S. vaginata, Callitriche*
*hermaphroditica*, and *Batrachium* sp. were abundant during the LIG but are completely absent today. These
findings are another indication of warmer than present-day conditions (Kienast et al., 2008, 2011; Stoof-
Leichsenring et al., 2022). The aquatic freshwater LIG habitats are further characterized by fossil chironomid,
cladocera, ostracod, and mollusk communities. The presence of semi-terrestrial chironomid taxa at high
abundances and low abundances of cold-stenotherm taxa can indicate a transition from the preceding colder MIS
6 conditions to a warmer MIS 5e setting and probably correspond to the lower part of the R35 section on Bol'shoy
Lyakhovsky Island (Ilyashuk et al., 2006). Here, the subdominant taxa *Chironomus anthracinus*-type, *Cricotopus*
*laricomalis*-type, *Tanytarsus pallidicornis*-type, and *Endochoronomus albipennis*-type prefer relatively warm and
productive lakes (Nazarova et al., 2015, 2023), while *Tanytarsus lugens*-type and *Parakiefferiella triquetra*-type
taxa are in contrast cold stenotherm and characteristic for oligotrophic cold subarctic lakes. *Limnophyes,*
*Metriocnemus eurynotus*-type, and *Parametriocnemus- Paraphaenocladius* are typical for lake-level fluctuations
(Massaferro and Brooks, 2002). *Smittia* might indicate shore erosion processes or unstable lake-level conditions
(Brooks et al., 2008). The Oya 5-1 chironomid record from Oyogos Yar does not resolve lake development as R35



from Bol'shoy Lyakhovsky Island but instead aggregates different lake stages into one paleo-assemblage. The
Oya5-1 cladoceran assemblages indicate habitats with a well-developed vegetated shallow littoral zone as well as
pelagic open-water zones in the paleo-lake. The reconstructed WDs from chironomid data of 1.7±0.9 m to 5.6±1.0
m for Bol'shoy Lyakhovsky and 2.2±1.1 m for Oyogos Yar are within the range of measurements of modern
thermokarst lakes of 1 to 6 m depth (e.g., Wilcox et al., 2022, Morgenstern et al., 2013, Kallistova et a., 2019 and
references therein).
The LIG ostracod assemblages are characterized by species that tolerate considerable changes in temperature and
salinity regimes that are comparable to modern conditions in the periglacial landscapes of East Siberia, where
species such as *Candona candida*, *Fabaeformiscandona rawsoni*, and *Limnocytherina sanctipatricii* occur (Wetterich
et al., 2008a, 2008b). Single findings of the thermophilous species *Cyclocypris ovum* – not present in modern
environments of North Yakutia – indicate summer conditions distinctly warmer than today (Kienast et al., 2011).
The presence of benthic ostracods in lacustrine LIG deposits further indicates a sufficiently high oxygen content
in the water column of the host waters. Furthermore, the waters must have been ice-free for a certain period during
summer to facilitate the ostracod larvae development. As ostracods in high latitudes are apparently adapted to a
relatively short ice-free period, they conduct asexual reproduction (parthenogenesis), which explains relatively
high shares of adult females and only rare males in the fossil assemblages (Meisch, 2000). The high number of
juvenile shells indicates short summers in which the development of ostracods did not always reach its final stage.
The often-complete preservation of the fragile ostracod shells points to stillwater habitats and deposition, as well
as *in-situ* preservation. The shallow sublittoral zone of thermokarst lakes and ice-wedge casts formed by melting
wedge ice are the most probable habitats for the fossil ostracod assemblage (Wetterich et al., 2009; Kienast et al.,

21  2011).

The discrepancy in the species inventories of the last and the current interglacials is certainly the result of climatic
differences, i.e., the recent phase – the late Holocene – is comparatively cold in comparison to earlier warm stages
in Northern Siberia. This is even true for the MIS 3 Interstadial when larch trees existed as far north as Kotel'ny
Island (van Geel et al., 2017). Larch and other woody plants probably spread northward during the LIG when
Kotel'ny Island was still part of the mainland. The connection of the New Siberian Archipelago with the mainland
during the LIG can be deduced from the highly continental climate character existing adjacent to today's Dmitry
Laptev Strait. The persistence of this potential feeding base for herbivores during preceding warm stages helps
explain why large cold-adapted grazers survived earlier interglacials in Beringian refugia, which became centers
of their dispersal during subsequent cold stages.

## 6    Conclusions

New IRSL ages confirm the LIG (MIS 5e) origin of the Krest Yuryakh ice-wedge pseudomorphs and lake
sediments that are exposed at both coasts of the Dmitry Laptev Strait. The present study results are consistent with
previous work interpreting the MIS 5e as a key warm period during which extensive permafrost thawing and
thermokarst development occurred in north-eastern Siberia. The cryolithological features observed in the LIG
deposits bear striking similarities to those of Lateglacial and early Holocene refrozen thermokarst deposits,
suggesting comparable processes of deposition that were characterized by ground ice melting, surface subsidence,
and thermokarst formation driven by climate warming.





Paleoclimate data synthesized from a variety of proxies – including plant macrofossils, aquatic and terrestrial
invertebrates, and lipid biomarkers – indicate that temperatures during the LIG were significantly warmer than
today. Mean temperatures of the warmest month (MTWA) reconstructed from proxies show a range of 6.0°C
(pollen data of Bol'shoy Lyakhovsky) and 15.3 °C (plant macrofossils of Bol'shoy Lyakhovsky and chironomid
of Oyogos Yar) and overlap of 10.3°C and 10.7°C for Bolshoy Lyakhovsky) and 12.0 and 12.6 °C for Oyogos
Yar, demonstrating the pronounced warming of this period. However, one of the critical challenges in predicting
future ecosystem responses lies in the fact that the land-ocean distribution during the LIG was markedly different
from today, affecting the degree of continentality, which played a major role in modulating climate and ecosystem
dynamics.
Paleoclimate models generally agree well with the mean temperature of the coldest month (MTCO) proxy data but
consistently underestimate the mean temperature of the warmest month (MTWA) across proxy records when using
modern land-sea configurations. This mismatch is significantly reduced when models incorporate land-sea
distributions that more closely reflect those of the LIG. This adjustment highlights the importance of considering
past land-sea configurations in regional paleoclimate modeling when comparing proxy and model results, a critical
step in refining our understanding of Arctic climate dynamics during MIS 5e.
The strong ecosystem response to the LIG warming, reflected in the high diversity of proxies, shows the sensitivity
of permafrost regions to rising temperatures. In particular, the development of thermokarst landscapes created a
mosaic of terrestrial, wetland, and aquatic habitats, fostering an increase in biodiversity. This biodiversity is
evident in the rich variety of terrestrial insects, vegetation, and aquatic invertebrates preserved in these deposits.
In addition, the tree line extended 270 km further north during the LIG than it does today, yet the cold-adapted
mammoth fauna managed to persist in this region, probably finding refuge in the microclimates created by the
thermokarst landscape.
While the LIG is often used as an analog for future climate warming in the Arctic, there are important differences.
Most notably, the LIG warming was driven primarily by increased summer insolation, whereas current Arctic
warming so far is most pronounced in winter due to anthropogenic forcing and climate system feedback
mechanisms. This seasonal distinction is crucial because many of the environmental changes relevant today,
particularly those related to ecosystem processes and permafrost dynamics, occur during the transitional seasons
(spring and fall) - for which we currently lack proxy data from MIS 5e.
Ultimately, our results highlight the complexity of Arctic climate responses and emphasize the sensitivity to
seasonal factors, which are important aspects of future climate scenarios. Nevertheless, the lessons learned from
MIS 5e, particularly regarding thermokarst development and ecosystem adaptation to warming, provide valuable
insights into the potential future trajectory of permafrost regions in the context of ongoing climate change. Further
research is essential, particularly to fill gaps in proxy data for transitional seasons and to refine model-proxy
comparisons to improve our predictions of Arctic climate dynamics.





**7      Appendices**
Table A1: Overview of sample collections for cryolithological and fossil proxy studies of Krest Yuryakh deposits
exposed at both coasts of the Dmitry Laptev Strait.
**8      Code availability**
not applicable
**9      Data availability**
Andreev, A.A., Grosse, G., Schirrmeister, L., Kuznetsova, T.V., Kuzmina, S.A., Bobrov, A.A., Tarasov, P.E.,
Novenko, E.Y., Meyer, H., Derevyagin, A.Yu., Kienast, F., Bryantseva, A., Kunitsky, V.V. (2010): Pollen records
from Bol'shoy Lyakhovsky Island, Siberia. PANGAEA, https://doi.org/10.1594/PANGAEA.736069,
Andreev, A.A., Grosse, G., Schirrmeister, L., Kuznetsova, T.V., Kuzmina, S.A., Bobrov, A.A., Tarasov, P.E.,
Novenko, E.Y., Meyer, H., Derevyagin, A.Yu., Kienast, F., Bryantseva, A.,Kunitsky, V.V. (2010): Pollen record
of profile L11. PANGAEA, https://doi.org/10.1594/PANGAEA.736068,
Kienast, F., Schirrmeister, L. (2017): Plant macrofossil records from permafrost deposits of the Bol'shoy
Lyakhovsky Island (New Siberian Archipelago). PANGAEA, https://doi.org/10.1594/PANGAEA.882619
Kusch, S. (2021): GDGT data in Siberian permafrost deposits. PANGAEA,
https://doi.org/10.1594/PANGAEA.934054
Schirrmeister, L., Grosse, G., Kunitsky, V.V., Siegert, C. (2017): Sedimentological, biogeochemical and
geochronological data from permafrost exposures of the Bol'shoy Lyakhovsky Island (Expedition 1999), site
R23+40. PANGAEA, https://doi.org/10.1594/PANGAEA.880949
Schirrmeister, L., Grosse, G., Kunitsky, V.V., Siegert, C. (2017): Sedimentological, biogeochemical and
geochronological data from permafrost exposures of the Bol'shoy Lyakhovsky Island (Expedition 1999), site L14.
PANGAEA, https://doi.org/10.1594/PANGAEA.880937
Schirrmeister, L., Grosse, G., Kunitsky, V.V., Siegert, C. (2017): Sedimentological, biogeochemical and
geochronological data from permafrost exposures of the Bol'shoy Lyakhovsky Island (Expedition 1999), site
R22+60. PANGAEA, https://doi.org/10.1594/PANGAEA.880948
Schirrmeister, L., Grosse, G., Kunitsky, V.V., Siegert, C. (2017): Sedimentological, biogeochemical and
geochronological data from permafrost exposures of the Bol'shoy Lyakhovsky Island (Expedition 1999), site
L11+40. PANGAEA, https://doi.org/10.1594/PANGAEA.880935
Schirrmeister, L., Grosse, G., Kunitsky, V.V., Siegert, C. (2017): Sedimentological, biogeochemical and
geochronological data from permafrost exposures of the Bol'shoy Lyakhovsky Island (Expedition 1999), site
L13+80. PANGAEA, https://doi.org/10.1594/PANGAEA.880936
Schirrmeister, L. (2009): Lithology, color and structural description of sediment profile L7-11, Appendix 6.1.
PANGAEA, https://doi.org/10.1594/PANGAEA.727667
Schirrmeister, L. (2009): Documentation of ice wedge L7-11. PANGAEA,
https://doi.org/10.1594/PANGAEA.727710
Schirrmeister, L. (2009): Lithology, color and structural description of sediment profile L7-16, Appendix 6.1.
PANGAEA, https://doi.org/10.1594/PANGAEA.727671
Schirrmeister, L. (2009): Documentation of ice wedge L7-16. PANGAEA,
https://doi.org/10.1594/PANGAEA.727714
Schirrmeister, L. (2009): Lithology, color and structural description of sediment profile L7-14, Appendix 6.1.
PANGAEA, https://doi.org/10.1594/PANGAEA.727669



Schirrmeister, L. (2009): Documentation of ice wedge L7-14. PANGAEA,
https://doi.org/10.1594/PANGAEA.727712
Schirrmeister, L. (2009): Lithology, color and structural description of sediment profile L7-08, Appendix 6.1.
PANGAEA, https://doi.org/10.1594/PANGAEA.727666
Schirrmeister, L. (2009): Documentation of ice wedge Oy7-01. PANGAEA,
https://doi.org/10.1594/PANGAEA.727717
Schirrmeister, L. (2009): Lithology, color and structural description of sediment profile Oy7-01, Appendix 6.1.
PANGAEA, https://doi.org/10.1594/PANGAEA.727673
Schirrmeister, L. (2009): Documentation of ice wedge Oy7-07. PANGAEA,
https://doi.org/10.1594/PANGAEA.727725
Schirrmeister, L. (2009): Lithology, color and structural description of sediment profile Oy7-07, Appendix 6.1.
PANGAEA, https://doi.org/10.1594/PANGAEA.727676
Schirrmeister, L. (2009): Documentation of ice wedge Oy7-08. PANGAEA,
https://doi.org/10.1594/PANGAEA.727734
Schirrmeister, L. (2009): Lithology, color and structural description of sediment profile Oy7-08-A/B, Appendix
6.1. PANGAEA, https://doi.org/10.1594/PANGAEA.727677
Strauss, J., Laboor, S, Schirrmeister, L., Grosse, G., Fortier, D., Hugelius, G., Knoblauch, C., Romanovsky, V.E.,
Schädel, C., Schneider von Deimling, T., Schuur, E.A.G., Shmelev, D., Ulrich, M., Veremeeva, A., (2020):
Geochemical, lithological, and geochronological characteristics of sediment samples from thermokarst deposits in
Siberia and Alaska 1998-2016. PANGAEA, https://doi.org/10.1594/PANGAEA.919062,
Schwamborn, G., Wetterich, S. (2016): Geochemistry and physical properties of permafrost core L14-04.
PANGAEA, https://doi.org/10.1594/PANGAEA.868983
Schwamborn, G., Wetterich, S. (2016): Characteristics of samples obtained during the expedition to Bol'shoy
Lyakhovsky Island in July/August 2014. PANGAEA, https://doi.org/10.1594/PANGAEA.859265
Schwamborn, G., Wetterich, S. (2016): Sample list and field descriptions of the L14 profiles studied in summer
2014. PANGAEA, https://doi.org/10.1594/PANGAEA.859305
Zimmermann, H.H., Raschke, E., Epp, L.S., Stoof-Leichsenring, K.R., Schirrmeister, L., Schwamborn, G.,
Herzschuh, U. (2017): Pollen profile of sediment hand-pieces L14-04B. PANGAEA,
https://doi.org/10.1594/PANGAEA.878885,
Zimmermann, H.H., Raschke, E., Epp, L.S., Stoof-Leichsenring, K.R., Schirrmeister, L., Schwamborn, G.,
Herzschuh, U. (2017): Pollen profile of sediment hand-pieces L14-04C. PANGAEA,
https://doi.org/10.1594/PANGAEA.878886,
Zimmermann, H.H., Raschke, E., Epp, L.S., Stoof-Leichsenring, K.R., Schirrmeister, L., Schwamborn, G.,
Herzschuh, U. (2017): Pollen profile of sediment core L14-04. PANGAEA,
https://doi.org/10.1594/PANGAEA.878884,
Clumped isotope sample and normalisation data will be uploaded to the EarthChem Library pending publication
of this manuscript.
**10      Executable research compendium (ERC)**
not applicable
**11      Sample availability**



Original samples are available on request in the sample archives of the AWI Research Unit Potsdam.
**12. Video supplement**
not applicable
**13      Supplement link**
Supplement material_Figures
**14      Team list**
not applicable
**15      Author contribution**
LS designed the paper concept, compiled the various results, carried out the cryolithological studies and
evaluations, organized the writing process, and wrote the first manuscript draft. LS, TO, FK, TK, SK, VT, GG,
VK, HaMe, GS, SB, SW, and MCF participated in one or more of the five expeditions to Bol'shoy Lyakhovsky
Island and the Oyogos Yar mainland coast between 1999 and 2014. MCF conducted the geochronological studies.
HaMe was responsible for isotope chemistry studies. A number of co-authors were responsible for certain
paleoproxies and environmental reconstructions based on them: AA - pollen, FK - plant macrofossils, mussels, AS
- ostracods, plant macrofossils, LN - chironomids, LF - cladocerans, SK - insects, TK - mammals, UH, TB - pollen-
based climate reconstructions, HHZ - *seda*DNA, SB, SU, SM - clumped isotope analysis, SK - lipid biomarkers,
SW - ostracods. AP contributed with data on marine MIS 5e deposits. HeMa and GL carried out paleoclimate
modeling. All authors were involved in the data interpretation, took part in the scientific discussions and helped
with writing and editing the manuscript.
**16      Competing interests**
The authors declare that they have no conflict of interest.
**19      Acknowledgements**
We acknowledge funding from the following projects: BMBF project SYSTEM LAPTEV SEA 2000 (03G0134),
INTAS project "Permafrost Dating" (INTAS 8133), IPY project 15 Past Permafrost "From the beginning of the
Pliocene cooling to the modern warming – Past Periglacial Records in Arctic Siberia", DFG project "Late
Quaternary warm stages in the Arctic" (SCHI 975/1-1), RFFI project № 06-05-64197, BMBF project
CARBOPERM "Carbon in permafrost: formation, transformation, and release" (03 G 0836), Leverhulme Trust
(Research Project Grant RPG-2020-334 for project "IsoPerm").



Labs and field parties
We would like to thank all colleagues involved with sample processing in the various laboratories. We would also
like to thank many AWI colleagues and local partners in Tiksi for their excellent and long-standing logistical
support for field work at these remote study sites. This includes the expeditions to Bol'shoy Lyakhovsky Island in
the summer of 1999, 2007 and in spring and summer 2014, as well as the ship tour to the New Siberian Islands in
the summer of 2002 and the expedition to the Oyogos Yar coast in summer 2007.

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



2    **Appendix A**

3    Table A1 Overview of sample collections for cryolithological and fossil proxy studies of Krest Yuryakh deposits
4    exposed at both coasts of the Dmitry Laptev Strait





| Profile | Location number in Table A1 | Year | Coordinates °N, °E | Height [m asl] | Deposits type | cryo-lithology | Pollen | SedaDNA | Plant macro-fossils | Chironomids | Additional bioindicators, clumped isotopes, and IRSL–dated samples | Ostracods | References |
|---|---|---|---|---|---|---|---|---|---|---|---|---|---|
| **Bol'shoy Lyakhovsky Island** | | | | | | | | | | | | | |
| R22+ 60 (R22) | 1 | 1999 | 73.339000 141.295100 | 1-4.8 | pseudo-morph | x | R22+60 S-5, S-6, S-8, r2260s5_80, r22os54_200, r22os53_230 | | R22 OS-53, OS-54, OS-55 | | beetles and other small fossils, R22 B15 h=2.5-2.8 m (OS-53, OS-54); B16 h=4 m (OS-55); R22+30 BL-R-M2 | | Schirrmeister et al. (2000); Andreev et al. (2004); Kienast et al. (2008); https://doi.pangaea.de/10.1594/PANGAEA.880948 |
| R23+ 40 | 2 | 1999 | 73.339400 141.292500 | 1.8-7.3 | lake deposits | x | R23+40 S-1 to S-7 | | | | *Pisidium* sp. shell R23+40–10 | R23+40-S1, -4, -9 | Schirrmeister et al. (2000); Andreev et al. (2009); https://doi.pangaea.de/10.1594/PANGAEA.880949 |
| R32 | | 1999 | 73.344863 141.265733 | | lake deposits | | | | | | R 32 mollusks BL-R-M4 | | Schirrmeister et al. (2000) |
| R35 | 3 | 1999 | 73.345898 141.261696 | 1-5.2 | pseudo-morph | | R35 S-1 to S-11 | | R35 S-9 | R35 S-1 to S-11 | | | Schirrmeister et al. (2000); Ilyashuk et al. (2006); Kienast et al. (2008) |
| L11+ 40 | 4 | 1999 | 73.32330 141.38650 | 0.5-4.3 | lake deposits | x | L11+40 S-1 to S-10 OS-56, OS-57, OS-60 | | OS-56, OS-57, OS-60 | OS-56, OS-57 | beetles and other small animals B-17 h=4-4.3 (OS-56, OS-57); B-19 h=7-7.3 (OS-60, OS-61) | L11-S-6 | Schirrmeister et al. (2000); Kienast et al. (2008); Andreev et al. (2009); https://doi.pangaea.de/10.1594/PANGAEA.880935 |
| L13+ 80 | 6 | 1999 | 73.322300 141.393200 | 5-6.3 | lake deposits | x | | | | | | | Schirrmeister et al. (2000); https://doi.org/10.1594/PANGAEA.880936 |




| Site | No. | Year | Coordinates | Depth | Type | | | | | | | | References |
|---|---|---|---|---|---|---|---|---|---|---|---|---|---|
| L14 | 7 | 1999 | 73.322200 141.393800 | 3-5.4 | pseudo-morph | x | L14 S-1, S-2 | | | | | | Schirrmeister et al. (2000); Andreev et al. (2004); https://doi.pangaea.de/10.1594/PANGAEA.880937 |
| L7-11 A | 8 | 2007 | 73.31672 141.42628 | 4-5.4 | pseudo-morph | x | | | L7-11-07 to -12 | | Insects. daphnia, moss | L7-11-07 to -12 | Boike et al. (2008); Schneider (2010); https://doi.pangaea.de/10.1594/PANGAEA.727667 |
| L7-14 B-C | 9 | 2007 | 73.2877 141.69097 | 2.5-5 | pseudo-morph | x | L7-14-04 to -15 | | | | | L7-14-04 to -15 | Wetterich et al. (2009); Boike et al. (2008); https://doi.pangaea.de/10.1594/PANGAEA.727669 |
| L7-16 | 10 | 2007 | 73.31385 141.4505 | 4.5-7 | pseudo-morph | x | | | | | | | Boike et al. (2008); https://doi.pangaea.de/10.1594/PANGAEA.727669 |
| L14-04 | 11 | 2014 | 73.34100 141.28586 | 3.9-12 core | lake deposits | x | L14-03 6.73 to 12.87, L14.04 6.15 to 8.03 | L14.04 6.15 to 8.03 | | | GDGT | | Zimmermann et al. (2017); https://doi.pangaea.de/10.1594/PANGAEA.868983 https://doi.pangaea.de/10.1594/PANGAEA.878884 |
| L14-04 B | 11 | 2014 | 73.34100 141.28586 | 0.6-5.0 outcrop | lake deposits | x | | L14-04B 2.45, 3.45 | | | GDGT | | Zimmermann et al. (2017) |
| L14-04 C | 11 | 2014 | 73.34100 141.28586 | 2.5-7.5 outcrop | lake deposits | x | L14-04C 2.55 to 4.45 | L14-04C 2.55 to 4.45[LS3] | | | GDGT | | Schwamborn and Wetterich (2015); Zimmermann et al. (2017) |
| L14-12 | 12 | 2014 | 73.34055 141.28498 | 3-7 | lake deposits | | | | | | IR-OSL: L14-12-OSL3, L14-12-OSL1 | | Schwamborn and Wetterich (2015); this study |
| **Oyogos Yar coast** | | | | | | | | | | | | | |
| Oya 3 | 13 | 2002 | 72.679317 143.551417 | 1-4 | pseudo-morph | x | | | | | | | Grigoriev et al. (2003) |
| Oya 5-1 | 14 | 2002 | 72.68000 143.53000 | 3.5 | pseudo-morph | x | Oya 5-1 | | Oya 5-1 | Oya 5-1 | beetles, cladocerans, molluscs, clumped isotopes | Oya 5-1 | Grigoriev et al. (2003); Kienast et al. (2011) |
| Oy7-01 A-C | 15 | 2007 | 72.67454 143.60981 | 1-2.5 | lake deposits | x | | | Oy7-01-01 to 13 | | Insects. Daphnia, moss, mollusks | Oy7-01-01 to 13 | Schneider (2010); https://doi.pangaea.de/10.1594/PANGAEA.727673 |



| | | | | | | | | | | | | | |
|---|---|---|---|---|---|---|---|---|---|---|---|---|---|
| Oy7-07 B | 16 | 2007 | 72.67865 143.55718 | 5-6 | lake deposits | x | | | | | | | Boike et al. (2008); https://doi.pangaea.de/10.1594/PANGAEA.727676 |
| Oy7-08 A-B | 17 | 2007 | 72.68002 143.53181 | 2-6 | pseudo-morph | x | Oy7-08-02 to 24 | | | | | Oy7-08-02 to 24 | Wetterich et al. (2009); Opel et al. (2017) |

