# Peer review of "Newly dated permafrost deposits and their paleo-ecological 1"

_Climate of the Past, 2024_

## Referee Comment (RC1)

**General Comments**
This manuscript essentially summarizes more than two decades of research on stratified ice-wedge-filled lacustrine deposits indicative of interglacial warmth at two nearby sites in NE Siberia: Bol'shoy Lyakhovsky Island and the Oyogos Yar coast. Much of the biostratigraphic evidence has been published previously in specialty journals but is helpfully summarized and compared here, and includes important new geochronological evidence confirming an Eemian (MIS 5e) age and summarizes all of the biostratigraphic evidence in climate terms.  As such, this is a very useful synthesis for the Quaternary community and is well suited for Climate of the Past.  It provides an important synthesis of a massive amount of work, including a wider range of climate proxies than for almost any other LIG Arctic site. I am not especially familiar with these deposits, but there is quite an extensive literature on them.

An OSL expert should review those results as these are essential to the story and as near as I can tell, have not been previously published, whereas much of the other data have been published in specialty journals previously.  I have no reason to be suspicious, although the stated precision is somewhat better than most OSL ages in this time range.

There are an amazing amount of specific analytical results, all at least modestly useful, but certainly not of equal informative power. Still, the less significant results take up less space and document the breadth of effort put into these studies.

Paragraph indents would have been very helpful

**Specific Comments**

**Abstract**  This should be a single paragraph that succinctly explains what is newly published in this paper and what is being summarized from previous (mostly proxy-specific) publications.  And ending with a summary of what the authors think are the key interpretations for Eemian climate (both winter and summer), how these compare with model-based reconstructions and what appears to be the important factor of the higher Eemian sea level, and ending with how what they have learned has relevance for predictions of future Arctic warming in an enhanced greenhouse world.   The current abstract is several paragraphs long and reads more like an Introduction than an Abstract.

Specific comments by page and line number

**p.2 Abstract**
**Line 16**  *seda*DNA not *seda*DNA; looks OK in main text
**Line 11**.  *"new luminescence ages"* Are the luminescence dates new with this publication? If so, then line 8 might best read *"Here, we present new geochronological results and synthesize cryolithological,...."*

**Line 19**:  *"proxy data and also paleoclimate model results indicate a **regional** LIG climate significantly (ca. 5 to 10 °C) warmer than today"*  What region? Maybe make this specific to high northern latitudes?

**p.3 Line 19**  *"The globally warmer-than-today Last Interglacial (LIG, ca. 130-115 ka)*

Do we really know LIG is globally warmer than today? The primary forcing (insolation) is limited to Northern Hemisphere summer and is actually negative for summer in the S Hemisphere. Rising sea levels from NH ice sheets can destabilize some of Antarctica without warming. The Holocene appears to show no early Holocene warmth globally but strong early Holocene warmth in the Arctic…can Eemian be the same? This needs a reference it the authors want LIG warmth to be global.

**p.8 Line 20 and following** *3.2 Luminescence dating*
The section on Luminescence dating is important because it seems to be new results that confirm the age of the deposits to be indeed MIS 5e. Please clarify when the sampling occurred and when the analyses were made and whether the dates cited came from only one of the two sites. Were earlier efforts inconclusive? Is this the first time these MIS-5e dates are being published? Can you show a section where the OSL samples were taken and the context of other biostratigraphic samples in the same section. This seems important to convince the reader that the dates have direct relevance to the climate reconstructions.

There are no citations in Section *4.2 Luminescence dating*, hence I gather these results have not been previously published, and should be reviewed by an OSL expert.

**p.10 Top**
Pollen data are discussed in terms of processing, but no mention of how to deal with pollen from taxa with highly efficient wind-dispersal mechanisms. Particularly *Alnus, Salix, and Betula* that are very efficiently wind transported. However, it appears that actual plant fragments of at least *Betula* and *Alnus* wee recovered. I suggest presenting the plant macrofossil evidence first as its authenticity for on-site plant grow this much higher than for pollen, especially for taxa dependent on wind dispersal of their pollen.
Table 2 is very helpful in this regard
Also a discussion on page 37 addresses some of these issues

**Fig 4** very helpful and convincing for ID of Eemian

**p.12 Section 3.5** *Clumped isotope analysis of biogenic carbonates and derivation of lake water δ18O*
*Wouldn't this make more sense to read: **Clumped isotope derived lakewater paleotemperatures***

**p.30-32** *5.2 Last Interglacial chronology and dating uncertainties*
I'm not sure the summary of the range of ages available is essential here. Seems like focusing on luminescence techniques, as that is all that is presented for age control of these deposits. Other dated deposits are listed but as those results are not really discussed, I don't see why they are relevant to the paper. Although Fig 11 is somewhat helpful even though not particularly relevant to the main thrust of this paper.

Section 5.3 is important and very helpful, as is Table 7

**p.36, line 13** "farther vs further" "farther describes physical distance; further describes figurative distances"

**p.38&39 Conclusions**
This is the one paragraph that most "general readers" will look to.  Page 39 first paragraph discusses the temperature estimates from a range of proxies, especially warmest month.  But it gets a bit muddled on exactly "how much warmer than present day", or pre-industrial, summer temperature estimates they are. It would be very helpful to have a better presentation of
1) Recorded summer temperatures (or estimated pre-recent-warming warmest month temperatures
2) The range of LIG estimated warmest month temperatures for the various proxies and an attempt to summarize how these might be compared to contemporary measured air or lakewater temperatures
3) The modeled Eemian warmest month temperatures

And then the discussion of how a higher sea level during teh Eemian may in its own altered warmest month temperatures
This section is so important to the general reader that a bit more effort to distill all their amazing data into a comprehensive summary is important.

---

## Referee Comment (RC2)

[referee-annotated manuscript omitted]

---

## Author Comment (AC1)

OVERVIEW

The Last Interglacial (LIG) is an important analog for current climate changes, but terrestrial LIG sites are scarce. This manuscript provides a detailed review and paleoclimatic analysis of LIG deposits from lacustrine sediments found in permafrost settings at a series of sites in northern coastal Russia. This is a major synthesis and update, building upon fieldwork conducted over the past 25 years and a remarkably wide-ranging suite of paleoclimatic and paleoecological proxies. (Many prior papers have been published from these sites, this work also serves as a review and synthesis of these papers.) Highlights include new age estimates, new paleoclimatic reconstructions from a wide variety of proxies, and comparisons of the proxy data to paleoclimatic simulations.

This work is also highly valuable given the current geopolitical situation and difficulty of conducting fieldwork in Russia. For at least the next several years, these observations are essentially irreplaceable.

In some ways, this reads as more of a 'Quaternary' paper than a 'paleoclimate' paper, given that the paper includes extensive detail about the stratigraphic sections and ecological interpretations in addition to the paleoclimatic reconstructions. All of the proxies presented provide useful information about local environments and ecosystems, but not all are directly relevant to past climatic interpretations. This paper is very long and very detailed, but its discussion is thorough and all lines of evidence are carefully weighed and integrated.
Thank you for the very encouraging feedback in the overview. We hope that our work will contribute to a better understanding of the MIS 5e interglacial in the Arctic.

I provide below a few comments on the individual proxy-based (and model-based reconstructions), some of which may require revisions to the analyses. These are followed by minor line-by-line comments and see also the attached PDF for recommended edits to make minor grammatical fixes, shorten wording, and sharpen some phrasings.
We have carefully read all the comments and proposed changes in the PDF and have incorporated many of them. Thank you very much.

PROXY-BY-PROXY COMMENTS

Some of the climatic reconstructions are based on the Mutual Climate Range approach (for plants, insects), which is a fairly crude way to derive temperature estimates (Table 2, page 20). But the MCR-based results agree well with each other and other approaches.
Therefore, we believe that even coarser reconstruction approaches, if they are comparable to others, are worth using here.

The pollen-based climatic reconstructions use a modern pollen training dataset that spans the Northern Hemisphere, including Eurasia and North America. I recommend redoing these analyses to remove the North American samples, because they mostly represent different species with somewhat different climatic tolerances from their Eurasian counterparts.
We already used a modern pollen training dataset, including Eurasia and North America, for the pollen-based climate reconstructions (e.g., Andreev et al., 2021). The results show that if we remove the North American pollen spectra from the modern pollen dataset, it does not change the results. Moreover, the North American samples contain the same pollen taxa, although these pollen taxa are produced by different plant species. Basically, the Eurasian and

North American species have similar climatic tolerances, the possible differences are rather insignificant for pollen-based climate reconstructions.

Andreev, A. A., Raschke, E., Biskaborn, B. K., Vyse, S. A., Courtin, J., Böhmer, T., Stoof-Leichsenring, K., Kruse, S., Pestryakova, L. A. and Herzschuh, U. (2021): Late Pleistocene to Holocene vegetation and climate changes in northwestern Chukotka (Far East Russia) deduced from lakes Ilirney and Rauchuagytgyn pollen records, Boreas 50 (3), 652-670. https://doi.org/10.1111/bor.12521

Additionally, the pollen-based climatic reconstructions also do not indicate whether a minimum SCD threshold was used to remove or flag no-analog samples, i.e. fossil samples likely to have no modern analogue.  This no-analog analysis and thresholding should be performed.

We have carried out the dissimilarity analysis requested to identify possible non-analog samples. W created plots for the two locations, "Bol'shoy Lyakhovsky Island" and "Oyogos Yar Coast" (Figure S2)

There is only one sample for which no close modern analogs were found (OY7-8B s4 600), but the minimum dissimilarity is still below the 5% threshold that is usually applied to non-analogs.

We added this sentence in the method chapter:

"The dissimilarity analysis shows very high-quality analogs. All samples are below the 5% threshold, which corresponds to "good analogs"; the majority of the samples even have "close analogs" (threshold < 1%). "

The barplot shows the number of analogs found – the higher the bar, the more modern analogs were found for the corresponding sample.

It's exciting to see that sedaDNA analyses were performed at a core in this site, but it's not clear what is added by these analyses.  No attempt is made to make a paleoclimatic inference based on the sedaDNA data (this is probably wise).  Grouping the aeDNA results into functional groups is reasonable (Figure 7, p. 21) but misses a lot of the interesting ecological detail.  Recommend adding a figure that shows the stratigraphic occurrences of the taxa described in the text on p21, at maximum feasible taxonomic resolution.

Even though we refrain from paleoclimatic reconstructions, comparisons among pollen, macrofossils, and *sed*aDNA have shown that the three proxies contain complementary information and only together provide the most accurate picture. This is highlighted by our sedaDNA results challenging the previous northern distribution limits reconstructed, for example, for *Larix* and *Picea*, thus adding value of this proxy to the overall manuscript.

We hope the following clarification strengthens our approach and addresses your concerns (added to page 22, L22-29):

"We refrain from paleoclimatic inferences as *sed*aDNA (compared to traditional proxies like pollen assemblages) provides only qualitative or semi-quantitative assemblage information, is rather local in the origin of the signals, and the lack of taxonomic resolution to species level in many taxa hampers the accurate inference of past temperatures. However, the northern distribution limit of *Larix* is clearly spatially linked to the 10–12.5°C isotherm (based on its modern ecology (MacDonald et al. 2008)), and the co-occurrence of *Picea* (likely *P. obovata*) suggests an active layer depth of at least 1.5–2 m (Tchebakova et al. 2009). Thus, our results align with other proxy reconstructions presented in this study, supporting the interpretation of warmer-than-present temperatures during the Last Interglacial (LIG)."

Tchebakova et al 2009 Environ. Res. Lett. 4 045013; DOI 10.1088/1748-9326/4/4/045013

MacDonald et al. 2008 Phil. Trans. R. Soc. B 363, 2285–2299; doi:10.1098/rstb.2007.2200

We appreciate your suggestion regarding the inclusion of a figure showing the stratigraphic occurrences of taxa at the highest feasible taxonomic resolution. We would like to note that Zimmermann et al. (2017) has previously published such figures open-access. Given this, we focused our presentation on functional groupings to emphasize broader ecological patterns while avoiding redundancy.

We added the following sentence to the caption of Figure 7:

"For detailed stratigraphic co-occurrences of all taxa at the highest feasible taxonomic resolution, see Zimmermann et al. (2017)."

Similarly, the ostracode and mollusk analyses (pp12-13) are interesting but do not contribute to the paleoclimatic reconstructions shown here.  Delete or move to supplementary information?
We believe that both proxies provide important information for paleoenvironmental reconstruction during the MIS 5e interglacial. In particular, the thermokarst landscapes with lakes would not have been possible without the greatly elevated paleotemperatures during this period. Furthermore, the ostracods and bivalves as proxies provide basic information that is important for the clumped isotope studies.

We suggest highlighting that the reviewer may have misunderstood or missed our use of these specimens. The ostracod and bivalve analyses were used for paleoclimate reconstruction; some temperature information and water oxygen isotope compositions were derived from these specimens. using clumped isotopes.

Paleoclimatic simulations often include a spin-up period, during which the model is in a transient state that is artificially influenced by the starting conditions.  Was this spin-up period removed prior to calculating climatological means from the simulations?
The model spin-up procedure, as part of the PMIP4 protocol, recommends spin-ups similar to those of the PIcontrol simulations for all participating models. Details about the spin-up procedures employed by the different modeling groups are discussed where the simulations were discussed in separate, special papers (e.g., for HadGEM (https://cp.copernicus.org/articles/16/1429/2020/), ACCESS (https://cp.copernicus.org/articles/17/869/2021/), EC-Earth (https://gmd.copernicus.org/articles/14/1147/2021/) or AWI-CM). However, there is no conclusive overview. However, for all models with available details, we can confirm that the modeling groups did not submit their spin-up simulation results to the ESGF, so the spin-up data was not included in the analysis. We assume that none of the modeling groups submitted spin-up data to the ESGF labeled as PMIP4 simulations.

Also, for the paleoclimatic simulations:  On P13, L23-27, I can't quite follow this description of the calculation of the anomalies, which seems to repeat the subtraction used to calculate the anomalies, when it should only be performed once.
In general, we follow the procedure suggested in the IPCC AR6 and also applied, e.g. by Otto-Bliesner et al., where we use the climate change signal from the PMIP models, which are calculated with reference to the pre-industrial simulations of the models to determine their paleoclimate. In order to derive absolute values for the LIG from these climate change signals, we have to add the observed pre-industrial to the climate change signals.
        The calculations are admittedly complicated, which stems mainly from the fact that the database for the observed pre-industrial climate is only available as an anomaly itself, and not

as absolute values, which requires using another dataset that delivers the mean climate of the reference period used in the anomaly calculation for the pre-industrial climate. We have altered the text in the manuscript to describe the process more clearly, and added a formula in the supplement that shows how the absolute model values for comparison with the proxy data were derived.

LINE-BY-LINE COMMENTS

P2L2-4:  Long sentence, awk
We've changed that to two sentences.
"Fossil proxy records in Last Interglacial (LIG, ca. 130-115 ka) lacustrine thermokarst deposits now preserved in permafrost can provide insights into terrestrial Arctic environments during a period when northern hemisphere climate conditions were warmer than today. This period might be considered a potential analog for a near-future warmer Arctic."

P2L31-33:  Model results are presented as anomalies but all proxy-based reconstructions earlier in this paragraph are presented as absolute values, making it difficult to compare the model-based and proxy-based results.
We have included the absolute values derived from the models in the abstract to allow easy comparison.

P2L36:  Again, proxy-model mismatch is highlighted here, but the exact magnitude and nature of the mismatch is unclear.
This was fixed by adding the absolute temperature values derived from the models as well as the present-day values for comparison to modern conditions.

P2L36:  What does 'more systematic' mean?
"More systematic" means a reconstruction approach for the paleoproxies that is as comparable as possible.

P3L7:  'promote' is present tense but prior sentences in past tense.  Check for verb tense consistency throughout ms.
Thank you very much. We have corrected this error.

P3L19-20:  Add ref(s) here to support LIG as analog for future, e.g. (Burke et al. 2018; Gulev et al. 2021; Otto-Bliesner et al. n.d., 2)
Thank you very much. We have added the quotations.
"The globally warmer-than-today Last Interglacial (LIG, ca. 130-115 ka) (Wilcox et al., 2020) is commonly seen as a potential analog for future climate warming (Burke et al., 2018; Gulev et al., 2021; Otto-Bliesner et al., 2013)."

P3L26-27:  reduced ice sheet – reduced by how much?
This is meant as a general statement and should not be supported by specific data here.

P3L35:  A data-model mismatch is invoked here as if already introduced to the reader, but so far no data-model mismatch has been described.  Instead, the paper describes data-model agreement earlier in the paragraph.
Thank you for this comment. We state that there is a general agreement from modelling studies and proxies analyzed for the LIG earlier in the introduction. What we refer to here are

mismatches on a local scale, like shown in our study. However, since this is not the focus here, the sentence was moved to the discussion about the mismatch between MTWA from models and from proxies in section 5.3.

P4L9:  'were conducted'… by who?  Unclear if this is introducing work by the authors or reviewing the broader literature.
The references are listed one line below:
"Both the southern coast of Bol'shoy Lyakhovsky Island near the Zimov'e River mouth and the Oyogos Yar mainland coast near the Kondrat'eva River mouth have been studied for LIG pollen, plant macrofossils, fossil insect remains including beetles and chironomids, lacustrine invertebrates such as ostracods, cladocera, and mollusks, and testate amoebae and sedimentary ancient DNA (Andreev et al., 2004, 2011; Ilyashuk et al., 2006; Kienast et al., 2008, 2011; Wetterich et al., 2009;  Schneider, 2010; Zimmermann et al., 2017). However, only scarce chronological control is available (Andreev et al., 2004; Opel et al., 2017) for lacustrine deposits, which are locally named the Krest-Yuryakh stratum (Tumskoy and Kuznetsova, 2022) and commonly interpreted as deposits of the LIG, i.e., MIS 5e (Eemian)."

P9L5:  Italicize Rumex and all other genus and species names.
Thank you very much. We have corrected all the taxonomic names that were not yet in italics.

P6 Figure 1:  Recommend replacing the 1-16 site labels with the site codes used in Figure 2.  Right now there is no easy way to look between Figures 1 & 2 and understand which sites are which.  Or, add the numeric codes in Fig. 1 to Fig. 2.
Due to a lack of space, the site codes from Figure 2 cannot be entered in Figure 1. Therefore, we decided to write the numbers from Figure 1 in parentheses before the site codes in Figure 2.

P7 Figure 2:  What is a 'taberal' deposit?
We have added this explanation to the caption for Figure 2:
"Taberal deposits means thawed and refrozen permafrost deposits."

P16 Table 1:  A couple of the numbers are using commas instead periods to indicate decimal places.
Thank you. This has been corrected.

P18L21:  'relatively' – relative to what?
We changed the sentence to "Based on the quite high percentage of *Larix* pollen, …"

P19L2:  avoid 'optimum' when referring to a peak in temperature.  Any given temperature will be optimum for some species and suboptimal or adverse for others.
It is not about a specific peak temperature, but rather about the general shift of the tree line during the peak LIG.

P22L6:  Is Morychus viridis a steppe species?  This is implied but unclear.
Yes, *M. viridis* lives today on a specific type of short grass steppe – hemicryophytous steppe (Berman et al., 2001).
Berman, D. I., Alfimov, A.V., Mazhitova, G. G., Grishkan I. B., Yurtsev, B. A., 2001. Cold steppe in north-eastern Asia. Dal'nauka, Vladivostok, 183. (in Russian)….

P22L8:  Which 'other Pleistocene samples' – those at the site, worldwide, or other domain?  Unclear.

It concerns other samples from our study area. The sentence has been modified to clarify this point:
Several identified thermophilous steppe species … are absent in other Pleistocene samples from our sites (Andreev et al., 2004, 2009; Kiselev and Nazarov, 2009),

P26L15: Should A. Kossler be invited to be a co-author, given their intellectual contribution in identifying molluscs and the large interdisciplinary team?
A. Kossler was co-author of the cited paper by Kienast et al. (2011). While preparing the manuscript, we have tried to contact Anette Kossler several times, unfortunately without success.

P28L13-14: agreement with each other? Agreement with the proxy data?
We have completed the sentence: "The PMIP models show the highest agreement with each other over the sea and the lowest agreement over the continental grid cells."

P33-34, Table 7:
*Recommend moving the ERA reanalysis data to the top row for each region, so that this modern reference is clearly established as the basis of comparison for all the paleoestimates.
Done
*Also note that the ERA reference is for 1990-2019, which already includes >1-2C of anthropogenic warming. Consider also adding a row with a temperature estimate for the early 20th century or pre-industrial period.
Thank you for this suggestion. We have added the PIobs values to the table.
*For numbers with a +/- uncertainty estimate, does this represent one or two standard deviations? Clarify in table legend.
We have added a description of the uncertainties to tables 6 and 7.

P34, Fig. 12: Recommend a standard order of first showing the modern values, then the paleoclimatic simulations, then the proxy results. This ordering will better clarify the distinction between paleoclimatic simulations vs. proxy data. Among the proxy results, I suggest moving the clumped isotopes to the bottom of the order, because they represent different variables than the other proxies.
We have changed the order of the data sets such that the data from Bol'shoy Lyakhovsky now come before the data from Oyogos Yar and the clumped isotope data are at the bottom. The further internal order (modern, simulation, proxy) should be retained.

P37L7-10: Another reason why pollen is unlikely to be an important source of aeDNA is a mass effect: the total biomass of airborne pollen rain is very low to all local aeDNA sources, so any pollen-sourced aeDNA is swamped out.
Yes, that is correct. We have added the following sentence:
"Moreover, the relatively low biomass contribution of pollen, combined with its limited endogenous chloroplast DNA content, likely results in the dilution or loss of any *sed*aDNA signal derived from pollen (Also et al., 2024)."

REFERENCES CITED

Burke, K. D., M. Chandler, A. M. Haywood, D. J. Lunt, B. L. Otto-Bliesner, and J. W. Williams. 2018. "Pliocene and Eocene Provide Best Analogues for Near-Future Climates." *Proceedings of the National Academy of Sciences* 115: 13288–93. doi:https://doi.org/10.1073/pnas.1809600115.

Gulev, S. K., P. W. Thorne, J. Ahn, F. J. Dentener, C. M. Domingues, S. Gerland, D. Gong, et al. 2021. "Changing State of the Climate System" eds. V. Masson-Delmotte, P. Zhai, A. Pirani, S. L. Connors, C. Pean, S. Berger, N. Caud, et al. *Climate Change 2021: The Physical Science Basis. Contribution of Working Gorup I to the Sixth Assessment Report of the Intergovernmental Panel on Climate Change.*

Otto-Bliesner, B. L., N. Rosenbloom, E. J. Stone, N. P. McKay, D. J. Lunt, Esther C. Brady, and J. T. Overpeck. "How Warm Was the Last Interglacial? New Model–Data Comparisons." *Proceedings of the Royal Society A* 371: 20130097. doi:https://doi.org/10.1098/rsta.2013.0097.

---

## Author Comment (AC2)

**General Comments**

This manuscript essentially summarizes more than two decades of research on stratified ice-wedge-filled lacustrine deposits indicative of interglacial warmth at two nearby sites in NE Siberia: Bol'shoy Lyakhovsky Island and the Oyogos Yar coast. Much of the biostratigraphic evidence has been published previously in specialty journals but is helpfully summarized and compared here, and includes important new geochronological evidence confirming an Eemian (MIS 5e) age and summarizes all of the biostratigraphic evidence in climate terms. As such, this is a very useful synthesis for the Quaternary community and is well suited for Climate of the Past. It provides an important synthesis of a massive amount of work, including a wider range of climate proxies than for almost any other LIG Arctic site. I am not especially familiar with these deposits, but there is quite an extensive literature on them.

Many thanks for this very friendly assessment.

An OSL expert should review those results as these are essential to the story and as near as I can tell, have not been previously published, whereas much of the other data have been published in specialty journals previously. I have no reason to be suspicious, although the stated precision is somewhat better than most OSL ages in this time range. There are an amazing amount of specific analytical results, all at least modestly useful, but certainly not of equal informative power. Still, the less significant results take up less space and document the breadth of effort put into these studies.

Many thanks for this very friendly assessment. Indeed, the luminescence ages (IRSL) are new and unpublished, and we addressed the remarks of the reviewers on luminescence dating in our revised manuscript and replies.

Paragraph indents would have been very helpful

Thank you for this suggestion. We have made the paragraph indentations.

**Specific Comments**

**Abstract:** This should be a single paragraph that succinctly explains what is newly published in this paper and what is being summarized from previous (mostly proxy-specific) publications. And ending with a summary of what the authors think are the key interpretations for Eemian climate (both winter and summer), how these compare with model-based reconstructions and what appears to be the important factor of the higher Eemian sea level, and ending with how what they have learned has relevance for predictions of future Arctic warming in an enhanced greenhouse world. The current abstract is several paragraphs long and reads more like an Introduction than an Abstract.

Thank you for your thoughtful comments and suggestions regarding the abstract of our manuscript. We appreciate your guidance on refining the abstract to ensure it is succinct while remaining informative. We agree with your suggestion to streamline the abstract and have shortened it considerably. However, we would like to clarify our approach to structuring the abstract, which we believe aligns with the journal's directive that the abstract should be intelligible to the general reader without reference to the text. We intend to provide not only a summary of our findings but also a brief introduction to the topic, which sets the stage for the significance of the new proxy analyses conducted. In response to your comments, we have made efforts to condense the abstract while ensuring that it remains a comprehensive

overview of the study's contributions. We added a summary of the key interpretations for the Eemian climate in both winter and summer. We do not believe that the discussion of sea level differences between models and the real world fits into the abstract. We think that our revised abstract now effectively communicates the core contributions of our work.

Specific comments by page and line number

**p.2 Abstract**

**Line 16**  *seda*DNA not *sedaDNA*; looks OK in main text
Thank you very much. We change the "a" in *seda*DNA from italic to non-italic.

**Line 11**.  "*new luminescence ages*" Are the luminescence dates new with this publication? If so, then line 8 might best read "*Here, we present new geochronological results and synthesize cryolithological,…*"
The abstract was generally rewritten and condensed.

**Line 19***:  "proxy data and also paleoclimate model results indicate a **regional** LIG climate significantly (ca. 5 to 10 °C) warmer than today"*  What region? Maybe make this specific to high northern latitudes?
The abstract was generally rewritten and condensed

**p.3 Line 19**  "*The globally warmer-than-today Last Interglacial (LIG, ca. 130-115 ka)*
Do we really know LIG is globally warmer than today?  The primary forcing (insolation) is limited to Northern Hemisphere summer and is actually negative for summer in the S Hemisphere.  Rising sea levels from NH ice sheets can destabilize some of Antarctica without warming. The Holocene appears to show no early Holocene warmth globally but strong early Holocene warmth in the Arctic…can Eemian be the same?  This needs a reference it the authors want LIG warmth to be global.
We added
Past Interglacials Working Group of PAGES: Interglacials of the last 800,000 years, Rev. Geophys., 54, 162–219 (2016). https://doi.org/10.1002/2015RG000482.
Fischer, H., Meissner, K.J., Mix, A.C. et al. Palaeoclimate constraints on the impact of 2 °C anthropogenic warming and beyond. Nature Geosci. 11, 474–485 (2018). https://doi.org/10.1038/s41561-018-0146-0.
Snyder, C. Evolution of global temperature over the past two million years. Nature 538, 226–228 (2016). https://doi.org/10.1038/nature19798.
Wilcox, P.S., Honiat, C., Trüssel, M., Edwards, R.L., and Spötl, C.: Exceptional warmth and climate instability occurred in the European Alps during the Last Interglacial period. Commun Earth Environ 1, 57 (2020). https://doi.org/10.1038/s43247-020-00063-w.

**p.8 Line 20 and following        *3.2 Luminescence dating***
The section on Luminescence dating is important because it seems to be new results that confirm the age of the deposits to be indeed MIS 5e.  Please clarify (1.) when the sampling occurred and (2.) when the analyses were made and (4.) whether the dates cited came from only one of the two sites.  (3.) Were earlier efforts inconclusive?  Is this the first time these MIS-5e dates are being published?  (5.) Can you show a section where the OSL samples were taken and the context of other biostratigraphic samples in the same section. This seems

important to convince the reader that the dates have direct relevance to the climate reconstructions.

In the text, it was written:

1. "In 2014, Krest-Yuryakh deposits were sampled at the southern coast of Bol'shoy Lyakhovsky Island for luminescence dating. [...] exposure was sampled for luminescence dating at heights of 4.5 m a.s.l. (L14-12-OSL1) and 2.7 m a.s.l. (L14-12-OSL3) (Table A1)."

2. We inserted the information on sample processing in the luminescence laboratory at the Institute of Applied Physics, TU Bergakademie Freiberg and clarified the information on samples involved in sample processing.

3. Earlier dating of such ice-wedge pseudomorphs (see Andreev et al., 2004 and Opel et al., 2017) does not show an exact MIS 5e age. However, it does point to the same age range. The ages published here are the first to fit this time frame directly.

4. We added: "Krest-Yuryakh deposits were sampled **at one site** (L14-12) at the southern coast of Bol'shoy Lyakhovsky Island for luminescence dating

5. We have included a photo of the sampling site with the sampling points as Figure S1 in the supplement.

There are no citations in Section *4.2 Luminescence dating*, hence I gather these results have not been previously published, and should be reviewed by an OSL expert.
That's right. The new ages have not been published yet.

**p.10 Top**
Pollen data are discussed in terms of processing, but no mention of how to deal with pollen from taxa with highly efficient wind-dispersal mechanisms. Particularly *Alnus, Salix, and Betula* that are very efficiently wind transported.  However, it appears that actual plant fragments of at least *Betula* and *Alnus* were recovered.  I suggest presenting the plant macrofossil evidence first as its authenticity for on-site plant grow this much higher than for pollen, especially for taxa dependent on wind dispersal of their pollen.
We completely agree that plant macrofossils evidence better the presence of plants in local vegetation. It is also correct that wind-pollinated plants such as *Betula, Alnus, Pinus, Artemisia,* Poaceae, and many others produce a large number of pollen grains, and their pollen is easily transported by wind. However, only single grains are normally transported for long distances. The biggest part of the pollen of such wind-pollinated plants is accumulated a few hundred meters from the flowering plants. Thus, if pollen percentages in the sediments are relatively high (>10-20%), the producing plants grew in the close vicinity of the study site. Moreover, pollen-based vegetation and climate reconstructions are not based on the presence of single grains but on the pollen percentages.
We prefer to leave the order of the proxy results as it is. Considerably more profiles and sediment samples were analyzed for pollen (10) than for plant macrofossils (5).

Table 2 is very helpful in this regard
Thank you for this friendly comment.

Also a discussion on page 37 addresses some of these issues
 Thank you for this friendly comment.

**Fig 4** very helpful and convincing for ID of Eemian

Thank you for your encouraging comment.

**p.12   Section 3.5 *Clumped isotope analysis of biogenic carbonates and derivation of lake water δ18O***
*Wouldn't this make more sense to read:* **Clumped isotope derived lakewater paleotemperatures**
We calculate the isotopic composition of lake waters in addition to the clumped isotope temperature, but this is not included in the reviewer's suggested title. However, we agree that a more succinct title may be advantageous. We would suggest:
"Clumped isotope derived lake water temperature and δ¹⁸O signatures"

**p.30-32** *5.2 Last Interglacial chronology and dating uncertainties*
I'm not sure the summary of the range of ages available is essential here. Seems like focusing on luminescence techniques, as that is all that is presented for age control of these deposits. Other dated deposits are listed but as those results are not really discussed, I don't see why they are relevant to the paper. Although Fig 11 is somewhat helpful even though not particularly relevant to the main thrust of this paper.
We believe that the new luminescence ages of the MIS5e deposits are a substantial part of the study and worth to be shown as they provide age control for the proxy interpretations. Thus, the aim of showing Fig. 11 is not only to present the newly obtained luminescence ages but also the stratigraphic context of previous dating attempts. Therefore, also the other available MIS5 age determinations from Bol'shoy Lyakhovsky Island and the Oyogos Yar coast are shown here. As the previous age determinations are extensively discussed in the cited references, we focused on highlighting the new dating results. Therefore, we changed the title of section 5.2 to "Luminescence dating results of Last Interglacial deposits". Given the age of these deposits, luminescence dating is indeed the chosen technique.

Section 5.3 is important and very helpful, as is Table 7
Thank you for this encouraging comment!

**p.36, line 13**  "farther vs further" "farther describes physical distance; further describes figurative distances"
corrected

**p.38&39 Conclusions**

This is the one paragraph that most "general readers" will look to.  Page 39 first paragraph discusses the temperature estimates from a range of proxies, especially warmest month.  But it gets a bit muddled on exactly "how much warmer than present day", or pre-industrial, summer temperature estimates they are. It would be very helpful to have a better presentation of

- Recorded summer temperatures (or estimated pre-recent-warming warmest month temperatures
We added the sentence:

"According to ERA5 (1990-2019) simulations, the present-day MTWA of Bol'shoy Lyakhovsky Island and the Oyogos Yar coast is 2.7 and 7.5 °C, respectively, and the MTCO is -32.7 and -34.1 °C."

- The range of LIG estimated warmest month temperatures for the various proxies and an attempt to summarize how these might be compared to contemporary measured air or lakewater temperatures

We added the sentence:
"This suggests summers warmer than today by 5.5 to 12.8°C for Bol'shoy Lyakhovsky Island and by 0.2 to 7.5°C for Oyogos Yar coast, and winters warmer than today by up to 7.1°C and 8.4°C, respectively."

- The modeled Eemian warmest month temperatures

We added the sentence:
"The PIobs+(lig127k-PI) MTWA for Bol'shoy Lyakhovsky and Oyogos Yar are very close to each other (4.4±1.0 and 4.5±1.2 °C) and also the MTCO with –31.1±1.4 and –31.6±1.4, respectively."

And then the discussion of how a higher sea level during teh Eemian may in its own altered warmest month temperatures.
We agree that changes in circulation due to the higher sea level during the Eemian might have influenced the warmest month temperatures. Looking at present day climate, the coastline of Siberia is more impacted by cyclones than areas farther away from the coast. An inward shift of the coastline during the Eemian could, therefore, have caused higher cyclone frequencies at our proxy locations in comparison to the present-day climate, with associated lower temperatures of the warmest month. However, without model simulations that reflect the paleogeography - modeling protocol for the lig127k simulations asked modelers to use present-day land sea masks - it is difficult to investigate what circulation changes in the Eemian compared to today might really look like, and the impacts they may have had on warmest month temperatures would be rather speculative, which is why this was not included in the discussion.
The simple existence of our terrestrial profile already shows that there was no coastline in the study area during the LIG.

In the chapter 5.3 Proxy-based quantitative paleoclimate reconstructions compared to PMIP model simulations are written:
"The differences between the PMIP model's land-sea mask and the actual coastline during the LIG, especially MAP and MTWA, might lead to underestimated MAP and MTWA"

and in the conclusion chapter:
"Paleoclimate models generally agree well with the mean temperature of the coldest month (MTCO) proxy data but consistently underestimate the mean temperature of the warmest month (MTWA) across proxy records when using modern land-sea configurations. This mismatch is significantly reduced when models incorporate land-sea distributions that more closely reflect those of the LIG. This adjustment highlights the importance of considering past land-sea configurations in regional paleoclimate modeling when comparing proxy and model results, a critical step in refining our understanding of Arctic climate dynamics during MIS 5e."

This section is so important to the general reader that a bit more effort to distill all their amazing data into a comprehensive summary is important.

---

## Author Comment (AC3)

**Review of Lutz et al. Climate of the Past, https://doi.org/10.5194/cp-2024-74**

Thank you to the editorial staff and the authors for the extra time for reviewing this paper. My appreciation also goes to the authors and their collaborators for pursuing an archive like this exposed in collapsing coastal bluffs. These sites are very special in the arctic regions and especially important as we are cut off from Russian collaborations for the foreseeable future.
Thank you very much for those very kind words.

I am very excited about this paper overall, but have some concerns about presentation, figures, and context. I hope my comments increase the value of the manuscript. The extension of treeline to the arctic coast in northeastern Arctic Russia during the MIS 5e was first discussed by Lozhkin, A. and Anderson, P.: The last interglaciation in northeast Siberia, Quaternary Res., 43, 147–158, 1995. and then in Lozhkin and Anderson, CP 2013. These papers are about sites to the east of this paper by Lutz but the coherence or contrast is important to the community understanding of the nature of 5e in these remote arctic settings of Arctic Russia. I think Lutz et al should mention this work in the conclusions. Lets hope collaboration with Russian scientists continue but right now we need to celebrate and cite important collaborations that contribute to paleoclimate science in the Arctic.
We will incorporate the aforementioned work by Lozhkin and Anderson into our discussion. We also very much hope that times will improve and that we will be able to cooperate again more effectively with our Russian colleagues.

My comments on the paper are documented by page and by line number.

Abstract page 2 Line 15, Clayish Silt is not used. Should be clayey- silt or clay-rich silt.
Thank you. We changed accordingly.

Page 3 - line 4 -- superimposed "on" local factors.
done

Page 4 -- move Fig 1 closer to where its first used here line 10
We have moved Figure 1 to the introduction, after the first reference to it.

Page 6 -- figure 1 and Figure 1b and 1C have numbered locations that are totally different from the sections (e.g., L14-12) in figure 2. Also Figure 2 b should be flipped so that the east end of the coastline is in the east as shown in Figure 1C. Why not put the section IDs on Figure 1 and not the numbers?
For space reasons, the section IDs were not entered on the maps in Figure 1a, b. Instead, we have now written the section numbers from Figure 1 in front of the section IDs in Figure 2. This should make the individual sections recognizable in both figures.

Page 7 line 2 - eastern parts, Bol'shoy Lyakhovsky Island is shaped by hills reaching elevation of about 100-300 m. Could you include a LIDAR or shaded relief map? Its not shown in Fig. 1. Yedoma or yedoma uplands?
Unfortunately, despite an intensive search, we have not been able to find a more suitable map basis for a better elevation visualization at sufficient resolution. Therefore, we will leave Figure 1 as it was before. The noted hills are except for thin cover deposits built of bedrock and do not refer to Yedoma or Yedoma uplands.

Page 8 line 28 -- something missing in this sentence. Do you want to say " K-feldspars with grain sizes of 40–63 μm were extracted for IRSL dating from core samples L14-12-OSL1, L14-12-OSL3), but coarser grains, 63–90 μm, were extracted from cores at L14-12-OSL1.
Thank you; we have changed this sentence to: Sample preparation targeted K-feldspar extracts at grain sizes of 40–63 μm for IRSL dating from both samples, L14-12-OSL1 and L14-12-OSL3. Additionally, the coarser grain-size fraction 63–90 μm yielded sufficient material for sample L14-12-OSL1. Coarser material (>100 μm) did not provide enough material for further analysis.

Page 8 lines 33-36. Confusing to follow. Extracted medium-sized feldspar grains were used to prepare sets of 24 aliquots with 2 mm diameter?? is 40-63 um medium and what is 2 mm referring to?
Indeed, this was confusing as it covered two aspects, the grain-size fraction and the aliquot diameter comprising a monolayer of grains. We have clarified the respective section in the text.

Page 9 - line 1 -- what is cut-heat temperatures?
This details the SAR protocol. "cut-heat" is the pre-heat in test dose cycles - knowing preheat and cut-heat defines fundamental parameters in the SAR sequence for eliminating instable signal components and needs to be specified. The SAR protocol is the basic concept used to design measurement protocol.

We added information for clarification.

Page 10 - This might need more explanation and it's my lack of knowledge perhaps. Dot occurrences in this GBIF database. I had to go look that up.
We have explained the abbreviation GBIF in more detail:
"The Global Biodiversity Information Facility - GBIF (2023), which …"

And we explained the "dot occurrence"
"The temperature range of a certain species was determined by the correlation of the species occurrences within Yakutia published in GBIF (2023), with the mean July temperature as MTWA at the grid point closest to the respective occurrence with a resolution of 0.5° longitude/latitude (Leemans and Cramer, 1991)."

Page 12 - nitrogen-cooled coldfingers? Part of the instrumentation?
Yes, part of the instrumentation. Whilst this is a widely used term in the clumped community, it might make sense to change it for a wider audience:
"Analyte gas was dehydrated and cleaned following established methodologies (e.g., Bernasconi et al., 2018; Eiler and Schauble, 2004; Petersen et al., 2015). Briefly, $CO_2$ was dehydrated at –80 °C in two liquid nitrogen-cooled water traps (coldfingers) and scrubbed of contaminants by passing through a static cryotrap filled with PorapakTM Q absorbent (Waters Corporation) trap cooled to –30 °C. Traps were baked at 150 °C after each measurement to avoid cross-contamination."

Page 13-14 - The climate modeling was confusing, but I think you were using existing ensemble models from the CMIP6 project, (needs a year) and extracting their data for your field area. So, you are not running the models again but extracting from published models? Is this correct? This section might need a figure. The issue you point out about trying to run the model without knowing what the land configuration was during 5e is a good one so you might

want to bring Figure S3 up into the main paper, not the supplement.  Land Point needs to be capitalized in the figure caption as you have it through the paper.

Thank you for your comment. We changed the manuscript in several places to clarify the paleo modeling part. The heading of section 3.6 was changed to Paleoclimate Modelling Data to emphasize that we use existing data. We added the citation for the CMIP6 description paper to the manuscript. Also, we added text describing that the model data is provided by 13 different modeling groups from all over the world to emphasize that we have used existing model simulations that contributed to a model intercomparison study following a specific protocol. We also changed the figure caption in the supplement.

Regarding the land-sea distribution in the models contributing to PMIP, the protocol stated that paleogeography and ice sheets were to be the same as in the present-day simulations." (Otto-Blienser et al., 2017). In local studies, particularly in coastal areas, this leads to a potential mismatch between the distance to the coast a paleo proxy site experienced in the lig127k and has in the actual model setups, which is what we discuss in section 4.14. While we agree that an overview of the model setups would be interesting in the methods section, due to the length of the manuscript, we prefer to keep that figure in the supplement

Otto-Bliesner, B. L., Braconnot, P., Harrison, S. P., Lunt, D. J., Abe-Ouchi, A., Albani, S., Bartlein, P. J., Capron, E., Carlson, A. E., Dutton, A., Fischer, H., Goelzer, H., Govin, A., Haywood, A., Joos, F., LeGrande, A. N., Lipscomb, W. H., Lohmann, G., Mahowald, N., Nehrbass-Ahles, C., Pausata, F. S. R., Peterschmitt, J.-Y., Phipps, S. J., Renssen, H., and Zhang, Q.: The PMIP4 contribution to CMIP6 – Part 2: Two interglacials, scientific objective and experimental design for Holocene and Last Interglacial simulations, Geosci. Model Dev., 10, 3979–4003, https://doi.org/10.5194/gmd-10-3979-2017, 2017.

Page 14- line 25 -- clay-rich silt, not clayish silt
done.

Page 16 -- line 14. -- TIC values are not shown in Figure 5a.  Typographic error somewhere?   should this be TOC?
Thank you. The reference to Figure 5a will be deleted. It was more about calculating the carbonate estimate, which was made stoichiometrically on the basis of the TIC values.

Results section -- It was long to read but you do go step by step through the proxies and point out how some match and some don't. But that probably to be expected because of calibration errors for each.  Very through.
Thank you very much.

Page 24 -- line 3 -- add reference to Figure 2 for Profile R35.
Done

Page 28 -- Figure 9 and S3 are interesting, and it shows the failure of not having a fine enough grid size.  Not your fault but of models and trying to work that this local scale.
Thank you for your comment. We agree that there is often a distinct mismatch between scales from in situ observations and model grid cells, especially in simulations as coarse as those available for PMIP lig127k. The proxy data presented here represent different spatial scales themselves, but none represent areas as big as the grid cells in PMIP. Still, in situ proxy information about past climates are often the only data sets available for model validation, so the mismatch in scale needs to be taken into consideration when discussing the results. We discuss the uncertainties in the model simulations caused by employing different land sea masks and by not representing paleogeography in section 4.14.

Page 30 line 22-23 -- that first sentence is a mess. I am not sure how to even reword it. perhaps it needs to be 2 sentences?

Thanks for this feedback, we simplified the wording in order to introduce the following details on geochronological results.

Page 31 lines 25-34. Yes, water or ice would impede the dose rate. So, agree that all you can do is assume an uncertainty. And it is good to point out the overburden issue, but I thought it was not an issue below 1-2 meters of the exposed surface. I am not surprised that the age differences between 35 vs. 29 m is tiny.

Yes exactly. We include the uncertainty for the water content because of the unknown effect of water versus ice and related density versus volume effects.

And yes, the effect of cosmic dose rate declines significantly below 2 m, we include the estimates for transparency.

Table 7 page 33 and 34 -- hard to read. I suggest leaving all of the n/a boxes empty so that boxes with data stand out.

done

Page 36 -- around lines 15 and/or 27. Here is where you should cite the work in Chukotka by Lozkhin and Anderson at Lake El'gygytgyn and in an older synthesis the migration of treeline in 5e (QR paper). Your work and theirs paints a very clear image of the near loss of tundra long the coast during that time.

We added in the discussion chapter "5.5 Last Interglacial ecosystems", after the sentence:
"A major discrepancy to prior studies but also a highlight of our record is the detection of Larix, Picea, and Populus in the interglacial strata, suggesting that the treeline reached indeed as far north as Bol'shoy Lyakhovsky Island (73.3 °N)."
The two sentences:
"This is in line with MIS 5e pollen spectra at Lake Elgygytgyn (Lozhkin and Anderson, 2013) that indicate the extensive presence of forest in northern areas of the Russian Far East and the likely establishment of deciduous forest in the Chukchi Uplands. Similar MIS 5e pollen spectra are also known from the Yana Lowland, the Lower Indigirka Basin, and the Kolyma Lowland. (Lozkin and Anderson 1995)."

In addition, we add to the "Conclusions" the sentence:
"The LIG treeline shift is a transregional record that affected both Northeastern Siberia and the Chukchi Peninsula, as well as the Far East."

Page 39 -- lines 10-15 -- very good points here for future work.

Thank you very much.

Page 39 -- Strong summary.

Thank you very much.